# Mitochondrial Fission Process 1 controls inner membrane integrity and protects against heart failure

Erminia Donnarumma[1], Michael Kohlhaas[2], Elodie Vimont[1], Etienne Kornobis [3,4], Thibault Chaze[5], Quentin Giai Gianetto[4,5], Mariette Matondo [5], Maryse Moya-Nilges[6], Christoph Maack [2] & Timothy Wai [1] ✉

Mitochondria are paramount to the metabolism and survival of cardiomyocytes. Here we show that Mitochondrial Fission Process 1 (MTFP1) is an inner mitochondrial membrane (IMM) protein that is dispensable for mitochondrial division yet essential for cardiac structure and function. Constitutive knockout of cardiomyocyte MTFP1 in mice resulted in a fatal, adult-onset dilated cardiomyopathy accompanied by extensive mitochondrial and cardiac remodeling during the transition to heart failure. Prior to the onset of disease, knockout cardiac mitochondria displayed specific IMM defects: futile proton leak dependent upon the adenine nucleotide translocase and an increased sensitivity to the opening of the mitochondrial permeability transition pore, with which MTFP1 physically and genetically interacts. Collectively, our data reveal new functions of MTFP1 in the control of bioenergetic efficiency and cell death sensitivity and define its importance in preventing pathogenic cardiac remodeling.

Mitochondria are multifaceted organelles that are essential in every tissue of the body and are most abundant in the heart, where they control the metabolism and survival of cardiac cells[1]. Mitochondria are double membrane-bound organelles, composed of an inner (IMM) and an outer mitochondrial membrane (OMM), which separate the intermembrane space (IMS) from the matrix. The IMM extends internally to form cristae, which harbor essential macromolecular complexes such as the machinery of oxidative phosphorylation (OXPHOS). The OXPHOS system is comprised of two functional entities: the electron transport chain (ETC) and the phosphorylation system, which includes the integral membrane ATP synthase and carriers such as the adenine nucleotide translocase (ANT), which catalyzes ADP uptake and ATP release in energized mitochondria[2]. The ETC is composed of four macromolecular complexes (I, II, III, and IV) and of mobile electrons carriers as coenzyme Q (CoQ) and cytochrome-c (Cyt $c$). The energy available for ATP synthesis is directly derived from the membrane potential ($\Delta\Psi$) and proton motive force generated across the IMM by electron transfer from by the ETC, which is then harnessed by the ATP synthase (complex V) to generate ATP[3]. Continuous generation of ATP is fundamental for the function of cardiomyocytes, enabling them to meet the enormous energy requirement for the contraction–relaxation cycles that drive their contractility[4]. Defects that impair OXPHOS assembly and function can promote fatal cardiomyopathies[5–9]. The beating heart demands near-maximum OXPHOS capacity, with scant aerobic ATP reserves under normal conditions[10]. Consequently, even modest uncoupling of the ETC from ATP synthesis, which can occur when protons are diverted back across the IMM through uncoupling channels rather than being used by the ATP synthase, would be expected to yield fatal consequences for cardiac function and health, although this has never been tested.

[1]Institut Pasteur, Mitochondrial Biology Group, CNRS UMR 3691, Université Paris Cité, Paris, France. [2]Department of Translational Research, Comprehensive Heart Failure Center (CHFC), Medical Clinic 1, University Clinic Würzburg, Würzburg, Germany. [3]Institut Pasteur, Biomics Technological Platform, Université Paris Cité, Paris, France. [4]Institut Pasteur, Bioinformatics and Biostatistics Hub, Université Paris Cité, Paris, France. [5]Institut Pasteur, Proteomics Core Facility, MSBio UtechS, UAR CNRS 2024, Université Paris Cité, Paris, France. [6]Institut Pasteur Ultrastructural Bio Imaging, UTechS, Université Paris Cité, Paris, France. ✉e-mail: timothy.wai@pasteur.fr

While most famous for their role as the powerhouse of the cell, mitochondria have proven to be essential for cardiac homeostasis through the regulation of various biosynthetic and signaling functions beyond OXPHOS, such as calcium buffering, reactive oxygen species (ROS) generation and maintenance, programmed cell death (PCD) and innate immune responses[11]. Under stress conditions, extrinsic or intrinsic signals can lead to a permeabilization of the outer membrane (MOMP) resulting in activation of caspases and PCD[12]. At the IMM, long-lasting opening of the mitochondrial permeability transition pore (mPTP) allows for unselective diffusion of low molecular weight solutes and water (<1.5 kDa), causing an osmotic pressure in the matrix that causes mitochondrial swelling[13] and rupture, thereby releasing pro-apoptotic and pro-inflammatory mitochondrial factors into the cytosol[12]. While the molecular identity of the mPTP is still the subject of feverous debate, to date several factors have been identified to be unequivocally crucial for its activation. Cyclophilin D (CYPD, encoded by *Ppif*), a mitochondrial matrix isomerase which can be inhibited pharmacologically by cyclosporin A (CsA), calcium overload and ROS are known to promote mPTP opening and PCD induction[13–15]. Notably, ablation of CYPD or treatment with CsA can protect animals from cardiomyocyte death and cardiomyopathy induced by genetic[16], infectious[17], and surgical lesions[15].

The maintenance of mitochondrial morphology and structure is of critical importance for cardiac function[18]. Mitochondrial morphology is regulated by opposing forces of mitochondrial fusion and division, which must be tightly regulated to ensure organellar function and quality control[19,20]. Intrinsic[16,21–24] or extrinsic[25–27] lesions that upset the balance between mitochondrial fission and fusion have devastating consequences for cardiomyocyte function and cardiac health. Mitochondrial dynamics is orchestrated by dynamin-like GTPases: OPA1 and mitofusins (MFN1/2) execute inner and outer membrane fusion, respectively, while DRP1 performs mitochondrial constriction and division once recruited to the OMM via interactions that require integral membrane proteins and receptors, such as MFF, MiD49/51, and FIS1. DRP1 is the lynchpin of the mitochondrial fission apparatus, which is triggered to coalesce at contact sites with the ER, lysosomes, trans-golgi network, and actin by signals that can originate both outside[19,28] and inside[29–31] mitochondria.

While it is unclear how IMM fission is executed, the aptly-named inner membrane protein Mitochondrial fission process 1 (MTFP1/MTP18), has emerged as a promising scaffold for the IMM division apparatus, whose formal identification remains elusive[20]. As per its namesake, *Mtfp1* was initially identified as a gene whose ablation was reported to reduce mitochondrial fission, as well as the proliferation and viability of cultured cells[31,32]. The pro-apoptotic effects of *Mtfp1* depletion have been replicated in various cell lines[33,34], however, more recent studies by the Li group[34–36] have reported the contrary in cultured cells, thus fueling the existing narrative that elongated mitochondria resulting from MTFP1 depletion protects cells from PCD. Whether this paradigm holds true in vivo has never been explored.

In this study, we created a cardiomyocyte-specific *Mtfp1* knockout mouse model to specifically investigate the role of this protein in vivo. Contrary to previous in vitro studies[34], we show that MTFP1 plays an essential role in maintaining cardiac energy metabolism as its deletion in post-natal cardiomyocytes drives a progressive dilated cardiomyopathy (DCM) culminating in HF and middle-aged death in mice. Surprisingly, MTFP1 ablation does not appreciably alter mitochondrial morphology in the heart and is entirely dispensable for mitochondrial fission. Unexpectedly, we discovered that MTFP1 depletion reduces OXPHOS efficiency in cardiac mitochondria by increasing proton leak through the adenine nucleotide translocase (ANT). Finally, we show MTFP1 ablation increases mPTP opening and renders cardiomyocytes and embryonic fibroblasts more sensitive to PCD. Altogether, our data reveal an unexpected role of MTFP1 in mitochondrial bioenergetics and provide mechanistic insights into how MTFP1 regulates the life and death of the cell.

## Results

### *Mtfp1* deletion in cardiomyocytes causes dilated cardiomyopathy and middle-aged death

MTFP1 is predicted to localize in mitochondrial inner membrane (Fig. 1a), and *MTFP1* is highly expressed in human cardiac tissue (GTEx plot, Fig. S1a). To investigate the importance of MTFP1 for cardiac function we began by confirming its expression and submitochondrial localization in the mouse heart. Protease protection assays and alkaline carbonate extraction experiments performed on isolated cardiac mitochondria allowed us to demonstrate the inner membrane localization of MTFP1 in cardiac mitochondria (Fig. S1b, c). To investigate its role in vivo, we generated a conditional mouse model for *Mtfp1* (*Mtfp1LoxP/LoxP*) on a C57Bl6/N background. Conditional mice were crossed with transgenic mice expressing Cre recombinase under the control of alpha myosin heavy chain promoter[37] (Myh6) to specifically ablate MTFP1 in post-natal cardiomyocytes (Fig. 1b, S1d). We previously showed that genetic deletion mediated by Myh6-Cre occurs during the perinatal period[24] and by 8 weeks of age cardiomyocyte-specific *Mtfp1* KO mice (Myh6-CreTg/+ *Mtfp1LoxP/LoxP*, cMKO mice) exhibited a 7-fold reduction in *Mtfp1* mRNA (Fig. 1c) and undetectable levels of MTFP1 protein in cardiac lysates assessed by immunoblot analysis (Fig. 1d) and shotgun proteomics (Fig. 1e, Supplementary Data 1). Both male and female cMKO mice were generated according to Mendelian proportions and were outwardly normal and viable. However, cMKO mice had significantly shortened lifespans [median life span: 26.4 weeks (male), 37.5 weeks (female)] relative to wild type (WT) littermates (Fig. 1f, S1e), demonstrating that MTFP1 is required to protect against middle-aged death.

To directly assess the importance of MTFP1 for cardiac structure and function, we performed longitudinal echocardiographic (echo) studies in male and female mice (Fig. 1g–l, S1f–k). Echo analyses beginning at 10 weeks of age revealed normal cardiac structure and function in cMKO mice. By 18 weeks, however, despite normal cardiac structure we observed a progressive decrease in systolic function, culminating in dilated cardiomyopathy (DCM) and left ventricle (LV) remodeling. By 34 weeks, cMKO mice exhibited all the hallmarks of DCM and heart failure (HF): significantly reduced LV ejection fraction [% EF, WT 60.31 ± 5.4% versus cMKO 25.87 ± 11.8%; Fig. 1h], thinning of the interventricular septum in systole [IVS (mm)], WT 1.23 ± 0.097 versus cMKO 0.9 ± 0.092; Fig. 1i, posterior wall during systole [PWs (mm)], WT 1.39 ± 0.10 vs. cMKO 0.952 ± 0.17; Fig. 1j, dilated LV chamber during the cardiac cycle of diastole [LVDD (mm), WT 4.00 ± 0.15 versus cMKO 5.02 ± 0.65, Fig. 1k] and systole [LVSD (mm)], WT 2.72 ± 0.16 vs. cMKO 4.43 ± 0.84; Fig. 1l with pulmonary congestion (Fig. S1l) and increased heart mass at severe HF (Fig. S1m).

To determine whether the progressive cardiac contractile dysfunction observed in cMKO mice was caused by primary defects in cardiomyocyte function, we assessed sarcomere length and shortening (Fig. S1n–p) coupled to intracellular Ca²⁺ levels [Ca²⁺]c (Fig. S1q–s) in field-stimulated cMKO cardiomyocytes isolated from pre-symptomatic mice (8–10 weeks of age). Diastolic and systolic sarcomere length as well as [Ca²⁺]c transients were similar between WT and cMKO myocytes both at baseline (0.5 Hz stimulation frequency) and during β-adrenergic stimulation (5 Hz stimulation frequency) despite a modest increase of sarcomere shortening of cMKO myocytes under stress conditions and a normal sarcomere re-lengthening at baseline conditions. Together, these data indicate that *Mtfp1* deletion does not impinge upon cardiomyocyte excitation-contraction coupling before the onset of cardiomyopathy in vivo.

Histological analyses of cMKO hearts of mice at HF (34 weeks) confirmed defects in cardiac structure: hematoxylin-eosin (H&E) staining of cardiac cross sections showed LV chamber expansion and

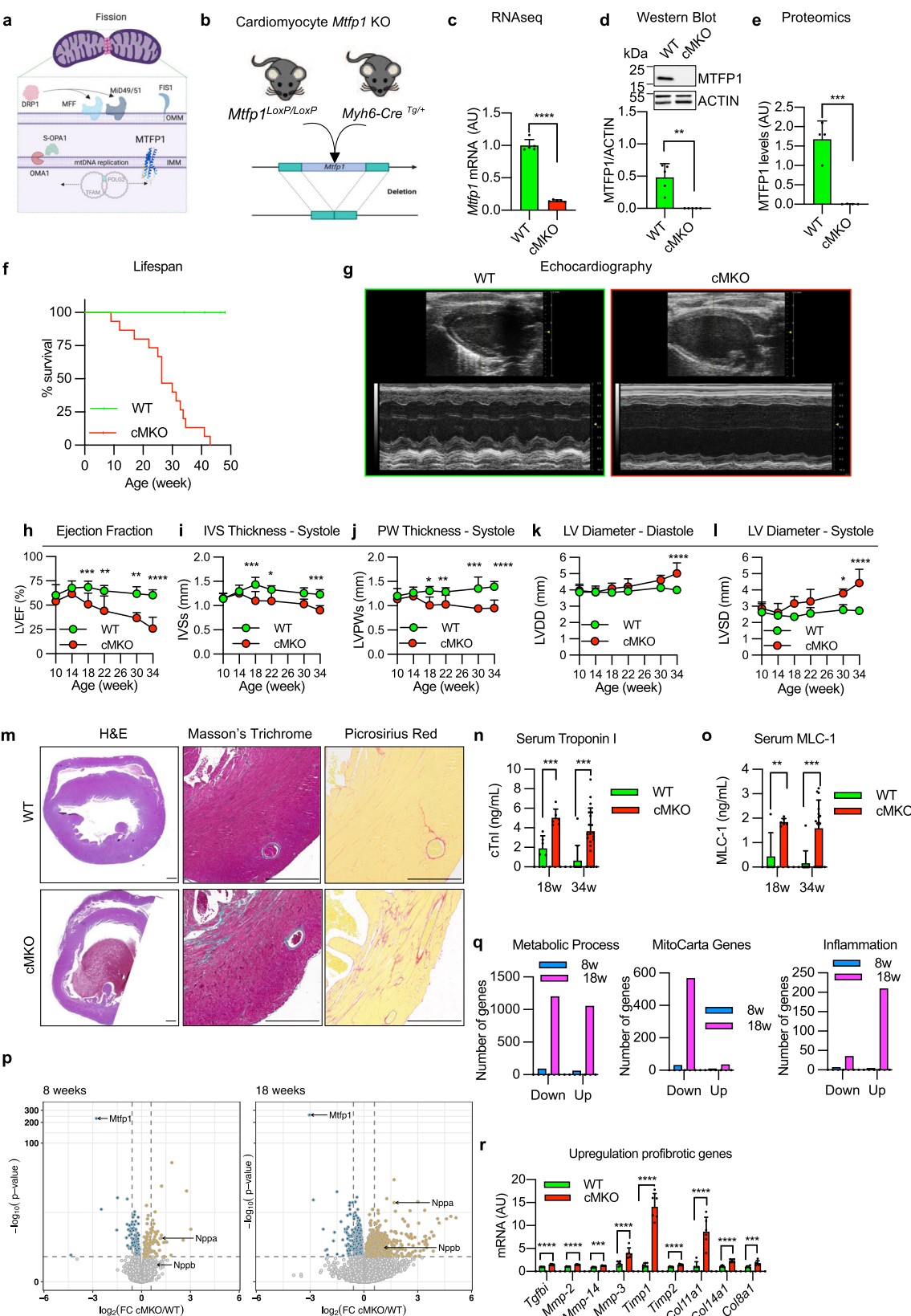

myocardial wall thinning, while Masson's Trichrome and Picrosirius Red staining showed disruption of the myofibril architecture by dramatic fibrosis and collagen deposition at DCM (Fig. 1m). We also found increased serum levels of cardiac troponin I (cTNI, Fig. 1n) and myosin light chain 1 (MLC-1, Fig. 1o) in cMKO mice sampled at 18 and 34 weeks, further substantiating the ongoing cardiomyocyte damage and death. We did not observe sex differences in the development of the cardiac

dysfunction (Fig. S1f–k), highlighting the essential nature of MTFP1 for cardiac structure and function.

## Metabolic and inflammatory gene expression dysregulation in cMKO mice

To gain further insights into the molecular and cellular mechanisms underlying cardiac pathology in cMKO mice, we performed

**Fig. 1 | *Mtfp1* deletion in cardiomyocytes causes dilated cardiomyopathy and middle-aged death in mice. a** AlphaFold prediction of MTFP1 at the inner mitochondrial membrane (IMM). DRP1 binds to its receptors MFF or MiD49/51 to initiate mitochondrial constriction. IMM fission occurs with mtDNA replication mediated by TFAM and POLG2. S-OPA1 accumulation by OMA1 accelerates fission. Figure created with BioRender. **b** Generation of a cardiomyocyte-specific *Mtfp1* KO (cMKO) mouse model. **c** *Mtfp1* mRNA expression (RNAseq arbitrary units; AU) in heart tissue of WT ($n = 5$) and cMKO ($n = 5$) mice at 8 weeks (Supplementary Data 2). Data represent mean ± SD; 2-tailed unpaired Student's *t*-test, ****$p < 0.0001$. **d** Quantification of immunoblot analysis of cardiac lysates from WT ($n = 5$) and cMKO ($n = 5$) male mice at 8 weeks using the indicated antibodies. Data represent mean ± SD; Unpaired *t*-test, **$p < 0.01$. **e** MTFP1 protein expression in cardiac tissue of WT ($n = 4$) and cMKO ($n = 4$) at 18 weeks measured by mass spectrometry (MS) (Supplementary Data 1). Data represent mean ± SD; 2-tailed unpaired Student's *t*-test, ***$p < 0.001$. **f** Kaplan-Meier survival curve of WT ($n = 9$) and cMKO ($n = 15$) male mice. Median lifespan of cMKO mice is 26.4 weeks. **g–l** Longitudinal echocardiography of WT and cMKO male mice from 10 to 34 weeks of age. **g** Representative M-Mode echocardiographic images of left ventricles from WT (left) and cMKO (right) of male mice at 34 weeks. **h** Left ventricular ejection fraction (% LVEF) **i** Systolic interventricular septum thickness (IVSs, mm), **j** Left ventricle posterior wall thickness at systole (LVPWs, mm), **k** Left ventricle end diastolic diameter (LVDD, mm), **l** Left ventricle end systolic diameter (LVSD, mm) of WT and cMKO male mice at indicated ages. Data represent mean ± SD; 2way Anova–Sidak's multiple comparison test: 10 week WT ($n = 13$) vs cMKO ($n = 18$); 14 week WT ($n = 4$) vs. cMKO ($n = 6$); 18 week WT ($n = 4$) vs cMKO ($n = 7$) except for LVEF: WT ($n = 10$) vs cMKO ($n = 13$); 22 week WT ($n = 4$) vs cMKO ($n = 6$); 30 week WT ($n = 3$) vs cMKO ($n = 7$); 34 week WT ($n = 4$) vs. cMKO ($n = 10$): *$p < 0.05$; **$p < 0.01$; ***$p < 0.001$; ****$p < 0.0001$. **m** Representative histological images of cardiac short axis view of WT ($n = 4$) and cMKO ($n = 4$) at 34 weeks. H&E (left), Massons's Trichrome (middle) and Picrosirius red (right) staining show cardiac remodeling and collagen deposition within the myocardium of cMKO mice. Scale bar 500 μM. **n** Cardiac troponin-I (cTNI) measured in serum of WT and cMKO male mice at 18 [WT ($n = 6$) vs cMKO ($n = 6$)] and 34 weeks [WT ($n = 11$) vs cMKO ($n = 20$)]. Data represent mean ± SD; 2-tailed unpaired Student's *t*-test at 18 weeks and 34 weeks (w); ***$p < 0.001$. **o** Myosin light chain 1 (MLC-1) levels measured in serum of WT and cMKO mice at 18 [WT ($n = 5$) vs cMKO ($n = 6$)] and 34 weeks (w) [WT ($n = 11$) vs cMKO ($n = 17$)]. Data represent mean ± SD; 2-tailed unpaired Student's *t*-test at 18 w and 34 w; **$p < 0.01$ and ***$p < 0.001$. **p** Volcano plots generated from the RNAseq analysis (Supplementary Data 2) of the differentially expressed genes in cardiac tissue of WT and cMKO male mice at 8 (left) and 18 weeks (right). **q** Numbers of genes up-regulated and down-regulated in cMKO male mice at 8 (blue) and 18 (pink) weeks (w) within the gene ontology (GO) term: metabolic process (left), mitochondrial genes (MitoCarta, middle) and inflammation (right) obtained from RNAseq analysis (Supplementary Data 2). **r** Expression of indicated profibrotic genes in heart tissue of WT ($n = 6$) and cMKO ($n = 6$) male mice at 18 weeks by RNAseq (arbitrary units; AU). Data represent mean ± SD; 2-tailed unpaired Student's t-test; ***$p < 0.001$, ****$p < 0.0001$.

transcriptomic analyses by bulk RNA sequencing (RNAseq) of LVs isolated from WT and cMKO mice at a pre-symptomatic stage (8 weeks of age) and at the onset of DCM (18 weeks). We observed up-regulation of *Nppa* in pre-symptomatic cMKO mice, which was associated to up-regulation of *Nppb* at the onset of DCM, prototypical cardiomyocyte stress-response genes that are activated in response to hemodynamic load[38] and metabolic or contractile abnormalities[39] (Fig. 1p). At the pre-symptomatic stage, we observed limited transcriptional remodeling in hearts from cMKO mice with differential expression of only ~1% of 25815 genes: 137 genes were downregulated, and 122 genes were upregulated in cMKO mice (Supplementary Data 2, Fig. 1p), whereas at 18 weeks, we observed a much broader transcriptional response with 3642 differentially expressed genes in cMKO mice manifesting signs of DCM (Supplementary Data 2, Fig. 1p).

Functional enrichment analyses performed with g:Profiler[40] and Enrichr[41] revealed a number of dysregulated genes involved in various metabolic processes (Fig. 1q, left). Among the downregulated genes with the gene ontology (GO) term metabolic process (Supplementary Data 2) were genes required for OXPHOS, TCA cycle, fatty acid oxidation and pyruvate metabolism, suggestive of cardiometabolic changes previously observed in mitochondrial models of HF[8,11,24,42]. In fact, further examination revealed that half of all mitochondrial genes referenced on MitoCarta[43] were downregulated (Fig. 1q, middle). On the other hand, we observed a strong inflammatory gene expression signature and innate immune engagement (Fig. 1q, right), which together with dysregulated extracellular matrix-remodeling genes [profibrotic cytokines such as TGFb, collagen precursor genes (*Col11a1, Col14a1, Col8a1*) and matrix metalloproteinases (*Mmp-2, Mmp-14, Mmp-3*)] (Fig. 1r) corroborates the cardiac fibrosis revealed by histological analysis of cMKO hearts (Fig. 1m). Notably, the suppression of mitochondrial gene expression and the activation of sterile inflammation measured in cMKO mice at 18 weeks was absent in pre-symptomatic cMKO mice, implying that these transcriptional responses are downstream consequences *Mtfp1* deletion in adult cardiomyocytes.

## Mtfp1 is required for bioenergetic efficiency in cardiac mitochondria

To directly assess the effects of energy metabolism in cMKO mice, we measured mitochondrial respiration in cardiac mitochondria from WT and cMKO mice (Fig. 2a). High resolution respirometry studies showed a general impairment of mitochondrial $O_2$ consumption in cMKO hearts at both early and late stage of DCM: complex I-, complex II- and complex IV-driven mitochondrial respiration were all significantly lower at either age (Fig. 2b, c). Reduced mitochondrial respiration was not accompanied by a global reduction in all mitochondrial proteins, according to proteomic analyses of cardiac tissue (Fig. S2a, left), including those involved in the assembly and structure of the OXPHOS system (Fig. S2b, Supplementary Data 1), indicating that bioenergetic decline was not the result of increased wholesale mitophagy. However, BN-PAGE analysis of OXPHOS complexes revealed a modest reduction in Complex III and Complex V levels in failing cMKO hearts (Fig. S2c (right), Supplementary Table 1), which is consistent with concomitant disruption of mitochondrial respiration.

Since reduced mitochondrial content and/or OXPHOS activity has been observed in other genetic models of cardiomyopathy[44], we wondered whether impaired mitochondrial respiration observed in failing cMKO hearts reflected the consequences of cardiac dysfunction and cardiac remodeling, rather than the ablation of an essential component or regulator of OXPHOS. We therefore turned our attention to cMKO mice at the pre-symptomatic stage. We observed no general reduction in mitochondrial protein content (Fig. S2a, right), Supplementary Data 1) nor OXPHOS complexes of cardiac tissue from pre-symptomatic cMKO mice (Fig. S2c (left)) nor deficits of the mitochondrial Krebs cycle in intact field-stimulated cMKO cardiomyocytes isolated at 8–10 weeks of age (Fig. S2d), assessed by determining the autofluorescence of NAD(P)H/NAD(P)$^+$ and FADH$_2$/FAD, intrinsic biomarkers of mitochondrial metabolic activity. NAD(P)H/FAD redox state was monitored during a protocol in which cells were exposed to an increase in stimulation frequency and β-adrenergic stimulation[45], creating a typical ADP-induced oxidation ("undershoot") followed by Ca$^{2+}$-dependent regeneration ("recovery") behavior[46] (Fig. S2d). This behavior was similar between cMKO and WT myocytes, ruling out gross alterations in Krebs cycle activity and Ca$^{2+}$-induced redox adaptation pre-symptomatically.

To uncover the functional alterations of OXPHOS that potentiate the development of DCM, we performed bioenergetic measurements in cardiac mitochondria isolated from pre-symptomatic mice. Using a dual respirometer-fluorimeter (O2k, Oroboros), simultaneous kinetic measurements of oxygen consumption rates (JO$_2$) and mitochondrial membrane potential (ΔΨ) were performed in cardiac mitochondria isolated from pre-symptomatic WT and cMKO mice (8–10 weeks). Mitochondria were pre-labeled with the potentiometric dye Rhodamine 123 (RH123) and energized with substrates whose metabolism promotes complex I- [state 2; pyruvate, glutamate, and malate (PGM)]

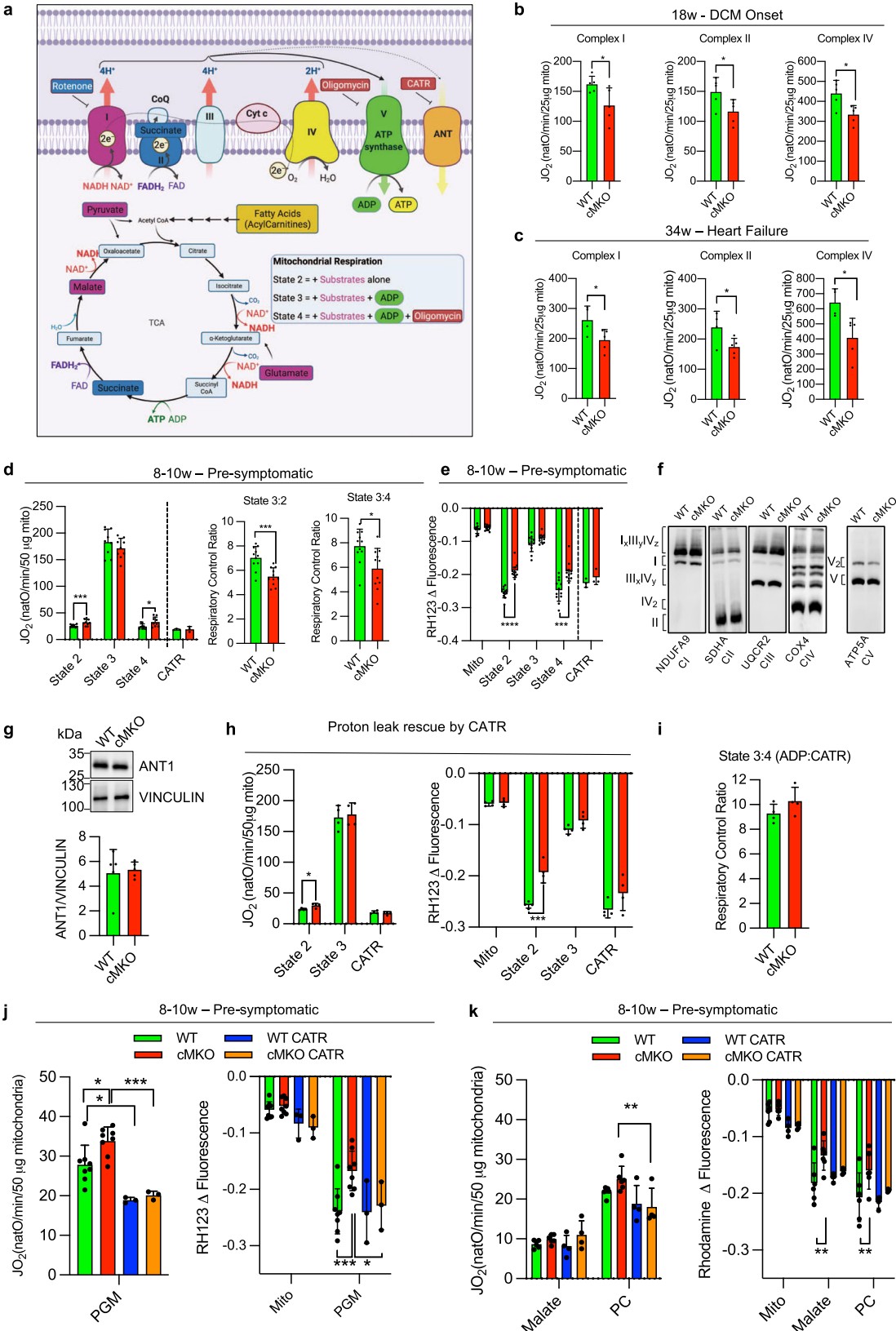

or complex II-linked respiration (state 2; succinate in the presence of rotenone), followed by ADP to promote phosphorylating (state 3: ADP) respiration. Finally, the ATP synthase inhibitor oligomycin (Omy) was applied to assess non-phosphorylating (state 4) respiration (Fig. 2a).

We observed no differences in state 3 $JO_2$ rates between WT and cMKO cardiac mitochondria fueled with either PGM (Fig. 2d (left),

succinate with rotenone (Fig. S2e (left)) or palmitoyl-carnitine (PC) with malate, respectively (Fig. S2g). However, respiration was ~30% higher in cMKO mitochondria in state 2 (+27.4%) and state 4 (+30.3%) when complex I was fueled with PGM (Fig. 2d), significantly decreasing respiratory control ratios [State 3:2, State 3:4, Fig. 2d (middle, right)]. In addition, altered $JO_2$ rates in cMKO mitochondria were

**Fig. 2 | Mtfp1 is required for bioenergetic efficiency in cardiac mitochondria.**
**a** Substrates from fatty acid oxidation (mustard) and glycolysis (purple, blue) are metabolized in the TCA cycle which fuels the electron transport chain (ETC) complexes located in the inner mitochondrial membrane by providing NADH and FADH to complexes I (purple) and II (blue), respectively. Complexes I, III and IV extrude protons from matrix into the intermembrane space creating an electrochemical gradient driving the phosphorylation of ADP at the ATP synthase (complex V). The electron flow is limited by the availability of oxygen, a terminal acceptor of electrons at the complex IV (cytochrome oxidase). Uncoupling proteins such as ANT promote a proton leak, playing an important role in regulation of membrane potential and oxidative phosphorylation efficiency. Specific inhibitors of complex I (rotenone), complex V (oligomycin), and ANT (carboxyatractyloside, CATR). Figure created with BioRender. **b** Oxygen consumption rates (JO$_2$) of cardiac mitochondria from WT ($n = 5$) and cMKO ($n = 5$) male mice at 18 weeks measuring complex I driven respiration (left) in presence of pyruvate, malate, glutamate (PGM) and ADP followed by the addition of rotenone and succinate to assess complex II driven respiration (middle) and antimycin A, carbonyl cyanide m-chlorophenyl hydrazine (CCCP), and N,N,N′,N′-Tetramethyl-p-phenylenediamine (TMPD) and ascorbate to measure complex IV driven respiration (right). Data represent mean ± SD; unpaired Student's t-test, *$p < 0.05$. **c** Oxygen consumption rates (JO$_2$) of cardiac mitochondria from WT ($n = 4$) and cMKO ($n = 5$) female mice at 34 weeks measuring complex I driven respiration (left) in presence of pyruvate, malate, glutamate (PGM) and ADP followed by the addition of rotenone and succinate to assess complex II driven respiration (middle) and antimycin A, carbonyl cyanide m-chlorophenyl hydrazine (CCCP), and N,N,N′,N′-Tetramethyl-p-phenylenediamine (TMPD) and ascorbate to measure complex IV driven respiration (right). Data represent mean ± SD; unpaired Student's t-test, *$p < 0.05$. **d** Oxygen consumption rates (JO$_2$) of cardiac mitochondria isolated from WT and cMKO female mice between 8–10 weeks (left). Respiration was measured in presence of pyruvate, malate, and glutamate (PGM) (state 2) followed by the addition of ADP (state 3), Oligomycin (Omy- state 4) (WT $n = 9$, cMKO $n = 9$) and Carboxyatractyloside (CATR) (WT $n = 3$, cMKO $n = 3$). Data represent mean ± SD. Multiple t-test, state 2 ***$p < 0.001$, state 4 *$p < 0.05$. Respiratory control ratios (RCR) of state 3:2 (middle: JO$_2$ ADP/PGM) and RCR of state 3:4 (right: JO$_2$ ADP/Omy). Data represent mean ± SD; 2-tailed unpaired Student's t-test, *$p < 0.05$, ***$p < 0.001$. **e** Mitochondrial membrane potential (ΔΨ) measured by quenching of Rhodamine 123 (RH123)

fluorescence in cardiac mitochondria isolated from WT and cMKO female mice between 8-10 weeks. ΔΨ was measured in presence of pyruvate, malate, and glutamate (PGM) (state 2) followed by the addition of ADP (state 3) and Oligomycin (state 4) (WT $n = 12$, cMKO $n = 12$) and Carboxyatractyloside (CATR) (WT $n = 3$, cMKO $n = 3$). Data represent mean ± SD; Multiple t-test, ***$p < 0.001$,****$p < 0.0001$. **f** Representative BN-PAGE immunoblot analysis of cardiac OXPHOS complexes isolated from WT and cMKO male mice at 8–10 weeks using the indicated antibodies, repeated on biological replicates WT ($n = 4$) and cMKO ($n = 3$) samples (see Fig. S2c, left) with similar results. **g** Equal amounts of protein extracted from WT ($n = 5$) and cMKO ($n = 5$) hearts of male mice between 8–10 weeks were separated by SDS–PAGE and immunoblotted with the indicated antibodies and quantified by densitometry using VINCULIN as a loading control. Data represent mean ± SD. **h** Oxygen consumption rates measured by high-resolution respirometry (left; JO$_2$) and mitochondrial membrane potential (right; ΔΨ) measured by quenching of Rhodamine 123 (RH123) fluorescence in cardiac mitochondria of WT ($n = 4$) and cMKO ($n = 4$) female mice between 8-10 week of age. JO$_2$ and ΔΨ were measured in presence of pyruvate, malate, and glutamate (PGM, state 2) followed by the addition of ADP (state 3) and Carboxyatractyloside (CATR) (state 4). Data represent mean ± SD; Multiple t-test, *$p < 0.05$, ***$p < 0.001$. **i** Respiratory control ratio (RCR) of state 3:4 (ADP/CATR) between WT ($n = 4$) and cMKO ($n = 4$) calculated from **h**. Data represent mean ± SD. **j** Oxygen consumption rates measured by high-resolution respirometry (left; JO$_2$) and mitochondrial membrane potential (right; ΔΨ) measured by quenching of Rhodamine 123 (RH123) fluorescence in cardiac mitochondria of WT and cMKO female mice between 8–10 week of age. JO$_2$ and ΔΨ were measured by adding pyruvate, malate, and glutamate (PGM, state 2) after the pre-treatment (WT $n = 3$, cMKO $n = 3$) or not of mitochondria (WT $n = 8$, cMKO $n = 8$) with Carboxyatractyloside (CATR). Data represent mean ± SD; Multiple t-test, *$p < 0.05$, ***$p < 0.001$. **k** Oxygen consumption rates measured by high-resolution respirometry (left; JO$_2$) and mitochondrial membrane potential (right; ΔΨ) measured by quenching of Rhodamine 123 (RH123) fluorescence in cardiac mitochondria of WT and cMKO female mice between 8-10 week of age. JO$_2$ and ΔΨ were measured by addition of malate and palmitoyl-carnitine (PC, state 2) after the pretreatment (WT $n = 4$, cMKO $n = 4$) or not (WT $n = 6$, cMKO $n = 6$) of mitochondria with Carboxyatractyloside (CATR). Data represent mean ± SD; Multiple t-test, **$p < 0.01$.

accompanied by impaired RH123 quenching in state 2 and state 4 (Fig. 2e, S2f, h), indicating defective IMM substrate-dependent hyperpolarization. Supporting the notion that MTFP1 ablation increases proton leak, we consistently observed reduced respiratory control ratios when complex II was energized (Fig. S2e, middle-right) and lower mitochondrial membrane potential under both state 2 and state 4 conditions regardless of the respiratory substrates that were provided (Fig. 2e, S2f, h).

The elevated JO$_2$ rates and reduced RH123 quenching in state 4 could be explained either by a reduced sensitivity of the ATP synthase to Omy treatment or by uncoupling caused by proton leak across the IMM. In the mouse heart, reduced Omy sensitivity can result from defects in the assembly of the ATP synthase[47] that alter the affinity of Omy binding to Complex V between two adjacent c-subunits in contact with the proton half-channel formed by subunit a[48]. However, BN-PAGE analyses of cardiac mitochondria isolated from pre-symptomatic cMKO mice revealed no defects in ATP synthase assembly/maintenance (Fig. 2f, S2c (left)), leaving us with increased proton leak as the most parsimonious explanation for the observed state 4 respiration and membrane potential differences (Figs. 2d, e; S2e–h).

Next, we sought to corroborate our findings in cultured cells. To this end, we generated MTFP1-deficient mouse embryonic fibroblasts (MEFs) (Mtfp1$^{-/-}$) (Fig. S2i) and MTFP1-deficient human U2OS cells by Crispr/Cas9 genome editing (MTFP1$^{KO}$) and corresponding WT (MTFP1$^{WT}$) controls (Fig. S2j) and then assessed oxygen consumption by Seahorse FluxAnalyzer. Intriguingly, we observed no changes in basal or maximal respiration rates nor any evidence of mitochondrial uncoupling (Fig. S2k–s) suggesting that MTFP1-dependent proton leak may be cell type specific. Taken together, our data reveal an unappreciated and critical role of MTFP1 in bioenergetic efficiency and

mitochondrial uncoupling, particularly evident in metabolically active cardiomyocytes, which precedes the manifestation of cardiac dysfunction and heart failure in cMKO mice.

## Mitochondrial uncoupling is mediated by the adenine nucleotide translocase

To uncover the mechanism responsible for the mild mitochondrial uncoupling caused by Mtfp1 deletion in cardiomyocytes, we turned our attention to known uncoupling proteins. Uncoupling proteins (UCPs, UCP 1/2/3) and adenine nucleotide translocase (ANT) IMM proteins have been reported to be the two main catalysts of futile proton leak in mammalian mitochondria[49,50]. UCP1 is a bona fide uncoupler that is primarily expressed in brown adipose tissue[51] and shares significant sequence similarity with UCP2 and 3, which are expressed in other tissues[52]. ANT is an integral IMM transporter that catalyzes ADP uptake and ATP release in energized mitochondria[53]. ANT exists in four different tissue specific isoforms (ANT1, 2, 3, and 4), with ANT1 being the most abundant protein in mitochondria[54–56]. ANT1 has long been known for its namesake role as a nucleotide translocator, and recent studies have proven it to be an essential transporter of protons across the IMM in mammals[57] and a rate-liming factor for proton leak in Drosophila[54]. While the steady state levels of ANT in cardiac tissue of WT and cMKO mice were unaltered according to immunoblot assays (Fig. 2g) and cardiac proteomics (Fig. S2a (right), Supplementary Data 1), we nevertheless sought to functionally assess the contribution of ANT to MTFP1-dependent proton leak in freshly isolated cardiac mitochondria by using the ANT antagonist carboxyatractyloside (CATR), which binds irreversibly to ANT on the IMS side of the IMM, blocking its activity[58]. The addition of CATR after Omy treatment rescued the reduced ΔΨ and elevated

$JO_2$ rates back to WT levels (Fig. 2d(left)−2e). Independently of Omy, CATR treatment rescued proton leak when added after fueling state 3 respiration with ADP (Fig. 2h), normalizing the respiratory control ratio for state 3:4 (Fig. 2i). Moreover, the addition of CATR before energizing mitochondria with PGM (Fig. 2j) or malate/PC (Fig. 2k) was also able to normalize the elevated $JO_2$ and the decreased membrane potential of cMKO mitochondria to WT levels. Taken together, our data strongly indicate that *Mtfp1* deletion increases ANT-dependent proton leak.

To exclude the unlikely possibility that other uncoupling proteins such as UCPs might contribute to increased proton leak caused by *Mtfp1* ablation, we measured $\Delta\Psi$ of cardiac mitochondria in the presence of GTP, a pyrimidine nucleotide previously demonstrated to potently inhibit uncoupling in vitro[52,59]. UCP inhibition with GTP was not able to rescue the defective membrane potential observed in Omy-treated cMKO mitochondria (Fig. S2s) suggesting that UCPs do not contribute to the futile proton leak in cMKO cardiac mitochondria. Taken together, these data demonstrate that depletion of MTFP1 in the IMM leads to an increased uncoupling activity of ANT, resulting in proton leak and bioenergetic inefficiency preceding the development of cardiac dysfunction.

### MTFP1 is dispensable for mitochondrial fission

Transient MTFP1 knock-down has previously been reported to promote mitochondrial elongation in a variety of cultured cell types including neonatal cardiomyocytes[33,34,60,61], yet the consequence in the adult heart has never been explored. Therefore, to determine the impact of *Mtfp1* deletion on mitochondrial morphology, we co-labeled primary WT and cMKO primary adult cardiomyocytes (CMs) with tetramethylrhodamine ethyl ester (TMRE) and Mitotracker Deep Red (MTDR) to visualize mitochondria. Contrary to previous reports purporting that *Mtfp1* depletion causes mitochondrial elongation[35,62], we failed to observe any obvious effects on the morphology, distribution, or content of mitochondria in primary adult CMs deleted of *Mtfp1* (Fig. 3a, b). Similarly, transmission electron microscopy (TEM) analyses of pre-symptomatic cMKO and WT hearts (LV) showed no indications of mitochondrial elongation (median mitochondrial surface area: WT 3198 $\mu m^2$ versus cMKO 2954 $\mu m^2$) nor altered cristae organization (Fig. 3c, d). We also did not observe changes in mtDNA content (Fig. 3e) or mitochondrial shaping proteins in cardiac biopsies: the steady-state levels of fusion (MFN1, MFN2, OPA1) and fission (DRP1, MID51, FIS1) proteins were no different between WT and cMKO mice according to immunoblot (Fig. 3f) and cardiac proteomic analyses (Fig. S2a, Supplementary Data 1). Thus, MTFP1 is dispensable for mitochondrial dynamics in the heart.

In light of these surprising findings, we decided to measure mitochondrial morphology in MEFs depleted of MTFP1 under basal and stress conditions using a recent supervised machine learning (ML) approach we developed for high-throughput image acquisition and analyses[28]. In contrast to previous reports, MEFs depleted (siRNA *Mtfp1*) or deleted (*Mtfp1*$^{-/-}$) of *Mtfp1* showed only modest elongation of the mitochondrial network: both transient (siRNA) or chronic (knockout) ablation of MTFP1 resulted in ~15% increase in hypertubular mitochondria (Fig. 3i, j), and *Mtfp1*$^{-/-}$ MEFs showed unaltered steady-state levels of mitochondrial fusion and fission proteins (Fig. S3a). Contrary to DRP1-deficient cells[28], *Mtfp1*$^{-/-}$ MEFs were not protected from mitochondrial fragmentation induced by established pharmacological triggers of mitochondrial fragmentation, such as oligomycin (Omy), Rotenone, hydrogen peroxide ($H_2O_2$), or carbonyl cyanide m-chlorophenylhydrazone (CCCP) (Fig. 3g, h). Similarly, *Mtfp1*$^{-/-}$ MEFs were not protected from genetic induction of mitochondrial fragmentation by depletion of *Yme1l* or *Opa1* (Fig. 3i, j). While we could confirm that MTFP1 overexpression is able to promote mitochondrial fragmentation in MEFs, without affecting steady-state level of fusion and fission proteins (Fig. S3b,c). Our data collectively indicate that

MTFP1, unlike DRP1, is not an essential fission protein, contrary to its namesake, either in vitro or in vivo.

### MTFP1 deletion promotes mitochondrial permeability transition pore opening and programmed cell death

Cardiomyocyte death is catastrophic for adult cardiac function because of the limited regenerative capacity of these post-mitotic cells. Given the appearance of cardiac cell damage and death in cMKO mice during DCM (Fig. 1n, o) and the concomitant mitochondrial swelling and cristae disorganization in cardiac tissue of symptomatic cMKO mice (Fig. S3d, e), we sought to investigate whether MTFP1 loss specifically increased cell death sensitivity. To this end, we isolated adult primary cardiac myocytes (CMs) from WT and pre-symptomatic cMKO mice between 8–10 weeks of age and kinetically measured cell survival in response to a variety of cell death triggers using supervised ML-assisted high-throughput live-cell imaging[28]. We were able to isolate equally viable CMs from both WT and cMKO mice, yet upon dissipation of the membrane potential with CCCP (Fig. 4a, b) or treatment with $H_2O_2$ (Fig. 4c, d), *Mtfp1*$^{-/-}$ CMs succumbed to cell death more rapidly than WT CMs. Moreover, the induction of cell death with doxorubicin (DOXO, Fig. 4e, f), a cardiotoxic chemotherapeutic agent that triggers programmed cell death (PCD)[63], induced a significant increase of death in *Mtfp1*$^{-/-}$ compared to WT CMs, indicating that MTFP1 is essential for cell survival.

Prolonged opening of mitochondrial permeability transition pore (mPTP) causes mitochondria swelling, membrane potential dissipation and bioenergetic collapse, becoming a determinant of cell death[13]. To test whether MTFP1 loss causes increased susceptibility to mPTP opening, we assessed mitochondria swelling by exposing cardiac mitochondria of pre-symptomatic WT and cMKO mice to a high concentration of $Ca^{2+}$ and kinetically measured the light scattering[64]. $Ca^{2+}$ overload induced an increased mPTP dependent swelling of MTFP1-deficient cardiac mitochondria, which could be inhibited by the mPTP inhibitor cyclosporin A (CsA), indicating that *Mtfp1* deletion sensitizes cardiac mitochondria to mPTP opening (Fig. 4g, S4a). These findings were corroborated in *Mtfp1*$^{-/-}$ MEFs, in which we observed increased mPTP sensitivity that could be suppressed by CsA treatment (Fig. S4b) or by knocking out Cyclophilin D (CYPD, encoded by *Ppif*), the pharmacological target of CsA in mitochondria (Fig. 4h)[65]. Thus, these data clearly indicate that loss of MTFP1 at the level of the IMM sensitizes mitochondria to mPTP opening.

To define the molecular mechanisms underlying the increased sensitivity to PCD and mPTP opening caused by MTFP1 ablation, we used WT and *Mtfp1*$^{-/-}$ MEFs. We began by confirming that like CMs, MEFs deleted of *Mtfp1* were more sensitive to PCD. *Mtfp1*$^{-/-}$ MEFs showed normal growth rates (Fig. S4c) yet increased sensitivity to multiple cell death stimuli, as evidenced by more rapid kinetics of caspase 3/7 activation and cell death monitored by ML-assisted live-cell imaging of CellEvent (CE) and Propodium Iodide (PI) uptake, respectively[28]. Treatment with cell death triggers actinomycin D (ActD) and ABT-737 (Fig. 4i–k, S4d–f), staurosporine (STS, Fig. S4g–i) or DOXO (Fig. 4l–n) all promoted a more rapid and robust cell death response in *Mtfp1*$^{-/-}$ MEFs relative to WT cells, which could be blocked with the pan-caspase inhibitor Q-VD-OPh (qVD). These effects were independent of mitochondrial respiration, which was unaltered in *Mtfp1*$^{-/-}$ MEFs (Fig. S2k–n). Taken together, our data clearly demonstrate a protective role of MTFP1 in maintaining cell integrity and survival.

### Doxorubicin induced-cardiotoxicity accelerates the onset of cardiomyopathy in cMKO mice

To test whether MTFP1 protected against PCD induction in vivo, we injected pre-symptomatic (aged 8 weeks) cMKO and WT mice with DOXO, which is known to promote onset of the permeability transition[66] and assessed cardiac function at 14 days post treatment

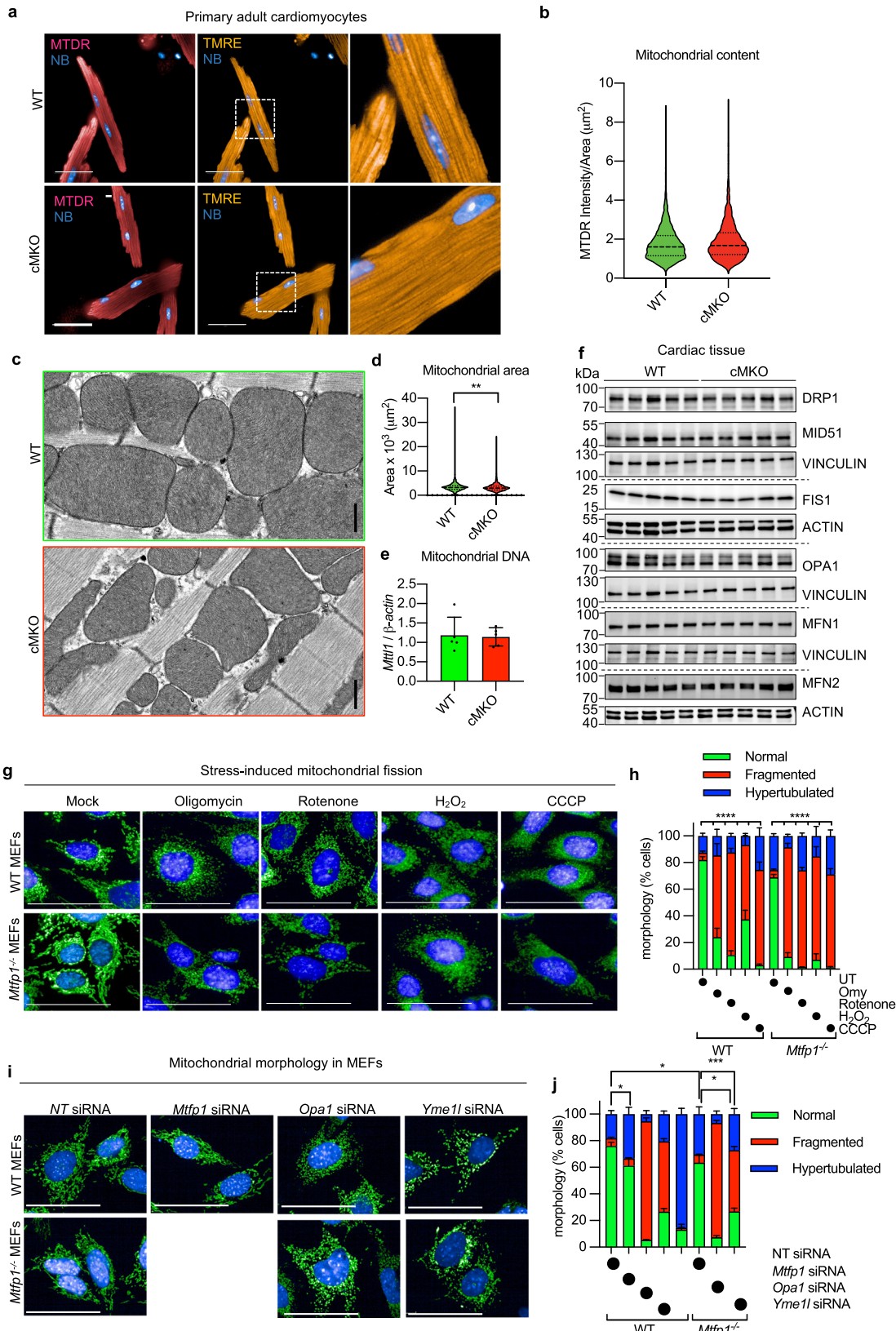

by echo (Fig. S4j–n). Consistent with the data obtained in vitro, we observed that DOXO accelerated the onset of cardiac dysfunction in cMKO mice by lowering LVEF (WT $60.85 \pm 6.2\%$ vs. cMKO $47.02 \pm 9.1\%$), PW thickness during systole (WT $1.082 \pm 0.099$ mm vs. cMKO $0.887 \pm 0.062$ mm), while increasing systolic LV diameter

(WT $2.393 \pm 0.27$ mm vs. cMKO $3.009 \pm 0.36$ mm) and diastolic LV diameter (WT $3.505 \pm 0.20$ mm vs. cMKO $3.918 \pm 0.25$ mm). These results clearly indicate that cMKO mice are more susceptible to DOXO induced cardiotoxicity, accelerating the onset of cardiomyopathy.

**Fig. 3 | MTFP1 is dispensable for mitochondrial fission. a** Representative images of primary adult cardiomyocytes isolated from WT and cMKO female mice at 8 weeks and were labeled with MitoTracker Deep Red (MTDR) and tetra-methylrhodamine, ethyl ester (TMRE), and NucBlue (NB). Scale bar = 50 μm. **b** Violin plot of mitochondrial content (MTDR Intensity/Area) of WT ($n = 6085$) and cMKO CMs ($n = 3647$) measured in **a**. **c** Representative transmission electron micrographs of cardiac posterior walls of WT (top, $n = 3$) and cMKO (bottom, $n = 3$) mice at 8–10 weeks. Scale bar: 500 nm. **d** Violin plot of mitochondrial surface area (μm²) within cardiac posterior wall measured in c (WT mitochondria $n = 659$; cMKO mitochondria $n = 966$). Dotted line represents quartiles and dashed line represents median; **$p < 0.01$ Mann-Whitney test. **e** Mitochondrial DNA (mtDNA) content in WT ($n = 5$) and cMKO ($n = 5$) heart tissue of male mice quantified by amplification of the mitochondrial *Mttl1* gene relative to nuclear gene *b-Actin*. Data represent mean ± SD. **f** Immunoblot of mitochondrial fission and fusion proteins measured in cardiac WT and cMKO (8–10 week) extracts immunoblotted with the indicated antibodies (horizontal line denotes different membranes) performed thrice with similar results. VINCULIN or ACTIN are used as loading controls. **g** Representative images of WT and *Mtfp1⁻/⁻* MEFs treated with the fission-inducing drugs: oligomycin (Omy), Rotenone, $H_2O_2$ and carbonyl cyanide m-chlorophenyl hydrazine (CCCP). Mitochondria stained with MitoTracker Deep Red (MTDR, green) and nuclei with NucBlue (NB, blue). Scale bar = 100 μm. **h** Quantification of mitochondrial morphology in **g** by supervised ML using WT cells with normal (UT), fragmented (CCCP-treated) or hypertubular (cycloheximide-treated) mitochondria as ground truths. Data are means ± SEM of 7–16 independent replicates. 2way-ANOVA, Dunnet's multiple comparison test: % fragmentation ****$p < 0.0001$ treatment versus WT UT or *Mtfp1⁻/⁻* UT. **i** Representative confocal images of WT and *Mtfp1⁻/⁻* MEFs treated with indicated siRNAs (20 nM) for 72 h and labeled with MitoTracker Deep Red (MTDR, green) and NucBlue (NB, blue). Scale bar = 100 μm. **j** Quantification of mitochondrial morphology in **i** by supervised machine learning (ML) using WT cells with normal (non-targeting NT siRNA), fragmented (*Opa1* siRNA) or hypertubular (*Dnm1l* siRNA) mitochondria as ground truths. Data are means ± SEM of 2–8 individually plated wells measured in parallel. 2way-ANOVA, Dunnet's multiple comparison test: % hypertubular *$p < 0.05$; ***$p < 0.001$ versus WT NT siRNA; % fragmented ****$p < 0.0001$ versus *Mtfp1⁻/⁻* NT siRNA.

## Inhibition of mPTP rescues cell death sensitivity of MTFP1 deficient cells

It has been previously reported that DOXO mediates mPTP opening and cell death in lung cancer cells[67] and cardiac myocytes, and that $H_2O_2$ activates necrosis through the induction of mPTP opening[68]. We observed that $H_2O_2$-induced cell death was accelerated in *Mtfp1⁻/⁻* cells and was reduced, yet not totally abolished by caspase inhibition with qVD, indicating that MTFP1 loss also renders cells more susceptible to caspase-independent cell death (Fig. 5a–c). Since MTFP1-deficient cardiac and MEF mitochondria were more susceptible to mPTP opening (Figs. 4g, h, S4a, b), we investigated whether prolonged mPTP opening contributes to increased cell death sensitivity of MTFP1-deficient cells. To test the dependence of PCD on the mPTP, we disrupted CYPD in WT and *Mtfp1⁻/⁻* MEFs by introducing a truncating, homozygous frame shift mutation (p.Val65*) by Crispr/Cas9 genome editing (Fig. 5d) and subjected cells to $H_2O_2$ (Fig. 5e–g) or DOXO treatment (Fig. 5h–j). By tracking the kinetics of PI or CE uptake respectively, we observed that CYPD ablation in *Mtfp1⁻/⁻* cells (*Mtfp1⁻/⁻Ppif1⁻/⁻* MEFs) rescued the cell death sensitivity back to WT levels (Fig. 5e–j). Consistent with a cytoprotective effect of CYPD ablation, we observed that the association of CsA to qVD treatment had a synergic effect in suppressing cell death sensitivity of *Mtfp1⁻/⁻* MEFs to WT levels (Fig. 5a–c).

These findings were also corroborated in HL-1 cardiomyocytes depleted of *Mtpf1* and *Ppif* (Fig. S5a–d) and were independent of mitochondrial membrane potential (Fig. S5e–f) and respiration (Fig. S5g–j). Taken together, these results clearly indicate that loss of MTFP1 promotes mPTP opening via CYPD to lower the resistance to programmed cell death.

To gain insights into the molecular regulation of the mPTP by MTFP1 we sought to define the cardiac interactome of MTFP1. We expressed FLAG-MTFP1 at the *Rosa26* locus in C57Bl6/N mouse hearts via targeted transgenesis (Fig. 6a, S6a), specifically activating the expression of FLAG-MTFP1 in cardiomyocytes using Myh6-Cre recombinase (Cardiomyocyte^FLAG-MTFP1). Cardiomyocyte^FLAG-MTFP1 mice were outwardly normal and echocardiography studies revealed no impact of the modest level of FLAG-MTFP1 over-expression on cardiac function in vivo (Fig. 6a, S6b). Next, to analyze the cardiac interactome of MTFP1 we performed a co-immunoprecipitation study coupled to mass spectrometry (MS) analyses of cardiac mitochondria isolated from cardiomyocyte^FLAG-MTFP1 mice. We identified 60 mitochondrial proteins besides the bait protein (MTFP1) that were exclusively present in FLAG-MTFP1 eluates or significantly enriched greater than two-fold (Fig. 6b, Supplementary Data 3). Among these interactors we found factors involved in OXPHOS function (Fig. 6b, c), notably proteins required for the assembly and functions of Complex I (NDUFA10, NDUFA7, NDUFS6, NDUFB4, NDUFS6, MTND5), Complex IV (CMC2, COX4I1, COX6B1, COX7A1, SCO2), and Complex V (ATP5L, USMG5). In addition, we identified a number of proteins that have previously been implicated in mPTP regulation including the ADP/ATP translocase ANT1 (also termed SLC25A4)[55], the inorganic phosphate carrier SLC25A3[69], and the heat shock protein TRAP1[70] (Fig. 6b, c). To determine whether MTFP1 forms a complex with physical interactors identified in the cardiac proteome, we performed 2D BN-PAGE analysis on cardiac mitochondria isolated from pre-symptomatic cMKO mice. We observed that MTFP1, which has a monomeric molecular weight of 18 kDa, forms an oligomeric, high molecular weight complex of approximately 60-250 kDa that co-migrates with ANT1 (SLC25A4) and CYPD (PPIF) (Fig. 6d, top). In cMKO mitochondria, the migration pattern of ANT1 and CYPD is unaffected, implying that MTFP1 may affect mPTP activity without altering the maintenance or assembly of this complex (Fig. 6d, bottom). Altogether, our data have uncovered a functional and physical link between MTFP1 and the mPTP complex in the inner mitochondrial membrane, which is of paramount importance for cardiac health (Fig. 7).

## Discussion

We initially chose to focus studies on Mitochondrial Fission Process 1 (MTFP1) because we viewed this protein as a promising entry point to study the hitherto molecularly undefined process of inner membrane division[71], based largely on previous studies that had purported a pro-fission role upon over-expression and an anti-fission role upon depletion[33–36,72]. While we demonstrate MTFP1 to be a bona fide inner membrane protein in vivo (Fig. S1b, c), confirming previous in vitro studies[31,32], careful, unbiased mitochondrial morphology analyses unequivocally excluded this protein as an essential fission factor (Fig. 3g–j) for the following reasons: (1) acute or chronic depletion of MTFP1 in vitro (Fig. 3g–j) or deletion in vivo (Fig. 3c, d) had little impact on the elongation of the mitochondrial network and (2) MTFP1-deficient cells were not protected against mitochondrial fragmentation caused either by accelerated fission or impaired fusion induced either by genetic or pharmacological triggers (Fig. 3g–j). Consistent with these findings, we did not identify MTFP1 as a regulator of mitochondrial morphology in a recent, comprehensive siRNA-based phenotypic imaging screens of all mitochondrial genes performed in human fibroblasts[28], which did, however, identify essential fission proteins like DRP1 and receptors[73–75], forcing us to reconsider the existing models of mitochondrial fission we[20,71] and others had proposed[19]. In contrast, other studies have reported that chemical inhibitors of mTOR[72], PI3K[32], or miRNAs[33,34] can deplete MTFP1 protein levels and thus inhibit mitochondrial fragmentation, although the pleiotropic effects of these molecules and the signaling cascades they regulate make it the interpretation of these data very challenging. However, our in vitro studies confirmed that over-expression of MTFP1

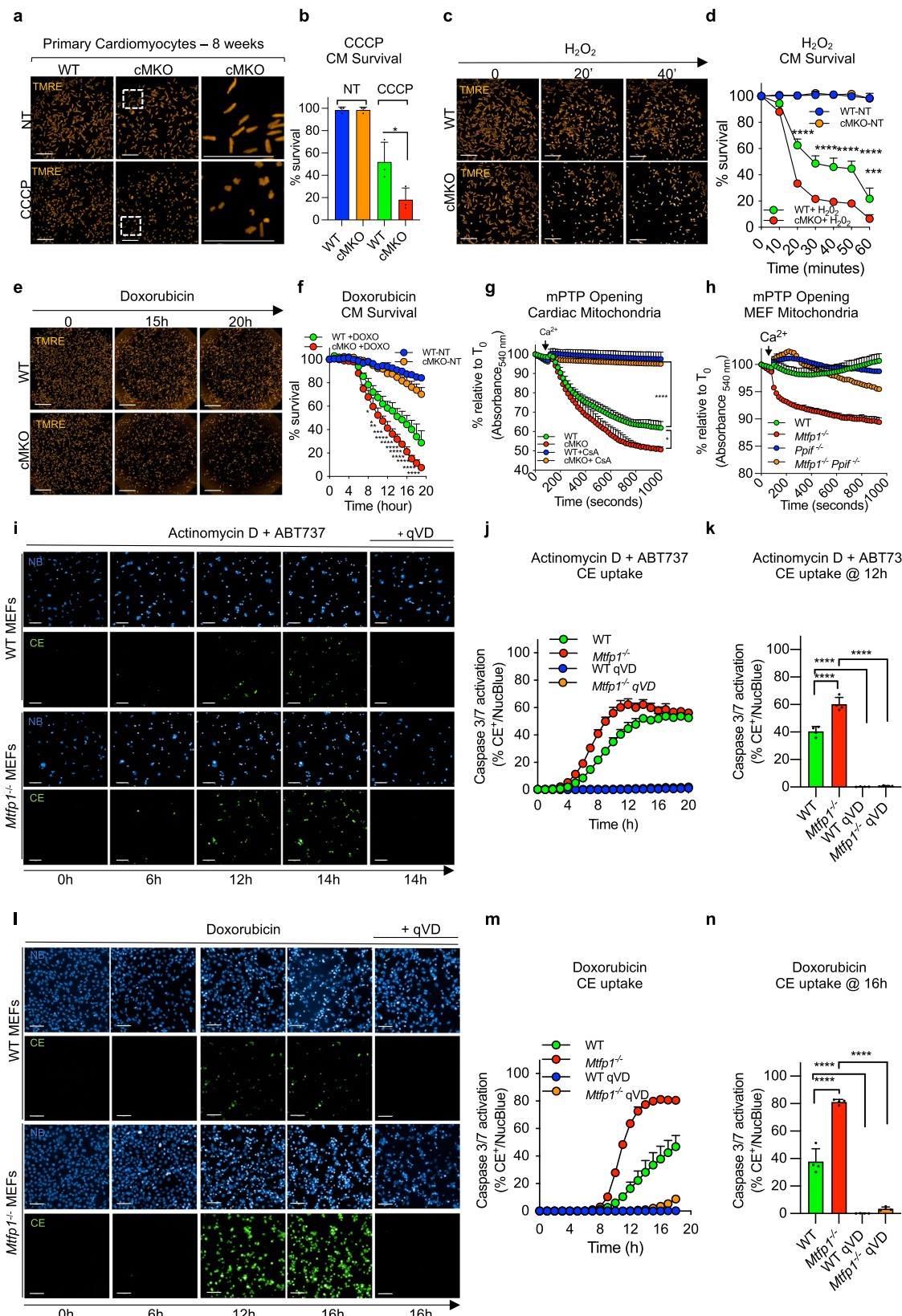

does indeed promote mitochondrial fragmentation, which appears to be independent of stress-induced OPA1 processing and alterations in the levels of the fission and fusion executors (Fig. S3b-c). In vivo, stable over-expression in cardiomyocytes does not appear to impact basal cardiac function assessed by echocardiography (Fig. S6b), although future studies will be required to determine the influence

of MTFP1 overexpression on cardiac physiology under stress conditions.

If MTFP1 is dispensable for fission, what role, if any, does this metazoan-specific factor play in mitochondria? Our in vivo studies revealed that cardiomyocyte-specific deletion of MTFP1 (cMKO) drives the progressive development of dilated cardiomyopathy (DCM)

**Fig. 4 | MTFP1 protects against mitochondrial PTP opening and cell death.**
**a** Representative confocal images of adult cardiomyocytes (CMs) isolated from WT and cMKO female mice at 8 weeks stained with tetramethylrhodamine ethyl ester (TMRE) treated with or without cyanide m-chlorophenyl hydrazine treatment (CCCP) for 15 min. Rod-shaped CMs: live cells, round-shaped CMs: dead cells. Scale bar: 500 µm. **b** Quantification of number of live cells (% survival) in **a** by supervised machine learning. Data are means ± SD of $n = 3$ independent experiments. Unpaired Student's $t$-test; *$p < 0.05$. **c** Representative confocal images of adult cardiomyocytes (CMs) isolated from WT and cMKO female mice at 8 weeks stained with tetramethylrhodamine ethyl ester (TMRE) and subjected to $H_2O_2$ treatment for 1 h. Rod-shaped CMs: live cells, round-shaped CMs: dead cells. Scale bar: 500 µm. **d** Quantification of number of live cells (% survival) over time measured in **c** by supervised machine learning. Data are means ± SD of 2–3 culture replicates and representative of $n = 3$ experiments. 2wayANOVA, Tukey's multiple comparison test, ***$p < 0.001$, ****$p < 0.0001$ vs WT $H_2O_2$. **e** Representative confocal images of adult cardiomyocytes (CMs) isolated from WT and cMKO female mice at 8 weeks stained with tetramethylrhodamine ethyl ester (TMRE) and subjected to Doxorubicin (DOXO) treatment. Scale bar: 500 µm. **f** Quantification of number of live cells (% survival) in **e** by supervised machine learning. Data are means ± SD of 2–3 culture replicates and representative of $n = 3$ experiments. 2wayANOVA, Tukey's multiple comparison test, *$p < 0.05$, **$p < 0.01$, ***$p < 0.001$, ****$p < 0.0001$ vs WT DOXO. Scale bar: 1 mm. **g** Mitochondrial swelling assay performed on cardiac mitochondria extracted from hearts of WT ($n = 3$) and cMKO ($n = 3$) female mice at 8–10 weeks. Relative absorbance at 540 nm was measured every 20 s before and after addition of a single pulse of $CaCl_2$ (arrowhead) in presence or absence of

Cyclosporin A (CsA). Data are means ± SD of $n = 3$ biological replicates. One-way ANOVA of maximal absorbance 540 nm (% relative to $T_0$) change, *$p < 0.05$, ****$p < 0.0001$. **h** Mitochondrial swelling assay performed on mitochondria isolated from WT and $Mtfp1^{-/-}$, $Ppif^{-/-}$, and $Mtfp1^{-/-}Ppif^{-/-}$ MEFs. Mitochondrial absorbance changes (absorbance 540 nm, % relative to $T_0$) are measured every 30 s prior and after addition of a single pulse of $CaCl_2$ (arrowhead) in presence or absence of Cyclosporin A (CsA). Data are means ± SD of $n = 3$ (WT, $Mtfp1^{-/-}$) and $n = 2$ (WT, $Mtfp1^{-/+}$ CsA) technical replicates. **i** Representative confocal images of WT and $Mtfp1^{-/-}$ MEFs subjected to actinomycin D (ActD) plus ABT-737 treatment in the presence or absence of the pan-caspase inhibitor q-VD-OPh hydrate (qVD). Live induction of the caspase 3/7 activation was monitored by using the CellEvent (CE, green) reagent and imaging cells every hour (h) for 20 h. Scale bar = 100 µm. **j** Kinetics of caspase 3/7 activation was determined by counting the number of $CE^+$ positive cells (green) over total number cells nuclear stained with NucBlue (NB, blue) and expressed as % $CE^+$/NucBlue. Data are means ± SD of $n = 4$ independent experiments. **k** one-way ANOVA of **j** at 12 h, ****$p < 0.0001$. **l** Representative confocal images of WT and $Mtfp1^{-/-}$ MEFs subjected to doxorubicin treatment in the presence or absence of the pan-caspase inhibitor q-VD-OPh hydrate (qVD). Live induction of the caspase 3/7 activation was monitored by using the CellEvent (CE, green) reagent and imaging cells every hour (h) for 18 h. Scale bar = 100 µm. **m** Kinetics of caspase 3/7 activation was determined by counting the number of $CE^+$ positive cells (green) over total number cells nuclear stained with NucBlue (NB, blue) and expressed as % $CE^+$/NucBlue. Data are means ± SD and representative of at least $n = 3$ independent experiments. **n** one-way ANOVA of **m** at 16 h, ****$p < 0.0001$.

beginning at 18 weeks of age culminating in chronic heart failure (Fig. 1g–l, S1f–k) and middle-aged death in both male and female mice (Fig. 1f, S1e). Cardiac dysfunction was accompanied by a reduction in mitochondrial gene expression and respiratory chain function, fibrotic remodeling (Fig. 1p–r) and a general dysregulation of metabolic genes (Fig. 1q), which are features that have been observed in other cardiomyocyte-specific knockout mouse models of mitochondrial genes[8,24,42,76–78]. Similarly, accumulation of pathogenic mutations in mitochondrial DNA have recently been shown to drive mitochondrial dysfunction and sterile inflammation in a number of different tissues including the heart and genetic inhibition of the latter phenotype appears to resolve tissue dysfunction, supporting a pathological role of cardiac inflammation triggered by mitochondrial dysfunction[79]. In cMKO mice, the aforementioned phenotypes were absent before the onset of DCM, indicating that MTFP1 deletion in perinatal cardiomyocytes does not compromise post-natal cardiac development in mice. Thus, we hypothesize that the metabolic and inflammatory remodeling that accompanies DCM manifests as downstream response to cardiac dysfunction, which we are currently testing. Surprisingly, functional characterization of field-stimulated primary adult cardiomyocytes from pre-symptomatic mice did not reveal defects in contractile capacity, excitation contraction (EC) coupling, or sarcomere integrity (Fig. S1n–s), prompting us to search for other homeostatic dysfunctions of cardiomyocytes that could account for the contractile defects of the beating heart caused by MTFP1 deletion. Indeed, characterization of primary cardiomyocytes and cardiac mitochondria from cMKO mice revealed an increased sensitivity to programmed cell death (Fig. 4a–f) and increased sensitivity to opening of the mitochondrial permeability transition pore (mPTP) (Fig. 4g, S4a), respectively[13–15]. Accelerated opening of the mPTP has been shown to control cardiomyocyte viability and cardiac function in genetic[16], infectious[17], and surgically-induced mouse models of cardiomyopathy[15]. The identity of the mPTP has been hotly debated[13,80,81] and a consensus has yet to be achieved regarding its structure and molecular constitution, although compelling evidence from knockout mice and in vitro reconstitution experiments have identified a number of mitochondrial proteins required for the efficient opening of the mPTP including subunits of the ATP synthase, Cyclophilin D (CYPD, encoded by the nuclear gene *Ppif*), and IMM carrier proteins of the SLC25 family[13–15,55,64,69,82–84]. We identified several

of these proteins to physically interact with MTFP1 by coimmunoprecipitation studies performed on cardiac mitochondria isolated from cardiomyocyte$^{FLAG-MTFP1}$ mice, including SLC25A3 (Phosphate carrier), SLC25A4 (ANT1), TRAP1, ATP5L and USMG5 (Fig. 6b, c). We performed 2D-BN-PAGE analysis of mouse cardiac mitochondria that show MTFP1 forms a complex of approximately 60–250 kDA, which co-migrates with SLC25A4 and CYPD (Fig. 6d). These observations are consistent with complexome profiling studies of rat cardiac mitochondria, which also revealed co-migration of MTFP1 with several binding partners[85], including SLC25A3, SLC25A4, and CYPD. Ablation of MTFP1 does not significantly impact the size of complexes containing ANT1 and CYPD, leaving us with the most likely explanation that MTFP1 modulates mPTP activity not by regulating its gross assembly or maintenance but rather via altering substrate accessibility and/or its activation. In MTFP1-deficient cells, inhibiting mPTP opening by pharmacological inhibition with Cyclosporin A (CsA) or genetic deletion of CYPD rescues the sensitivity to mPTP opening and decreased cell death (Figs. 4h, 5a–j, S5a–d). Our discovery that the genetic deletion of *Mtfp1* sensitizes both post-mitotic cardiac cells and mitotic epithelial cells to PCD without affecting cardiomyocyte differentiation nor cell proliferation argues for a general role for MTFP1 in cell survival, which is supported by studies in other cell lines[33,72]. Why other groups have reported that MTFP1 depletion can protect gastric cancer and cardiomyocyte cell lines from PCD induced by various cell death triggers including doxorubicin is unclear[34–36]. Although our in vivo data demonstrate that MTFP1 deletion in cardiomyocytes accelerates, rather than retards, the cardiotoxic effects of doxorubicin (Fig. S4j–n), which is consistent with the PCD sensitivity we measured in MEFs and primary cardiomyocytes, we cannot exclude that different cell lines and tissue may respond differently to the loss of MTFP1.

Indeed, MTFP1 ablation significantly impacts mitochondrial respiration in the adult heart but not in highly glycolytic epithelial cells such as MEFs (Fig. S2k–n) and human osteosarcoma cells (U2OS) (Fig. S2o–r) or even immortalized HL-1 cardiomyocytes (Fig. S5g–j), highlighting the unique bioenergetic profile and demands of primary adult cardiomyocytes.

Efficiently coupled oxidative phosphorylation requires proton gradient formed across the IMM by the mitochondrial ETC, which is then harnessed by ATP synthase to generate ATP from ADP and inorganic phosphate, imported into the matrix by the ADP/ATP translocase

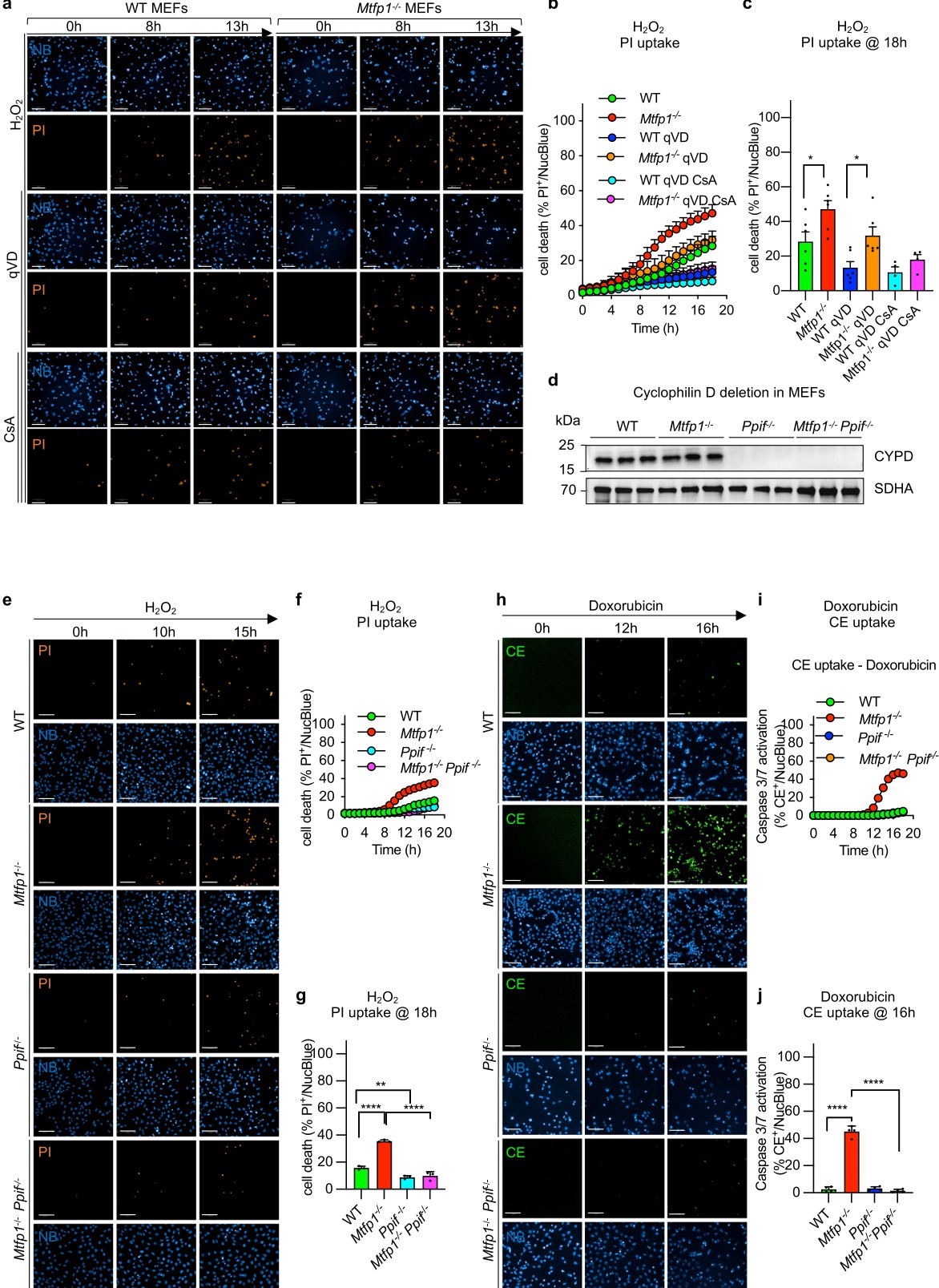

(ANT) and phosphate carrier (SLC25A3), respectively. Mitochondrial uncoupling occurs when proton motive force is dissipated by proton leak into the matrix and oxygen consumption is not coupled to ATP generation. Maintenance of constant cellular ATP concentration is critical for cell survival and the function, therefore, uncoupling of the ETC from ATP generation can have deleterious effects in

cardiac cells, whose constant energy supply is essential for the beating heart.

While the oxygen consumption rate (OCR) (Fig. S2k–r) and the maximal phosphorylating respiration (Figs. 2d, h, S2e, g), equivalent to state 3, were normal in cells and cardiac mitochondria isolated from cMKO mice, MTFP1 loss significantly reduced the respiratory control

**Fig. 5 | mPTP accounts for cell death sensitivity in MTFP1 deficient cells.**
**a** Representative confocal images of WT and $Mtfp1^{-/-}$ MEFs subjected to $H_2O_2$
treatment. The pan-caspase and cyclophilin D inhibitors, q-VD-OPh hydrate (qVD)
and cyclosporin A (CsA) respectively, were used to block both caspase and mPTP
dependent cell death. Cell death was monitored by Propidium Iodide uptake (PI,
orange) and imaging cells every hour (h) for 18 h. Scale bar = 100 µm. **b** Kinetics of
PI uptake was determined by counting the number of $PI^+$ positive cells (orange)
over total number cells nuclear stained with NucBlue (NB, blue) and expressed as %
$PI^+$/NucBlue. Data are means ± SD of $n = 6$ independent experiments. **c** one-way
ANOVA of **b** at 18 h, *$p < 0.05$. **d** Validation of Cyclophilin D (CYPD) ablation by
Crispr/Cas9 genome editing of $Ppif$ in WT and $Mtfp1^{-/-}$ MEFs. Equal amounts of
protein extracted from WT, $Mtfp1^{-/-}$, $Ppif^{-/-}$, $Mtfp1^{-/-} Ppif^{-/-}$ MEFs ($n = 3$) were sepa-
rated by SDS–PAGE and immunoblotted with the indicated antibodies. SDHA was
used as mitochondrial marker and loading control. **e** Representative confocal
images of WT, $Mtfp1^{-/-}$, $Ppif^{-/-}$, $Mtfp1^{-/-} Ppif^{-/-}$ MEFs subjected to $H_2O_2$ treatment.

Cell death was monitored by Propidium Iodide uptake (PI, orange) and imaging
cells every hour (h) for 18 h. Scale bar = 100 µm. **f** Kinetics of PI uptake was deter-
mined by counting the number of $PI^+$ positive cells (orange) over total number cells
nuclear stained with NucBlue (NB, blue) and expressed as % $PI^+$/NucBlue. Data are
means ± SD of $n = 3$ independent experiments. **g** one-way ANOVA of **f** at 18 h;
**$p < 0.01$, ****$p < 0.0001$. **h** Representative confocal images of WT, $Mtfp1^{-/-}$, $Ppif^{-/-}$,
$Mtfp1^{-/-} Ppif^{-/-}$ MEFs subjected to doxorubicin (DOXO) treatment. Live induction of
the caspase 3/7 activation was monitored by using the CellEvent (CE, green). Cel-
lEvent positive cells ($CE^+$, green) over total number cells NucBlue labeled (blue)
were imaged every hour (h) for 18 h. Scale bar = 100 µm. **i** Kinetics of caspase 3/7
activation was determined by counting the number of $CE^+$ positive cells (green)
over total number cells nuclear stained with NucBlue (NB, blue) and expressed as %
$CE^+$/NucBlue. Data are means ± SD and representative of $n = 3$ independent
experiments. **j** one-way ANOVA of **i** at 16 h, ****$p < 0.0001$.

ratios (RCR) in cardiac mitochondria energized with complex I (Fig. 2d)
or complex II substrates (Fig. S2e), suggesting a general mechanism of
uncoupling of the respiration from the ATP production as a result of
increased proton leak trough the IMM.

Proton leak has marked influence on energy metabolism.
Enhancement of this process in various tissues can counteract the
deleterious effects of nutrient overload via UCP1-dependent and
independent pathways[86] and thus may be beneficial in some settings.
In the heart, whose bioenergetic efficiency has evolved to maximize
ATP output, excessive proton leak has been shown to drive age-related
cardiomyocyte and cardiac dysfunction in mice. Studies performed by
the Rabinovitch lab clearly demonstrated that ANT-dependent proton
leak is increased in cardiomyocytes from old, but not young mice and
can be rejuvenated by blocking ANT and reducing sensitivity to mPTP
opening[87,88].

In line with the notion that increased proton leak is maladaptive
for the heart, our study shows that MTFP1 loss in cardiomyocytes
reduces the mitochondrial membrane potential as a result of increased
proton leak through the IMM (Fig. 2d, e, Fig. S2e–h) preceding the
onset of cardiomyopathy. We provide direct evidence that ANT is the
most likely site of proton leak in cardiac mitochondria, as its inhibition
with carboxyatractyloside (CATR) suppresses proton leak and re-
establishes normal membrane potential and respiratory control ratios
in MTFP1-deficient mitochondria (Fig. 2d, e, h–k). While we have clear
evidence of uncoupling via ANT on one hand and increased sensitivity
to mPTP opening and mitochondria swelling on the other, future
studies are still required to decipher whether these mechanisms are
interdependent and whether they must synergize to drive cardiac
decline and heart failure. A large number of genetic mouse models of
cardiomyopathy targeting mitochondrial genes have been generated
over the last 20 years, yet to the best of our knowledge cMKO mice
represent the first in which bioenergetic efficiency is compromised
without affecting maximal respiratory capacity (state 3), thus provid-
ing a novel model to study the relevance of cardiac mitochondrial
uncoupling and its progressive impact on cardiac homeostasis.

In summary our study reveals new and essential roles of MTFP1 in
cardiac homeostasis that are distinct from its previously reported
impact on mitochondrial fission (Fig. 1a), the latter of which our data
conclusively show is unaffected in vitro and in vivo. Thus, our work
now positions MTFP1 as a critical regulator of mitochondrial coupling
through ANT in cardiomyocytes and its loss leads to membrane
potential dissipation associated to mPTP opening, cell death and
progressive DCM that leads to heart failure, and middle-aged death in
mice (Fig. 7). These findings advance our understanding of the mito-
chondrial defects that can trigger the development of DCM and heart
failure. We propose that MTFP1 to be a valuable tool for the molecular
dissection of mitochondrial uncoupling and mPTP function and thus a
promising target to mitigate the pathological events of cardiac and
metabolic remodeling in heart disease.

## Methods
### Animals

Animals were handled according to the European legislation for animal
welfare (Directive 2010/63/EU). All animal experiments were per-
formed according to French legislation in compliance with the Eur-
opean Communities Council Directives (2010/63/UE, French Law 2013-
118, February 6, 2013) and those of Institut Pasteur Animal Care
Committees. The specific approved protocol numbers are
202005191046361 and 2018112017053431.

Mice were housed within a
specific pathogen-free facility and maintained under standard housing
conditions of a 14–10 h light-dark cycle, 50–70% humidity, 19–21 °C
with free access to food and water in cages enriched with bedding
material and gnawing sticks. To safeguard animal welfare, animals
were observed weekly, which allowed early identification of any clinical
sign, stress or pain. As for pain management, rigorous and regular
follow-up and use analgesics were standard of care. Since mice defi-
cient in MTFP1 exhibited progressive dilated cardiomyopathy, the limit
points corresponding to a score 2 according to the Institut Pasteur
Animal Care Committees evaluation grid of the Institut Pasteur justi-
fied euthanasia. These points corresponded to prostration, bent back
and very altered facial expression, major malnutrition, weight loss
greater than 15%, or very agitated or immobile animal, self-mutilation.
In some procedures, an ejection fraction of less than 20% (heart
ultrasound), led to euthanasia, via $CO_2$-dependent asphyxiation or
cervical dislocation. M$tfp1$ conditional mice ($Mtfp1^{LoxP/LoxP}$) were gen-
erated by PolyGene AG (Switzerland) on a C57Bl6/N background.
Cardiomyocyte specific $Mtfp1$ KO mice (cMKO; $Myh6\text{-}Cre^{tg/+}$
$Mtfp1^{LoxP/LoxP}$) were generated by crossing $Mtfp1$ conditional mice
($Mtfp1^{LoxP/LoxP}$) with transgenic (Tg) mice expressing the Cre recombi-
nase under the control of the cardiac alpha myosin heavy chain 6
promoter ($Myh6\text{-}Cre$)[37]. Littermates that were homozygous for the
conditional allele and negative for $Myh6\text{-}Cre$ were used as controls
(WT; $Myh6\text{-}Cre^{+/+}Mtfp1^{LoxP/LoxP}$). Heterozygous whole-body $Mtfp1$ KO
mice ($Mtfp1^{+/-}$) were generated by crossing $Mtfp1$ conditional mice
($Mtfp1^{LoxP/LoxP}$) with transgenic (Tg) mice expressing the Cre recombi-
nase under the control of the CMV promoter ($CMV\text{-}Cre$)[89] and back-
crossing to C57Bl6/N wild type (WT) mice.

Cardiomyocyte specific FLAG-MTFP1 Knock-In (KI) mice ($Myh6$-
$Cre^{Tg/+}Mtfp1^{+/+}$, $CAG^{Tg+/}$) were generated by crossing an inducible mouse
model for mCherry-P2A-FLAG-MTFP1 generated by PolyGene AG
(Switzerland) on a C57Bl6/N background expression under the CAG
promoter with mice expressing the Cre recombinase under the control
of the cardiac alpha-(α) myosin heavy chain 6 promoter ($Myh6\text{-}Cre$).

### Cell lines
WT ($Mtfp1^{+/+}$) and knockout (KO) $Mtfp1^{-/-}$ embryos were isolated at
E13.5 following F1 heterozygous intercrosses of $Mtfp1^{+/-}$ whole-body
KO mice. Immortalization of WT ($Mtfp1^{+/+}$) and $Mtfp1^{-/-}$ primary mouse
embryonic fibroblasts (MEFs) was performed as previously described[28]

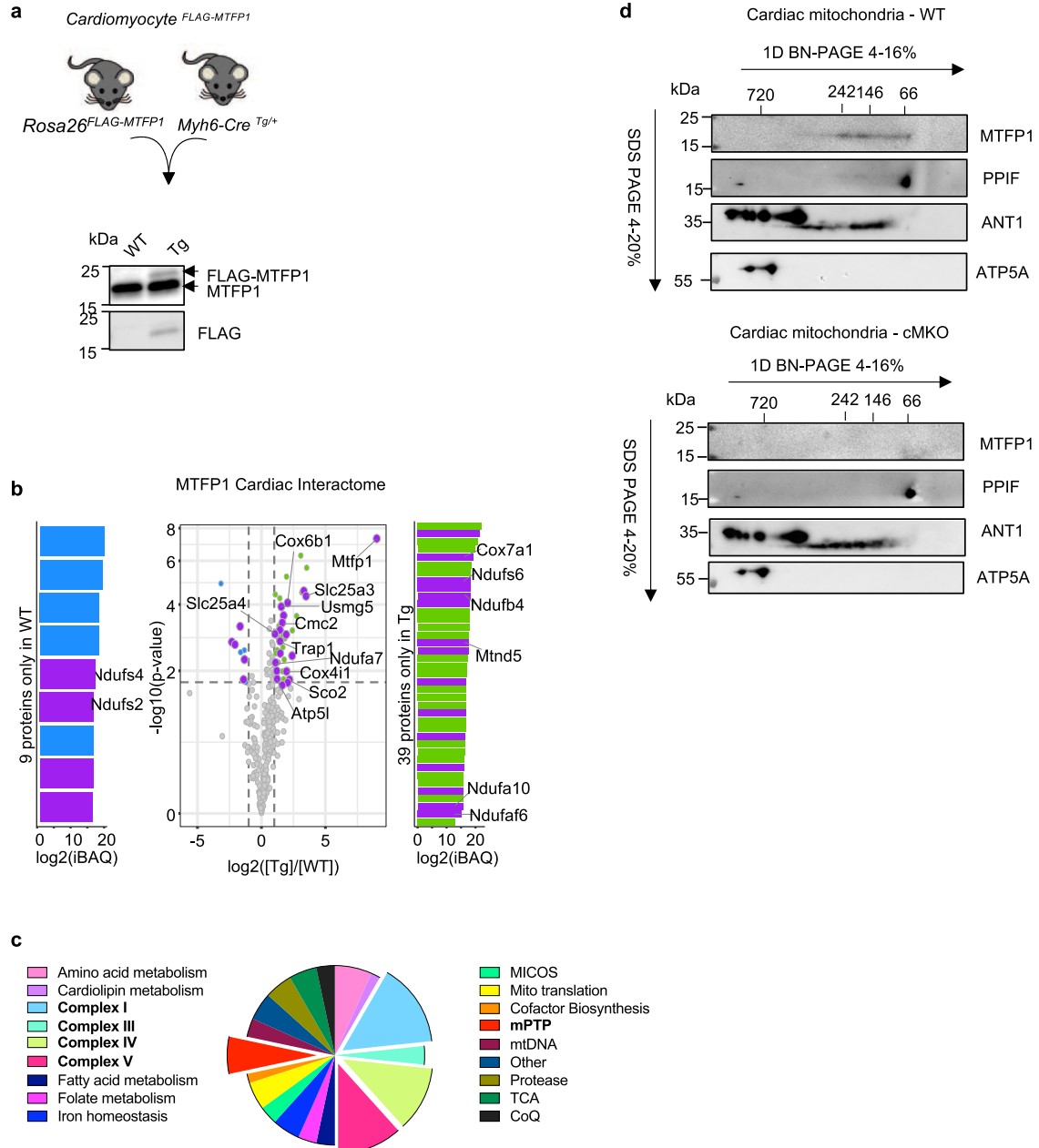

**Fig. 6 | MTFP1 interacts with components of the mPTP complex.**
**a** Representative immunoblot of the expression of FLAG-MTFP1 compared to endogenous MTFP1 levels in transgenic (Tg) Cardiomyocyte[FLAG-MTFP1] mice constitutively expressing FLAG-MTFP1 in cardiomyocytes. Similar results were obtained with $n = 4$ biological replicates. **b** Volcano plot of the FLAG-MTFP1 interactome analyzed by mass spectrometry. (Purple) Mitochondrial proteins exclusively present in FLAG-MTFP1 eluates or significantly enriched greater than twofold, listed in Supplementary Data 3. (Green) Non-mitochondrial proteins significantly more abundant in Tg heart. (Blue) Non-mitochondrial proteins significantly more abundant in WT heart. **c** Functional classification of 60 mitochondrial proteins identified in Co-IP eluates in **b** (Supplementary Data 3). **d** Second-dimension electrophoresis (2D BN-PAGE) of the cardiac OXPHOS complexes isolated from WT (top) and cMKO (bottom) mice at 8–10 weeks and previously separated in a BN-PAGE. Detection of components of the mPTP complex was performed using the indicated antibodies.

using a plasmid encoding SV40 large T antigen. MEFs cells were maintained in Dulbecco's modified Eagle's medium (DMEM + GlutaMAX, 4.5 g/L D-Glucose, pyruvate) supplemented with 5% FBS and 1% penicillin/streptomycin (P/S, 50 μg/ml) in a 5% $CO_2$ atmosphere at 37 °C.

Genetic disruption of *Ppif* in WT and *Mtfp1*[−/−] MEFs was performed via CRISPR-Cas9 gene editing targeting Exon 1 of *Ppif* (sgRNA): forward: aaacCCGGGAACCCGCTCGTGTAC and reverse: CACCGTACACGAGCGGGTTCCCGG. sgDNA oligonucleotides were annealed and cloned into pSpCas9(BB)−2A-GFP PX458 to generate pTW363

(pSpCas9(BB)−2A-GFP PPIFsgDNA). WT (*Mtfp1*[+/+]) and *Mtfp1*[−/−] immortalized MEFs were transfected with 2.5 μg of plasmid pTW363 using Lipofectamine 2000 (Life Technologies). After 48 h incubation, cells were individually isolated in 96 well plates by FACS. Clones were then expanded and validated by Sanger sequencing and western blotting. Both single Ppif KO MEFs (*Ppif*[−/−]) and double KO *Mtfp1*[−/−]*Ppif*[−/−] cells carry the same homozygous c.126delG insertion that is predicted to yield a truncated polypeptide product at amino acid position 65 (1-64 protein) p.Val65*. Immunoblot analysis was used to confirm the absence of PPIF protein.

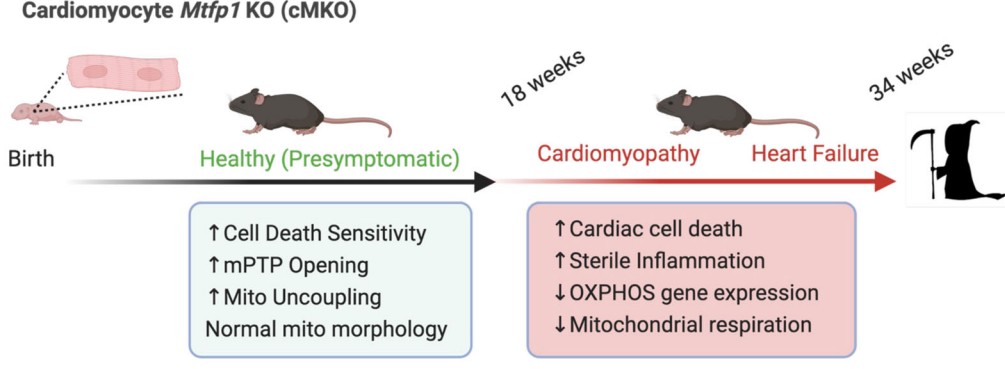

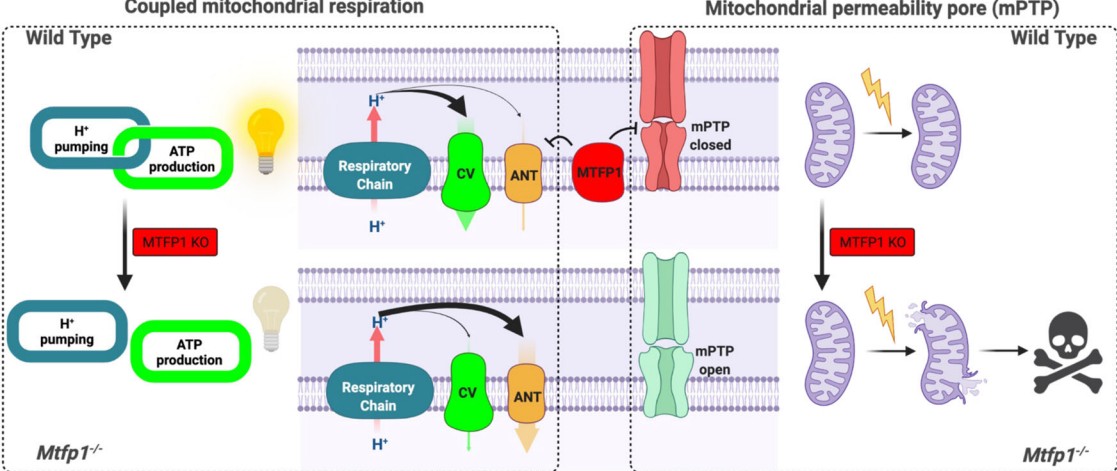

**Fig. 7 | Model for the regulation of mitochondrial and cardiac function by MTFP1.** *(Top) Mtfp1* deletion in cardiomyocytes occurs at birth (cMKO) and sensitizes cardiac myocytes to mitochondrial permeability transition pore (mPTP) opening, cell death and increases mitochondria uncoupling of the inner membrane. At the adult age of 8–10 weeks heart of cMKO mice have normal structure and function but undergoes to the development of a progressive dilated cardiomyopathy (DCM) at 18 weeks which progresses to severe heart failure and middle-aged death by 34 weeks. At onset of DCM, cMKO mice exhibit increased cardiac cell death, reduced mitochondrial respiration, and induction of a sterile inflammatory response. (Bottom) Coupled respiration and mPTP closure is maintained by MTFP1. Genetic deletion of *Mtfp1* promotes the ANT-dependent uncoupling of mitochondrial respiration and opening of the mPTP, sensitizing cells to programmed cell death. Figure created with BioRender.

MEFs expressing mitochondrially targeted YFP (mitoYFP) were generated from Gt(ROSA26)Sor[mitoYFP/+] embryos on a C57Bl6/N genetic background at E13.5 and immortalized using a plasmid encoding SV40 large T antigen as previously described[28]. MEFs stably expressing FLAG-MTFP1 were generated by lentiviral transduction with pTW142 (pLVX-EF1α-MTFP1) containing a puromycin-resistant marker. The empty vector (EV) pLVX-EF1α (pTW122) was used to generate control cells.

Human U2OS osteosarcoma cells were obtained from commercial sources (ATCC) and depleted for MTFP1 by CRISPR-Cas9 gene editing (*MTFP1^KO*). The single-guide RNAs (sgRNAs) were designed using the CRISPR-Cas9 design tool (benchling.com) to target Exon 1 of *MTFP1*. sgDNA oligonucleotides (forward: 5′- caccgGCGCAGAGCGCGATCTCTAC −3′ and reverse: 5′-aaacGTAGAGATCGCGCTCTGCGCc-3′) were annealed and cloned into the BbsI digested pSpCas9(BB)−2A-GFP vector (SpCas9(BB)−2A-GFP (PX458)) which was a gift from Feng Zhang (Addgene plasmid # 48138). U2OS cells were transfected in 6-well dishes with 2ug of pSpCas9(BB)−2A-GFP plasmid containing the respective sgRNA using Lipofectamine 2000 (Life Technologies, 11668027). After 24 h incubation, GFP positive cells were individually isolated by fluorescence-activated cell sorting. Clones were expanded and were validated by western blotting and DNA sequencing.

Mouse atrial cardiomyocytes HL-1 cells, obtained from the lab of Sigolene Meilhac (Institut Imagine/Université Paris Cite), were cultured on fibronectin (0.5%)/gelatin(0.02%) coated flasks in Claycomb medium (Sigma-Aldrich) supplemented with 10% FBS, 100 U/mL P/S, 0.1 mM norepinephrine and 2 mM L-glutamine.

### SDS-PAGE immunoblot analysis
Immunoblot analysis was used to assess steady-state protein levels in cardiac tissue and cell lysates. For tissue lysates, mice were sacrificed by cervical dislocation, the chests were opened, and the hearts were excised, weighed, flash frozen in liquid nitrogen, and stored at −80 °C until use. The left ventricle (LV) tissue or MEFs cellular pellet was homogenized in cold RIPA buffer [1 mg/ 20 μL, 1% Triton X-100, 1% sodium deoxycholate, 0.1% SDS, 150 mM NaCl, 50 mM Tris-HCl (pH 7.8), 1 mM EDTA, and 1 mM EGTA] in presence of protease and phosphatase inhibitors and kept on ice for 30 minutes. The homogenate was then centrifuged for 15 min at 16,000 g, 4 °C. The protein concentration was determined by Bradford assay (Bio-Rad) using a BSA standard curve. The protein absorbance was measured at 595 nm by using a microplate reader Infinite M2000 (Tecan). Equal amounts of protein were reconstituted in 4x Laemmli Sample Buffer [355 mM, 2-mercaptoethanol, 62.5 mM Tris-HCl pH 6.8, 10% (v/v) glycerol, 1%(w/v)

SDS, 0.005% (v/v) Bromophenol Blue] and heated at 95 °C for 5 min. Samples (10 µg) were resolved on 4-20% polyacrylamide gels (Mini Protean TGX Stain-Free gels, BioRad) and transferred to nitrocellulose membrane with Trans-Blot Turbo Transfer system (Bio-Rad). Equal protein amount across membrane lanes were checked by Ponceau S staining or Stain-free detection. Membranes were blocked for at least 1 h with 5% (w/v) semi-skimmed dry milk dissolved in Tris-buffered saline Tween 0.1% (TBST), incubated overnight at 4 °C with primary antibodies dissolved 1:1,000 in 2% (w/v) Bovine Serum Albumin (BSA), 0.1% TBST. The next day membranes were incubated in secondary antibodies conjugated to horseradish peroxidase (HRP) at room temperature for 2 h (diluted 1:10,000 in BSA 2% TBST 0.1%). Finally, membranes were incubated in Clarity Western ECL Substrate (Bio-Rad) for 2 min and luminescence was detected using the ChemiDoc Gel Imaging System. Densitometric analysis of the immunoblots was performed using Image Lab Software v.6.1.0 (Bio-Rad).

## Mitochondrial isolation

Isolated cardiac mitochondria were freshly isolated as previously described[47] with some modifications. Briefly, the heart was washed in ice-cold PBS solution. Ventricles were separated from atria and non-myocardial tissue, cut in small pieces, and then transferred to an ice-cold 2-ml homogenizer (Teflon pestle) and manually homogenized in IB buffer (sucrose 275 mM, Tris 20 mM, EGTA-KOH 1 mM, pH 7.2) containing Trypsin-EDTA (0.05%). Trypsin activity was then inhibited by adding to the homogenate bovine serum albumin (BSA) fatty acid free (0.25 mg/mL) and protease inhibitor cocktail (PIC, Roche).

Mitochondria isolation from MEFs was performed starting from 10 × 150 mm dishes at 100% confluence. Cells were collected into 10 mL of IB Buffer containing BSA fatty acid-free (0.25 mg/mL) and PIC and homogenized with 30 stokes of the plunger at 1500 rpm on ice.

Cardiac and/or cellular homogenates were then centrifuged at low speed (1000 g, 10 min, 4 °C) to discard nuclei and debris, and further centrifuged (3200 g, 15 min, 4 °C) to obtain the crude mitochondrial pellet and the cytosolic fraction. The crude mitochondrial pellet was finally resuspended in IB buffer containing BSA and PIC and protein concentration was determined by using the Bradford assay.

## Protease protection assay

Crude mitochondria isolated from WT mouse hearts was subjected to protease protection assay as previously described[90]. 50 µg of crude mitochondria were resuspended into the following buffers without protease inhibitors: (1) Mitochondrial isolation IB buffer (2) Mitochondrial isolation buffer with Proteinase K (100 µg/mL) (3) Swelling buffer (EDTA 1 mM, HEPES 40 mM) (4) Swelling buffer + Proteinase K (100 µg/mL) (5) Swelling buffer + Proteinase K (100 µg/ml) + Triton X-100 0.5% and incubated at 37 °C for 30 min at 750 rpm. Mitochondrial proteins were then precipitated in trichloric acid buffer (14% TCA, 40 mM HEPES, 0.02% Triton X-100) for 3 h at −20 °C and centrifugated for 20 min. Pellets were washed twice with ice-cold acetone and air-dried prior to re-suspension in 1x Laemmli sample buffer (Bio-Rad) for SDS-PAGE and immunoblot analysis.

## Alkaline carbonate extraction

Alkaline carbonate extraction of membrane proteins was performed as previously described[90]. Crude mitochondria isolated from WT mouse hearts were resuspended and incubated for 30 min on ice with 0.1 M Na$_2$CO$_3$ at the following pH: 12.5, 11.5, 10.5, or 9.5. The suspensions were ultra-centrifuged at 4 °C for 30 min at 90,000 g in Beckman polycarbonate tubes in a TLA 110 rotor. Supernatants and pellets were then incubated in trichloric acid buffer (14% TCA, 40 mM HEPES, 0.02% Triton X-100) for 15 min on ice, followed by centrifugation for 20 min at 28,000 g at 4 °C. The samples were washed 3x with 100% acetone, subsequently dried for 30 min at RT. The dried pellet was then

resuspended with 1x Laemmli sample buffer (Bio-Rad) for SDS-PAGE and western blot analysis.

## siRNA transfection

Silencing of the indicated genes was performed using forward transfection: 20 nM of the indicated SmartPool siRNAs listed in Supplementary Data 4 were mixed with Lipofectamine RNAiMax (Invitrogen), added on top of cells seeded in 6-well dishes (300,000 cells/well) and incubated at 37 °C in a CO$_2$ incubator. For live imaging experiments, cells were seeded in 96 or 384 well plates (Cell Carrier Ultra, Perkin Elmer) 48 h after transfection to perform live imaging after 72 h post-transfection.

## Cardiac RNA sequencing and RT-qPCR

WT and cMKO mice aged 8 and 18 weeks were sacrificed by cervical dislocation and hearts were quickly excised and rinsed with cold sterile PBS. Left ventricle posterior walls were collected and snap frozen in liquid nitrogen and stored at −80 °C until use. Total RNA was extracted by using TRIzol (Invitrogen, NY, USA) according to standard procedures. The Trizol/chloroform mixture was centrifuged at 12,000 g for 15 min at 4 °C, the supernatant corresponding to the aqueous phase was transferred to a new tube, mixed with ethanol 70%, and then applied to an RNA mini-column extraction kit (NucleoSpin RNA kit, MACHEREY-NAGEL) according to the manufacturer's recommendations. After DNA digestion, total RNA was eluted with RNase-free water and stored in liquid nitrogen. RNA was quantified by using the Bioanalyzer 2100 (Agilent) and only RNA samples (≥50 ng/µL) with an RNA integrity value (RIN) ≥ 6.8, OD$^{260}$/$^{280}$ ≥ 2, OD$^{260}$/$^{230}$ ≥ 2 were considered for the transcriptome profiling. For RNA sequencing, libraries were built using a TruSeq Stranded mRNA library Preparation Kit (Illumina, USA) following the manufacturer's protocol. Two runs of RNA sequencing were performed for each library on an Illumina NextSeq 500 platform using single-end 75 bp. The RNA-seq analysis was performed with Sequana 0.8.5[91] using an RNA-seq pipeline 0.9.13 (https://github.com/sequana/sequana_rnaseq) built on top of Snakemake 5.8.1[92]. Reads were trimmed from adapters using Cutadapt 2.10[93] then mapped to the mouse reference genome GRCm38 using STAR 2.7.3a[94]. FeatureCounts 2.0.0 was used to produce the count matrix, assigning reads to features using annotation from Ensembl GRCm38_92 with strand-specificity information[95]. Quality control statistics were summarized using MultiQC 1.8[96]. Statistical analysis on the count matrix was performed to identify differentially expressed genes (DEGs), comparing WT and cMKO. Clustering of transcriptomic profiles were assessed using a Principal Component Analysis (PCA). Differential expression testing was conducted using DESeq2 library 1.24.0 scripts based on SARTools 1.7.0[97] indicating the significance (Benjamini-Hochberg adjusted p-values, false discovery rate FDR < 0.05) and the effect size (fold-change) for each comparison.

For RT-qPCR, 1 µg of total RNA was converted into cDNA using the iScript Reverse Transcription Supermix (Bio-Rad). RT-qPCR was performed using the CFX384 Touch Real-Time PCR Detection System (Bio-Rad) and SYBR Green Master Mix (Bio-Rad) using the primers listed in Supplementary Data 1. Gapdh was amplified as internal standard. Data were analyzed according to the 2−ΔΔCT method[98].

## Proteomics

Heart extract proteins from pre-symptomatic (8–10 weeks) and symptomatic mice (18 weeks) were extracted and denatured in RIPA buffer. Samples were sonicated using a Vibracell 75186 and a miniprobe 2 mm (Amp 80%//Pulse 10 off 0.8, 3 cycles) and further centrifuged. Protein assay was performed on supernatant (Pierce 660 nm, according to manufacturer instructions) and 100 µg of each extract was delipidated and cleaned using a Chloroform/Methanol/Water precipitation method. Briefly, 4 volume of ice-cold methanol were added to the sample and vortex, 2 volume of ice-cold chloroform were

added and vortex, and 3 volume of ice-cold water was added and vortex. Samples were centrifuged 5 min at 5000 $g$. The upper layer was removed and proteins at the interface were kept. Tubes were filled with ice-cold methanol and centrifuged at max speed for 5 min. Resulting protein pellet was air-dried and then dissolved in 130 µl of 100 mM NaOH before adding 170 µl of Tris 50 mM pH 8.0, tris (2-carboxyethyl) phosphine (TCEP) 5 mM and chloroacetamide (CAA) 20 mM. The mixture was heated 5 min at 95 °C and then cooled on ice. Endoprotease LysC (1 µg) was use for a 8 h digestion step at (37 °C) followed with a trypsin digestion (1 µg) at 37 °C for 4 h. Digestion was stopped adding 0.1% final of trifluoroacetic acid (TFA). Resulting peptides were desalted using a C18 stage tips strategy (Elution at 80% Acetonitrile (ACN) on Empore C18 discs stacked in a P200 tips) and 30 µg of peptides were further fractionated in 4 fractions using poly(-styrenedivinylbenzene) reverse phase sulfonate (SDB-RPS) stage-tips method as previously described[99,100]. Four serial elutions were applied as following: elution 1 (80 mM Ammonium formate (AmF), 20% (v/v) ACN, 0.5% (v/v) formic acid (FA)), elution 2 (110 mM AmF, 35% (v/v) ACN, 0.5% (v/v) FA), elution 3 (150 mM AmmF, 50% (v/v) ACN, 0.5% (v/v) FA) and elution 4 (80% (v/v) ACN, 5% (v/v) ammonium hydroxide). All fractions were dried and resuspended in 0.1% FA before injection.

LC-MS/MS analysis of digested peptides was performed on an Orbitrap Q Exactive Plus mass spectrometer (Thermo Fisher Scientific, Bremen) coupled to an EASY-nLC 1200 (Thermo Fisher Scientific). A home-made column was used for peptide separation ($C_{18}$) 50 cm capillary column picotip silica emitter tip (75 µm diameter filled with 1.9 µm Reprosil-Pur Basic $C_{18}$-HD resin, (Dr. Maisch GmbH, Ammerbuch-Entringen, Germany)). It was equilibrated and peptide were loaded in solvent A (0.1% FA) at 900 bars. Peptides were separated at 250 nl min$^{-1}$. Peptides were eluted using a gradient of solvent B (ACN, 0.1% FA) from 3 to 7% in 8 min, 7 to 23% in 95 min, 23 to 45% in 45 min (total length of the chromatographic run was 170 min including high ACN level step and column regeneration). Mass spectra were acquired in data-dependent acquisition mode with the XCalibur 2.2 software (Thermo Fisher Scientific, Bremen) with automatic switching between MS and MS/MS scans using a top 12 method. MS spectra were acquired at a resolution of 35000 (at $m/z$ 400) with a target value of $3 \times 10^6$ ions. The scan range was limited from 300 to 1700 $m/z$. Peptide fragmentation was performed using higher-energy collision dissociation (HCD) with the energy set at 27 NCE. Intensity threshold for ions selection was set at $1 \times 10^6$ ions with charge exclusion of $z = 1$ and $z > 7$. The MS/MS spectra were acquired at a resolution of 17,500 (at $m/z$ 400). Isolation window was set at 1.6 Th. Dynamic exclusion was employed within 45 s.

Data were searched using MaxQuant (version 1.5.3.8) using the Andromeda search engine[101] against a reference proteome of Mus musculus (53449 entries, downloaded from Uniprot the 24th of July 2018). The following search parameters were applied: carbamidomethylation of cysteines was set as a fixed modification, oxidation of methionine and protein N-terminal acetylation were set as variable modifications. The mass tolerances in MS and MS/MS were set to 5 ppm and 20 ppm respectively. Maximum peptide charge was set to 7 and 5 amino acids were required as minimum peptide length. A false discovery rate of 1% was set up for both protein and peptide levels. All 4 fractions per sample were gathered and the iBAQ intensity was used to estimate the protein abundance within a sample[102]. The match between runs features was allowed for biological replicate only.

For this large-scale proteome analysis part, the mass spectrometry data have been deposited at the ProteomeXchange Consortium (http://www.proteomexchange.org) via the PRIDE partner repository[103,104] with the dataset identifier PXD028516 (symptomatic mice) and (pre-symptomatic mice).

## mtDNA content quantification
Genomic DNA was extracted using the NucleoSpin Tissue (MACHEREY-NAGEL) and quantified with NanoQuant Plate (Infinite M200, TECAN).

RT-qPCR was performed using the Real-Time PCR Detection System (Applied Biosystems StepOnePlus), 20 ng of total DNA, and the SYBR Green Master Mix (Bio-Rad). b-*Actin* was amplified as an internal, nuclear gene standard as previously described[28]. PCR primer sequences are listed in Supplementary Data 4. Data were analyzed according to the $2^{-\Delta\Delta CT}$ method[98].

## Mitochondrial imaging and quantification in MEFs
MEFs cells were seeded on 96-well or 384-well CellCarrier Ultra imaging plates (Perkin Elmer) 24 h before imaging. Nuclei were stained with NucBlue Live ReadyProbes Reagent (ThermoFisher Scientific) at 1 drop per 10 mL of media. Fluorescent labeling of mitochondria was achieved using MitoTracker DeepRed (MTDR) at 100 nM for 30 min at 37 °C, 5% $CO_2$. Cells were then washed with regular medium and images were acquired using the Operetta CLS or Opera Phenix High-Content Analysis systems (Perkin Elmer), with 20x Water/1.0 NA, 40x Air/0.6 NA, 40x Water/1.1 NA or 63x Water/1.15 NA. MTDR (615-645 nm) and NucBlue (355-385 nm) were excited the appropriate LEDs (Operetta CLS) or lasers (Opera Phenix). Automatic single-cell classification of non-training samples (i.e., unknowns) was carried out by the supervised machine-learning (ML) module using Harmony Analysis Software v.4.9 (PerkinElmer) as previously described[28].

## Analysis of oxygen consumption
Oxygen consumption of intact cells or crude mitochondria was measured with the XFe96 Analyzer (Seahorse Biosciences) and High-Resolution Respirometry (O2k-Fluorespirometer, Oroboros, AT), respectively. For Seahorse experiments, cells (experimentally optimized density of 20,000 cells/well for MEFs and U2OS cells, 10,000 cells/well for HL-1 cells) were seeded onto 96-well XFe96 cell culture plates. On the following day, cells were washed and incubated with Seahorse XF Base Medium completed with 1 mM Pyruvate, 2 mM Glutamine, and 10 mM Glucose. Cells were washed with the Seahorse XF Base Medium and incubated for 45 min in a 37 °C non-$CO_2$ incubator before starting the assay. Following basal respiration, cells were treated sequentially with: oligomycin 1 µM, CCCP 2 µM, and antimycin A plus rotenone (1 µM each) (Sigma). Measurements were taken over 2-min intervals, proceeded by a 1-min mixing and a 30 s incubation. Three measurements were taken for the resting OCR, three for the non-phosphorylating OCR, three for the maximal OCR, and three for the extramitochondrial OCR. After measurement, the XFe96 plate was washed with Phosphate-Buffered Saline (PBS) and protein was extracted with RIPA for 10 min at RT. Protein quantity in each well was then quantified by Bicinchoninic acid assay (BCA) by measuring absorbance at 562 nm and used to normalize OCR measurements as previously described[28].

For O2k respirometry, isolated cardiac mitochondria were freshly isolated from adult WT and cMKO mice (8, 18, or 34 weeks) as described above. Simultaneous measurement of mitochondrial respiration and membrane potential ($\Delta\psi$) was assessed by O2K-Fluorometry using a O2K-Fluorescence LED2-Module operated through the amperometric channel of the O2K. Briefly, 25 or 50 µg of cardiac mitochondria were resuspended in Mir05 buffer [$MgCl_2$–$6H_2O$ 3 mM, Lactobionic Acid 60 mM, Taurine 20 mM, $KH_2PO_4$ 10 mM, Hepes-KOH 20 mM, Sucrose 110 mM, EGTA-KOH 0.5 mM, BSA (1 g/L)] in presence of Rhodamine 123 (RH-123) (0.66 µM). Maximal mitochondrial respiration capacity (OXPHOS) was determined under consecutive administrations of PGM [pyruvate 10 mM, glutamate 5 mM, malate 5 mM, state 2] or malate (2 mM) and palmitoyl-carnitine (10 µM) in presence of ADP (1 mM, state 3) to assess complex I-driven respiration; rotenone (0.5 µM) and succinate (10 mM, state 2) for the measurement of complex II-driven respiration; antimycin A (2.5 µM), ascorbate (2 mM), *N,N,N',N'*-tetramethyl-*p*-phenylenediamine (TMPD, 0.5 mM) and carbonyl cyanide m-chlorophenyl hydrazone (CCCP, 2 µM) for the determination of the complex IV-driven respiration. The

addition of oligomycin (Omy, 25 nM), carboxyatractyloside (CATR, 1 μM) or GTP (1 mM) was used to determine the non-phosphorylating respiration (state 4). Cytochrome $c$ (2 μM)–mediated $O_2$ flux was used to evaluate the integrity of the OMM following the mitochondria isolation. Respiratory control ratios (RCR) of state 3: state 4 were used to evaluate the integrity of the IMM. RH-123 fluorescence quenching (Δ fluorescence) in energized mitochondria was used as a direct measurement of Δψ. Data was analyzed using Datlab v.7

### Echocardiography
Transthoracic echocardiographic acquisitions were performed by using a Vevo 3100 Imaging System coupled to a 25–55 MHz linear-frequency transducer (MX550D, FUJIFILM VisualSonics). Randomized WT and cMKO mice [10 to 34 weeks of age (male), 34 weeks (female)] were anesthetized and maintained with 2% isofluorane in oxygen, placed in a supine position on a 37 °C warmed pad. Limb electrodes and a rectal probe were used to monitor define (ECG) and body temperature. Before echocardiography and the addition of the prewarmed ultrasound gel, thorax fur was removed using hair-removal cream. To assess the overall left ventricle (LV) size and LV function, B- and M-Mode images were acquired in parasternal long axis view (PLAX) between 400-500 bpm. The systolic and diastolic LV dimensions [interventricular septum thickness (IVS; mm), LV diameter (LVD; mm), LV posterior wall thickness (LVPW; mm)] and cardiac output [ejection fraction (% EF)] were determined by acquiring and analyzing at least 3 independent cardiac cycles within at least 3 M-Mode images analyzed with Vevo Lab (VisualSonics).

### Doxorubicin treatment
WT and cMKO mice aged 8 weeks received a cumulative dose of 20 mg/kg of doxorubicin (Doxo, 2 intra-peritoneal injections of 10 mg/kg/saline, given at 3-day intervals; control mice received saline injections). Transthoracic echocardiography was performed at day 14 post-treatment. Body weight was recorded every 3 days and a loss higher than 20% was considered as endpoint of the study.

### Determination of serum levels of cardiac troponin I and cardiac MLC1
A mouse ELISA kit was used to compare serum levels of cardiac troponin I (cTnI, Life Diagnostics) and cardiac myosin light chain 1 (MLC1, Life Diagnostic) in WT and cMKO male mice aged 18 and 34 weeks. Blood was collected via submandibular vein puncture from non-anesthetized mice, left for 30 min at RT, and then centrifuged at 5000 $g$ at 4 °C for 10 min then snap-frozen into liquid nitrogen. Serum was stored at −80 °C until next use. The assays to determine cTnI and MLC1 levels were performed following the exact manufacturer's instructions.

### Histology
WT and cMKO mice aged 34 weeks were sacrificed by cervical dislocation and the hearts excised post-mortem. The whole hearts were then fixed with Formol 4% (VWR chemicals) overnight and then fully dehydrated by a series of ethanol gradients. Tissues were then paraffin-embedded and sectioned in a short view at a thickness of 4 μm on a microtome. The sections were de-paraffinized in xylene and rehydrated followed by hematoxylin and eosin (H&E), Masson's trichrome or Picrosirius Red staining according to the standard protocols. Images were acquired with the slide scanner Olympus vs120 and visualized with OlyVIA (Olympus).

### Transmission electron microscopy
Transmission electron microscopy was performed on cardiac tissue from WT and cMKO mice aged 8–10 weeks (pre-symptomatic) and 18 weeks (symptomatic). Small pieces (1 x 1 x 1 mm) from the left ventricle posterior wall were fixed in a 37 °C prewarmed mix of PHEM 1x

buffer (60 mM PIPES, 25 mM HEPES, 10 mM EGTA, 2 mM $MgCl_2$; pH 7.3), 2.5% glutaraldehyde and 2% PFA for 30 min, followed by an overnight fixation at 4 °C. Specimens were then rinsed 3 times with PHEM 2X buffer. Samples were incubated with 1% osmium tetroxide (Merck) and 1.5% ferrocyanide (Sigma Aldrich) in 0.1 M PHEM. After dehydration by a graded series of ethanol, samples were gradually infiltrated at RT with epoxy resin and after heat polymerization, sections with a thickness of 70 nm were cut with a Leica UCT microtome and collected on carbon, formvar coated copper grids. Sections were contrasted with 4% aqueous uranylacetate and Reynold's lead citrate. Generation of ultra-large electron microscopy montages at a magnification of 14500x (pixel size = 6.194 nm, bin 1, US4000 Ultrascan camera) were acquired using a TECNAI F20 Transmission Electron Microscope (FEI) with a field emission gun (FEG) as an electron source, operated at 200 kV. The SerialEM software[105,106] was used for multi-scale mapping as previously described[28]. The 'Align Serial Sections/Blend Montages' interface of IMOD16 was used for blending the stacks of micrographs collected to single large images. Quantification of mitochondrial cross- sectional area was determined by tracing the perimeter of individual mitochondria with the use of 3dmod software.

### Primary adult cardiomyocytes isolation
Primary adult ventricular cardiomyocytes (CMs) were isolated from WT ($Myh6$-$Cre^{+/+}$, $Mtfp1^{LoxP/LoxP}$) and $Mtfp1$ KO mice (cMKO, $Myh6$-$Cre^{Tg/+}$, $Mtfp1^{LoxP/LoxP}$) mice aged 8-10 weeks to perform live-cell imaging experiments by using the simplified Langendorff-free method as previously published[107] with some modifications. Briefly, mice were weighed, injected intraperitoneally with ketamine/xylazine (ketamine: $3 \times 80$ mg/kg/bw; xylazine $3 \times 10$ mg/kg/bw). Once the thorax was opened, the descending aorta was cut and the heart exposed and flushed into the right ventricle with EDTA buffer pH 7.8 (NaCl 130 mM, KCl 5 mM, $NaH_2PO_4$ 0.5 mM, HEPES 10 mM, Glucose 10 mM, 2,3-butanedione monoxime (BDM) 10 mM, Taurine 10 mM, EDTA 5 mM). The heart was then excised, transferred into a dish containing EDTA buffer, where the ascending aorta was clamped. Tissue digestion was performed by sequential injection of EDTA buffer, perfusion buffer (NaCl 130 mM, KCl 5 mM, $NaH_2PO_4$ 0.5 mM, HEPES 10 mM, Glucose 10 mM, BDM 10 mM, Taurine 10 mM, MgCl2 1 mM, pH 7.8) and collagenase buffer (Collagenase II 0.5 mg/mL, Collagenase 4 0.5 mg/mL, Protease XIV 0.05 mg/mL) into the left ventricle (LV) through the apex. To control the flow of the digestion buffer the LV was perfused via a flexible linker using an automated infusion pump (Graseby 3200). After digestion, ventricles were then separated from the atria and ultimately dissociated by gentle pipetting. The digestion process was then inhibited by the addition of the stop buffer (perfusion buffer containing FBS 10%). Cellular suspension was filtered using a 100 μm pore size strainer to remove undigested debris. CMs were then collected by gravity settling (15 min) while resuspended sequentially in 3 calcium reintroduction buffers to gradually restore calcium levels (0.34 mM, 0.68 mM, and 1.02 mM $Ca^{2+}$). CMs yields were quantified using a hemocytometer. CMs were resuspended in prewarmed culture media (M199 Medium, BSA 0.1%, Insulin-Transferrin-Selenium (ITS) 1x, BDM 10 mmol/l, CD lipid 1x, P/S 1x), and plated into laminin (10 μg/mL) precoated Cell Carrier Ultra-96 well (Perkin Elmer) plates, in a humidified tissue culture incubator (37 °C, 5% $CO_2$) for at least 1 hour before proceeding to live imaging.

### Cell death assay
CMs isolated from WT and cMKO mice (8-10 weeks) labelled with NucBlue Live ReadyProbes Reagent (ThermoFisher Scientific) and Tetramethylrhodamine Ethyl Ester Perchlorate (TMRE; 50 nM) for 20 min at 37 °C, 5% $CO_2$. Supervised machine learning (Harmony 4.9, Perkin Elmer) using TMRE intensity, cell geometry (rod- shape, area, length) and nuclei number as read-outs enabled the on-the-fly detection, imaging, and quantification of healthy CMs from dead and dying

CMs and non-myocytes. Live-cell imaging was performed in the presence or absence of a cell death stimuli: $H_2O_2$ (25 μM), or doxorubicin (Doxo, 60 μM,) or carbonyl cyanide m-chlorophenyl hydrazine (CCCP, 10 μM). Image acquisition was performed using the Operetta CLS High-Content Analysis system (Perkin Elmer) with a 5x air objective.

WT, $Mtfp1^{-/-}$, $Ppif^{-/-}$ and $Mtfp1^{-/-}Ppif^{-/-}$ MEFs were plated in Cell Carrier Ultra-96 well (Perkin Elmer) and incubated 24 h with regular media. Cells were then stained with NucBlue Live ReadyProbes to visualize nuclei. CellEvent Caspase 3/7 Green (CE) was added (1 drop per 1 mL of media) to visualize the activation of caspases 3/7 in cells undergoing to apoptosis. Propidium Iodide (PI, 1:500) was added to visualize dead cells. Cell death was induced with actinomycin D (1.5 μM) alone, or in association with ABT-737 (10 μM), an inhibitor of the Bcl-2 family proteins, or doxorubicin hydrochloride (DOXO, 1.5 μM), or staurosporine (1 μM) or hydrogen peroxide $H_2O_2$ (500 μM).

After 48 h silencing, HL-1 myocytes (3000 cells/well) were plated in a Cell Carrier Ultra-384 well (Perkin Elmer) and incubated 24 h with supplemented Claycomb media. Cell death was induced with staurosporine (1 μM) or hydrogen peroxide $H_2O_2$ (300 μM) during 17 h.

The pan-caspase inhibitor Q-VD-OPh (qVD, 20 μM) and cyclophilin D inhibitor cyclosporin A (CsA, 2 μM), were added where indicated.

Image acquisition was performed every hour using the Operetta CLS High-Content Analysis system (Perkin Elmer) with a 20x air objective. CE or PI positive nuclei number were automatically counted by using Harmony v.4.9 software.

## Membrane potential measurement

Membrane potential was determined by live confocal microscopy. HL-1 cardiac myocytes (10,000 cells/well) were plated in 96well CellCarrier Ultra imaging plates (PerkinElmer) coated with gelatin/fibronectin. The day of the experiment, nuclei were labeled with NucBlue Live Ready Probes Reagent (Thermo Fisher Scientific) and loaded with 100 nM Tetramethylrhodamine Ethyl Ester Perchlorate (TMRE) and 100 nM MitoTracker Deep Red (MTDR) for 20 min at 37 °C, 5% $CO_2$. Confocal images were acquired using the Operetta CLS High-Content microscope (PerkinElmer) with 40× Air/0.6 NA and excited with the appropriate LEDs. TMRE and MTDR signal intensities per cell were quantified using the Harmony Analysis Software v4.9 (PerkinElmer).

## Field-stimulated cardiomyocytes

Adult ventricular cardiomyocytes (CMs) from WT and cMKO mice (8–10 weeks) were isolated with a Langendorff system as previously described[108,109] and paced by electrical field stimulation at 37 °C using a customized IonOptix system. CMs were exposed to a protocol that simulates a physiological workload increase by first pacing cells at 0.5 Hz in Normal Tyrode´s solution pH 7.4 [NT; NaCl 130 mM, KCl 5 mM, $MgCl_2$ 1 mM, $CaCl_2$ 1 mM, Na-HEPES 10 mM, glucose 10 mM, sodium pyruvate 2 mM and ascorbic acid 0.3 mM]. The β-adrenergic receptor agonist isoproterenol (30 nM) was then washed in for 1 min and then the stimulation rate was increased to 5 Hz for 3 min. After this time, stimulation rate was set back to the initial 0.5 Hz, and isoproterenol washed out. During the measurements or sarcomere length and contraction, the system detection was combined to recordings of the autofluorescence of NAD(P)H/NAD(P)+ and $FADH_2$/FAD by alternately exciting cells at wavelengths ($\lambda_{exc}$) of 340 and 485 nm and collecting emission ($\lambda_{em}$) at 450 and 525 nm for NAD(P)H and FAD+, respectively. Calibration was performed inducing maximal oxidation and reduction of NAD(P)H/$FADH_2$ with FCCP (5 μmol/L) and cyanide (4 mmol/L), respectively.

## Intracellular calcium measurement

LV cardiac myocytes were isolated by enzymatic digestion and were paced by electrical field stimulation at 37 °C using a customized IonOptix system as described previously[110,111]. $[Ca^{2+}]_C$ was measured by incubating cells with indo-1 AM (5 μmol/L) for 20 min at 25 °C ($\lambda_{exc}$ = 340 nm, $\lambda_{em}$ = 405/485 nm). The bath solution contains (mM): 130 NaCL; 5 KCl; 1 MgCL2; 10 sodium HEPES; 2 sodium pyruvate; 0.3 ascorbic acid; 10 glucose and was adjust at pH 7.4 at 37 °C; Isoproterenol was used at $3 \times 10^{-8}$ M. The $Ca^{2+}$ transients were analysed with the IONWIZRD software from IONOptix. Data were collected and statistically analysed using paired or unpaired tTest or 2way ANOVA & Bonferroni's Multiple Comparison Test.

## Mitochondrial swelling and mPTP opening

Calcium-induced mitochondrial permeability transition pore (mPTP) opening was determined in freshly isolated cardiac mitochondria from WT and cMKO mice aged 8–10 weeks. Mitochondria were suspended in $Ca^{2+}$ uptake buffer pH 7.4 (120 mM) KCl, 5 mM MOPS, 5 mM $KH_2PO_4$, 10 mM Glutamate, 5 mM Malate *plus* cOmplete, EDTA-free Protease Inhibitor Cocktail (Roche) at a concentration of 0.5 mg/mL and stimulated by the addition of a single pulse of 120 μM $CaCl_2$ or multiple pulses ($n = 12$) of 10 μM $CaCl_2$.

mPTP opening was assessed in mitochondria of WT, $Mtfp1^{-/-}$, $Ppif^{-/-}$ and $Mtfp1^{-/-}Ppif^{-/-}$ MEFs as previously described[64] at a concentration of 1 mg/mL in mitochondrial swelling buffer (120 mM KCl, 10 mM Tris pH 7.4, 5 mM $KH_2PO_4$, 7 mM pyruvate, 1 mM malate, and 10 μM EDTA). Swelling was induced by pulsing mitochondria with 250 μM $CaCl_2$ in EDTA-free buffer.

The absorbance (540 nm) was measured at intervals of ~20 s at 37 °C using the microplate reader Infinite M Plex (Tecan). Cyclosporin A (CsA, 1 μM Sigma) was used as a control to inhibit mPTP-dependent mitochondrial swelling.

## 1D BN-PAGE of cardiac mitochondria

First-dimension (1D) BN-PAGE was performed as described previously[112] with some modifications. Briefly, cardiac mitochondria were isolated from WT and cMKO mice at 8-10 weeks (pre-symptomatic) or 30 weeks (symptomatic) and solubilized with sample buffer (30 mM HEPES, 150 mM Potassium Acetate, 12% glycerol, 2 mM acid 6-aminohexanoic, 1 mM disodium EDTA) containing digitonin 3% (w/w). Lysates were incubated for 1 h at 4 °C and then centrifuged at 30,000 $g$ for 20 min. Supernatants were then mixed with G250 sample additive 1% (Invitrogen, BN2004) and resolved on 3–12% or 4–16% Bis-Tris Gels (1.0 mm) (Invitrogen, BN2011BX10) using the anode running buffer (Invitrogen, BN2001) and Cathode Buffer Additive added to the anode buffer (InVitroGen, BN2002). Gels were then incubated in Transfer Buffer (Tris (0.3% p/v), Glycine (0.44% p/v), Ethanol (10% v/v)) supplemented with SDS (0.2% v/v) and β-mercaptoethanol (0.2% v/v) for 30 minutes at RT to denature proteins.

Gels were then subjected to a semi-dry transfer using PVDF membranes (GE, 10600023). Membranes were washed with methanol and then blocked for at least 1 h with 5% (w/v) semi-skimmed dry milk dissolved in Tris-buffered saline Tween 0.1% (TBST), incubated overnight at 4 °C with primary antibodies dissolved 1:1.000 in 2% (w/v) Bovine Serum Albumin (BSA), 0.1% TBST. The next day membranes were incubated in secondary antibodies conjugated to horseradish peroxidase (HRP) at room temperature for 2 h (diluted 1:10,000 in BSA 2% TBST 0.1%). Finally, membranes were incubated in Clarity Western ECL Substrate (Bio-Rad) for 2 min and luminescence was detected using the ChemiDoc Gel Imaging System. Primary antibodies are listed in Supplementary Data 4.

## 2D BN-PAGE of cardiac mitochondria

1D BN-PAGE gels were resolved on a second-dimension electrophoresis (2D-SDS PAGE) using 4–20% polyacrylamide gels (Mini Protean TGX Stain-Free gels, BioRad). First-dimension bands were cut and incubated in MOPS1x SDS Running Buffer (Fisher Scientific) supplemented with β-mercaptoethanol (0.02% v/v) for 30 min at RT before the second-dimension electrophoresis. The gels were run in MOPS

buffer and then transferred to a nitrocellulose membrane. Immuno-detection was performed as previously described.

## Co-Immunoprecipitation assay and LC-MS/MS analysis

500 μg of cardiac crude mitochondria were freshly isolated from heart tissue of cardiomyocyte specific Flag-MTFP1 Knock-In (KI) mice (*Myh6-Cre*<sup>tg/+</sup>*Mtfp1*<sup>+/+</sup>, *CAG*<sup>Tg+/</sup>) and WT mice (*Myh6-Cre*<sup>+/+</sup>*Mtfp1*<sup>+/+</sup>, *CAG*<sup>Tg/+</sup>) as described above. Mitochondria were lysed in IP buffer (20 mM HEPES-KOH pH 7.5, 150 mM NaCl, 0.25% Triton X-100, protease inhibitor cocktail) on ice for 20 min and then centrifugated at 10,000 $g$, 4 °C for 15 min. Supernatant obtained by centrifugation was then incubated with 20 μL of anti-FLAG magnetic beads (Sigma M8823) for 2 h at 4 °C. The immunocomplexes were then washed with IP buffer without Triton X-100 and eluted with Laemmli Sample Buffer 2x at 95 °C for 5 min. Protein were stacked in a 15% SDS-PAGE gel with a 10 min long migration at 80 V. Proteins were fixed in gel and migration was visualized using the Instant Blue stain (Expedeon). Bands were excised for digestion. Gel bands were washed twice in Ammonium bicarbonate (AmBi) 50 mM, once with AmBi 50 mM/ACN 50% and once with 100% ANC. Gel band were incubated for 30 min at 56 °C in 5 mM dithio-threitol (DTT) solution for reduction. Gel bands were washed in AmBi 50 mM and then in 100% ACN. Alkylation was performed at room temp in the dark by incubation of the gel bands in Iodocateamide 55 mM solution. Gel bands were washed twice in AmBi 50 mM and in 100% ACN. 600 ng of trypsin were added for 8 h digestion at 37 °C. Peptides were extracted by collecting 3 washes of the gel bands using AmBi 50 mM/50% ACN and 5% FA. Peptides clean up and desalting was done using Stage tips (2 disc Empore C18 discs stacked in a P200 tip).

LC-MS/SM analysis of digested peptides was performed on an Orbitrap Q Exactive HF mass spectrometer (Thermo Fisher Scientific, Bremen) coupled to an EASY-nLC 1200 (Thermo Fisher Scientific). A home-made column was used for peptide separation (C$_{18}$ 30 cm) capillary column picotip silica emitter tip (75 μm diameter filled with 1.9 μm Reprosil-Pur Basic C$_{18}$-HD resin, (Dr. Maisch GmbH, Ammer-buch-Entringen, Germany)). It was equilibrated and peptide were loaded in solvent A (0.1% FA) at 900 bars. Peptides were separated at 250 nl min$^{-1}$. Peptides were eluted using a gradient of solvent B (ACN, 0.1% FA) from 3 to 26% in 105 min, 26 to 48% in 20 min (total length of the chromatographic run was 145 min including high ACN level step, and column regeneration). Mass spectra were acquired in data-dependent acquisition mode with the XCalibur 2.2 software (Thermo Fisher Scientific, Bremen) with automatic switching between MS and MS/MS scans using a top 12 method. MS spectra were acquired at a resolution of 60,000 (at $m/z$ 400) with a target value of $3 \times 10^6$ ions. The scan range was limited from 400 to 1700 $m/z$. Peptide fragmentation was performed using HCD with the energy set at 26 NCE. Intensity threshold for ions selection was set at $1 \times 10^5$ ions with charge exclusion of $z = 1$ and $z > 7$. The MS/MS spectra were acquired at a resolution of 15000 (at $m/z$ 400). Isolation window was set at 1.6 Th. Dynamic exclusion was employed within 30 s.

Data were searched using MaxQuant (version 1.6.6.0) [1,2] using the Andromeda search engine [3] against a reference proteome of Mus musculus (53,449 entries, downloaded from Uniprot the 24th of July 2018). A modified sequence of the protein MTP18 with a Flag tag in its N-ter part was also searched.

The following search parameters were applied: carbamido-methylation of cysteines was set as a fixed modification, oxidation of methionine and protein N-terminal acetylation were set as variable modifications. The mass tolerances in MS and MS/MS were set to 5 ppm and 20 ppm respectively. Maximum peptide charge was set to 7 and 5 amino acids were required as minimum peptide length. A false discovery rate of 1% was set up for both protein and peptide levels. The iBAQ intensity was used to estimate the protein abundance within a sample. The match between runs features was allowed for biological replicate only.

For this Affinity Purification Mass Spectrometry analysis part, the mass spectrometry data have been deposited at the ProteomeXchange Consortium (http://www.proteomexchange.org) via the PRIDE partner repository[103,104] with the dataset identifier PXD028529.

Quantitative analysis was based on pairwise comparison of intensities. Values were log-transformed (log2). Reverse hits and potential contaminant were removed from the analysis. Proteins with at least 2 peptides (including one unique peptide) were kept for further statistics. Intensities values were normalized by median centering within conditions (normalizeD function of the R package DAPAR). Remaining proteins without any iBAQ value in one of both conditions have been considered as proteins quantitatively present in a condition and absent in the other. They have therefore been set aside and considered as differentially abundant proteins. Next, missing values were imputed using the impute.MLE function of the R package imp4. Statistical testing was conducted using a limma t-test thanks to the R package limma[113]. An adaptive Benjamini-Hochberg procedure was applied on the resulting p-values thanks to the function adjust.p of R package cp4p[114] using the robust method previously described[115] to estimate the proportion of true null hypotheses among the set of statistical tests. The proteins associated to an adjusted p-value inferior to a FDR level of 1% have been considered as significantly differentially abundant proteins.

## Statistical analyses

Experiments were repeated at least three times and quantitative analyses were conducted blindly. Randomization of groups (e.g., different genotypes) was performed when simultaneous, parallel measurements were not performed (e.g., Oroboros, CM isolation). For high-throughput measurements (e.g., mitochondrial morphology, cell death), all groups were measured in parallel to reduce experimental bias. Statistical analyses were performed using Graph-Pad Prism v9 software. Data are presented as mean ± SD or SEM where indicated. The statistical tests used, and value of experiment replicates are described in the figure legends. Comparisons between two groups were performed by unpaired two-tailed T-test. To compare more than two groups or groups at multiple time points 1-way ANOVA or 2-way ANOVA was applied. Tests were considered significant at $p$-value $< 0.05$ (*$p < 0.05$; **$p < 0.01$; ***$p < 0.001$; ****$p < 0.0001$).

## Reporting summary

Further information on research design is available in the Nature Research Reporting Summary linked to this article.

# Data availability

Source data for all experiments are available alongside this manuscript. The datasets generated during the current study are available in the European Nucleotide Archive (NEO) repository and Proteomics Identification Database (PRIDE). Accession numbers and the web links are as follows. Bulk RNAseq: (ENA: Project: PRJEB47968), Cardiac proteome of mice at pre-symptomatic (PXD034477) and symptomatic (PXD028516) ages, and cardiac interactome data (PXD028529).The total datasets of fluorescence microscopy images generated and analyzed during the current study are not publicly available due to the incompatibility of exporting comprehensible file names linked to cell, treatment, and time identifiers from the Harmony 4.9, Perkin Elmer software but are available from the corresponding author upon request. Source data are provided with this paper.

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

## Acknowledgements

We thank Pierre-Henri Commere and Sandrine Schmutz for flow cytometry services at the Institut Pasteur and Anastasia Gazi for electron microscopy services. We thank Corinne Lesaffre at the Paris Cardiovascular Research Center for the histology services, Anu Susan Kurian and Priscilla Lopes for technical assistance. We thank Sylvie Fabrega of the Viral Vector for Gene Transfer core facility of Structure Fédérative de Recherche Necker, Université de Paris for lentiviral particle synthesis and Nils-Göran Larsson for providing mitoYFP mice. We thank Arnaud Mourier for inciteful discussions on bioenergetics and Marie Lemesle for excellent administrative assistance. T.W. is supported by the European Research Council (ERC) Starting Grant No. 714472 (Acronym "*Mitomorphosis*"), Fondation pour la Recherche Medicale (MND202003011475), and the Agence Nationale pour la Recherche (ANR-20-CE14-0039-02). C. M. is supported by the German Research Foundation (DFG; SFB 894, TRR-219; Ma 2528/7-1) and the German Federal Ministry of Education and Research (BMBF; 01EO1504).

## Author contributions

Conceptualization: E.D., T.W.; Methodology: E.D., T.W., E.K., T.C., Q.G.G., M.K., C.M. Investigation; E.D., E.V., M.K., T.W., E.K., T.C., Q.G.G., M.K., M.M.-N., Visualization; E.D., T.W., E.K., Q.G.G., M.K., Funding acquisition: T.W., C.M.; Supervision: E.D., T.W., C.M., Project administration; T.W., M.M., C.M. Supervision; E.D., T.W. Writing – original draft: E.D. and T.W.; Writing- review and editing: all authors.

## Competing interests

The authors declare no competing interests.
