## [Peer Review File · Nature Communications]

Mitochondrial Fission Process 1 controls inner membrane integrity and protects against heart failureREVIEWER COMMENTS

Reviewer #1 (Remarks to the Author):

The study investigated the role of mitochondrial fission process 1 (MTFP1) on cardiac structure and function. MTFP1 deletion in the heart resulted in adult-onset dilated cardiomyopathy (DCM), reduced membrane potential, and increased non-phosphorylation-dependent respiration. MTFP1 deletion also increased the sensitivity to programmed cell death, which was accompanied by an opening of the mitochondrial permeability transition pore (mPTP) in vitro. Thus, the authors conclude that MTFP1 influences mitochondrial coupling and cell death sensitivity.

Major concerns:

- 1- Overall, I believe this study is descriptive and does not provide any mechanistic insights delineating the impact of MTFP1 on cardiac energy metabolism and cell death. I do not see any data that defines how MTFP1 alters either energy metabolism. Insights are also not provided as to how MTFP1 may be altering cell death.
- 2- It is not clear to me how MTFP1 influences bioenergetic efficiency, although the authors suggest that bioenergetic efficiency may change.
- 3- The authors do not clearly delineate how MTFP1 influences the activity of the ATP5 complex.
- 4- Figure S4 should report the baseline measurements of LV function and structure pre-doxorubicin treatment to ensure no significant difference in these parameters occurred prior to the treatment protocol.
- 5- The authors do not provide any insights into how MTFP1 modified PTP activity.

Reviewer #2 (Remarks to the Author):

In this manuscript, Authors characterize the cardiac-specific knock-out of mitochondrial fission process 1 (MTFP1), a protein considered to be a player in the mitochondrial fission process. They investigate heart structure and function at both the early and late stages of dilated cardiomyopathy (DCM) and show that transgenic mice lacking MTFP1 develop cardiac damage prematurely. Authors report that the observed higher propensity for cell death depends on increased mitochondrial uncoupling mediated by ANT, and by sensitization of the permeability transition pore (PTP) to opening. Accordingly, cell death occurrence upon bona fide PTP stimulation (Doxorubicin and H₂O₂) is more pronounced in MTFP1 KO cardiomyocytes while it can be prevented by CsA or CyPD ablation in KO MEFs. This is a well developed and carefully written study containing an impressive amount of work and I would like to compliment the Authors for their excellent manuscript. The results challenge current views about the role of MTFP1 in mitochondrial fission and provide evidence for a new role of the protein.

Specific points

1. Authors tested CATR after ADP (alone or after oligomycin addition) and found that it readily normalizes KO to WT JO₂ values, suggesting that the higher JO₂ in KO mitochondria is caused by ANT-mediated uncoupling (Figure 2H-I). It is important to assess also the effect of CATR BEFORE the addition of ADP, given that ADP may contribute to decrease H⁺ conductance on its own (we predict that the effect of CATR may be larger). Given that fatty acids are essential for the uncoupling activity of ANT, it seems also important to test the effect of e.g. palmitate and then CATR on basal respiration (state 2). Fatty acids are particularly important in cardiomyocyte metabolism and these experiments could significantly strengthen involvement of ANT.
2. Figure 4G-I, mitochondrial swelling is a useful method to evaluate PTP opening but several precautions are needed to interpret the results. To compare the propensity of PTP opening in

mitochondria of different genotypes a careful calibration is required, and alamethicin should be added at the end of each experiment (to test for the maximal possible absorbance change, which in turn allows to determine the fraction of mitochondria that have undergone the swelling process). Authors are normalizing the absorbance to the initial value, and they cannot tell if mitochondria are already (partially) swollen. For detailed protocols you may want to read Carraro and Bernardi (2020) Measurement of membrane permeability and the mitochondrial permeability transition, *Methods Cell Biol.* 155, 369-379. This calibration appears to be particularly critical for the experiment with mitochondria from MEFS (Figure 4I) where swelling is apparently marginal.

3. The Authors tested a single Ca²⁺ load, while a titration with Ca²⁺ could be extremely useful to define differences in Ca²⁺ sensitivity.

4. Page 3, line 58 and page 11, line 80, please specify that ANT catalyzes ADP uptake and ATP release in energized mitochondria.

5. Page 9, line 37 please remove "Active"; lines 39-40, as written rotenone is presented as a substrate, please say "succinate in the presence of rotenone".

6. Page 10, line 57, in spite of its name (oligomycin sensitivity conferral protein), oligomycin does not bind OSCP. The binding site is actually located between two adjacent c-subunits in contact with the proton half-channel formed by subunit a [Symersky et al. (2012) Oligomycin Frames a Common Drug-Binding Site in the ATP Synthase. *Proc. Natl. Acad. Sci. USA* 109, 13961–13965].

7. Page 13, line 64, the study of Halestrap and Davidson (1990) reported the presence of PPIase activity in mitochondria but did not identify the species involved. Mitochondrial cyclophilin was isolated only in 1996 [Nicolli et al. (1996) Interactions of Cyclophilin with the Mitochondrial Inner Membrane and Regulation of the Permeability Transition Pore, a Cyclosporin A-sensitive Channel. *J. Biol. Chem.* 271, 2185-2192].

8. Page 14, line 81, after "DOXO" please add due reference "which is known to promote onset of the permeability transition [Solem et al. (1994) Disruption of mitochondrial calcium homeostasis following chronic doxorubicin administration, *Toxicol. Appl. Pharmacol.* 129, 241-222]".

9. Page 18, line 96 I am not sure that the sentence is correct, shouldn't it read "mPTP opening and decreased cell death" rather than accelerated.

10. Page 19, line 31, a reference is missing to the Rabinovitch lab.

Paolo Bernardi and Michela Carraro

Reviewer #3 (Remarks to the Author):

The manuscript entitled "Loss of Mitochondrial Fission Process 1 (MTFP1) promotes mitochondrial uncoupling, cell death and heart failure in mice" by Donnarumma et al. investigate the role of MTFP1 in mice. MTFP1 is previously described as a protein involved in mitochondrial inner membrane fission. In this manuscript, a novel role for MTFP1 in mitochondrial uncoupling and increased sensitivity to programmed cell death is described. Cardiomyocyte specific MTFP1 knockout (cMKO) resulted in adult-onset dilated cardiomyopathy characterized by inflammation and cardiac fibrosis that progressed to heart failure and caused middle-age death in mice. In their attempt to find a mechanism, the authors demonstrated that MTFP1 promotes the ANT-dependent uncoupling of mitochondrial respiration and opening of the mPTP. They also suggest that the mPTP opening sensitizes the cells to programmed cell death. Overall, it is a well-thought-out paper; however, the following points need to be evaluated further for a novel role for MTFP1:

1) The role of MTFP1 in mitochondrial biogenesis is assessed using multiple tools from respiratory to

transcriptomics and proteomics studies. In cMKO mice approximately 30% difference observed in O₂ consumption at the onset and heart failure (Figs. 2B and C) as well as changes in expression levels of mitochondrial genes and proteins (dataset EV2 and EV1). However, the authors completely ignored their experimental findings, specifically at the proteome level, and suggested that the reduced mitochondrial respiration was not due to reduced levels of mitochondrial proteins (Page 8 line 18 (the number was cut)).

a. In addition to the changes observed in label-free quantitation of many proteins related to mitochondrial biogenesis, there is at least 30% reduction in ATP synthase and its doublet (marked as V and V2) in BN-PAGE results detected with ATP5A antibody (Fig. 2F) in 8–10-week-old cMKO mice. Although the proteomics studies performed using 18-week-old mice, log₂ value for KO/WT is -0.73679353 for ATP5E which corresponds to 40% decrease in one of the essential components of the ATP synthase (dataset EV1).

b. IMMT, also called mitofilin, is MICOS complex subunit Mic60 and it is one of the major components of inner mitochondrial membrane organizing cristae junction formation and distribution. It is detected by 76 peptides and its log₂(Mean.in.KO/Mean.in.WT) value is -1.753952004 and the KO/WT ratio is 0.30. Three-fold reduction in this protein would affect mitochondrial respiration and apoptosis described in this paper.

The list can continue like this, as mentioned above authors need to evaluate their data more carefully before proposing a new role for MTFP1 in ANT-dependent uncoupling of inner membrane and proton leakage.

2) It would be nice to compare complex I, II, and IV oxygen consumption rates in pre-symptomatic as shown for 18- and 34-week-old KO and WT mice for direct comparison. Steady-state expression of oxidative phosphorylation complexes should also be compared for these mice prior to make such a significant claim in addition to the respiratory studies.

3) It would be interesting to see steady-state expression of ANT1 against other mitochondrial relevant loading markers rather than vinculin (Fig 2G). Also, ANT1 is one the most abundant IM proteins in mitochondria; interestingly, only ANT3 was detected in the proteomics analysis (dataset EV1).

4) The difference, if there is any, between Figs 2D, 2H, and 2I needs to be clarified.

5) Studies performed for programmed cell death and opening of mPTP using primary cardiomyocytes and cell lines are complementary to what proposed in the manuscript.

6) The difference between the median lifespan of the male and female mice is also interesting. The authors claimed that the MTFP1 protects against proton leak across the inner membrane in ANT-dependent manner and its loss causes the age dependent decline in cardiac energy metabolism in ANT-dependent manner in mice. If the mechanism proposed in this article universal and proved in multiple model systems, why the lifespan between the male and female mice is so different? Figure 6 and the title also refers to a general mechanism in mice. This phenomenon needs to be addressed more carefully before generalizing their findings.

Reviewer #4 (Remarks to the Author):

In this manuscript, Erminia DONNARUMMA et al described the new roles of MTFP1, as a critical regulator of mitochondrial coupling in cardiac homeostasis, which are distinct from its previous role in mitochondrial fission. MTFP1 deletion in post-natal cardiomyocytes leads to membrane potential dissipation associated to mPTP opening, cell death, and a progressive DCM developing into heart failure and middle-aged death. The amount and quality of data is enough to insist their claim describing in this manuscript. The topic of MTFP1 as a valuable tool for the molecular dissection of mitochondrial uncoupling and mPTP function has emerged as a novel topic in heart diseases, and I believe that this paper will contribute to this new research direction. Here, I have a few concerns need

to be addressed prior to next consideration.

1. Authors showed no changes in basal or maximal respiration rates nor any evidence of mitochondrial uncoupling in cultured MTFP1-deficient MEFs (Mtfp1^{-/-}) and U2OS cells (Mtfp1^{-/-}). Whether is MTFP1-dependent proton leak also found in cultured cardiomyocyte cells or not, such as rat cardiomyocyte cells (H9c2) and human cardiomyocyte-like cells from fibroblasts (HCFs)? As authors suggested that MTFP1-dependent proton leak may be cell type specific, the data showing in other myocardial cells of different species is required to strengthen their claims.
2. In Figure 3, the authors report that no indications of mitochondrial elongation (median mitochondrial surface area: WT 3198 μm^2 versus cMKO 2954 μm^2) nor altered cristae organization (Figure 3C-D). But there is a statistically significant difference in the violin plot of mitochondrial surface area within cardiac posterior wall measured in WT and cMKO mitochondria, as $**p < 0.01$ Mann-Whitney test. Please clarify it.
3. The confocal images of Mtfp1^{-/-} MEFs treated with Mtfp1 siRNA was missing in Figure 3I.
4. In Figure 4, the authors showed that MTFP1 was essential for cell survival using the adult cardiomyocytes from 8-10-week-old WT and cMKO mice. How about the cell sensitivity under the treatment of CCCP or H₂O₂ in cardiomyocytes isolated from juvenile mice?
5. I didn't find the description of Figure 6 in the manuscript, where is it?
6. There are several typos or formatting issues throughout the manuscript. Authors should go through their manuscript carefully.

Donnarumma et al.

Loss of Mitochondrial Fission Process 1 (MTFP1) promotes mitochondrial uncoupling, cell death and heart failure in mice

NCOMMS-21-50617-T

Reviewer #1

The study investigated the role of mitochondrial fission process 1 (MTFP1) on cardiac structure and function. MTFP1 deletion in the heart resulted in adult-onset dilated cardiomyopathy (DCM), reduced membrane potential, and increased non-phosphorylation-dependent respiration. MTFP1 deletion also increased the sensitivity to programmed cell death, which was accompanied by an opening of the mitochondrial permeability transition pore (mPTP) in vitro. Thus, the authors conclude that MTFP1 influences mitochondrial coupling and cell death sensitivity.

Major concerns:

1- Overall, I believe this study is descriptive and does not provide any mechanistic insights delineating the impact of MTFP1 on cardiac energy metabolism and cell death. I do not see any data that defines how MTFP1 alters either energy metabolism. Insights are also not provided as to how MTFP1 may be altering cell death.

We respectfully disagree with the opinion of Reviewer #1 on this point. We believe our study has advanced the state-of-the art by providing significant molecular mechanistic insights into the importance of inner membrane integrity controlled by MTFP1 including both mitochondrial respiration efficiency and programmed cell death, while also formally excluding an essential role in mitochondrial fission.

First, our data clearly demonstrate that MTFP1 protects against proton leak across the IMM in ANT-dependent manner, which is of paramount importance to bioenergetic efficiency. ANT-dependent uncoupling has been recently shown to be the underlying cause of age-dependent decline of cardiac energy metabolism, function and lifespan in mice (Chavez et al., 2020; Chiao et al., 2020; Zhang et al., 2020) and our study reveals this mechanism to be at play prior to the onset of dilated cardiomyopathy. In our study, we perform several high-resolution respirometry and fluorometry measurements (Oroboros) to simultaneously measure oxygen consumption rates and membrane potential at states 2 (pyruvate/malate/glutamate or succinate/rotenone or malate/palmitoylcarnitine), 3 (ADP), and 4 (oligomycin). These data (now Fig. 2D-K, Fig. S2E-H) allowed us to identify specific defects in state 4, but not state 3, revealing a potential proton leak across the IMM of MTFP1 mitochondria in pre-symptomatic mice. We were able to demonstrate this leak, as evidenced by reduced mitochondrial membrane potential and elevated respiration in the presence of oligomycin, could be rescued by addition or substitution of carboxyatractyloside (CATR), a specific inhibitor of ANT (Bertholet et al., 2019, 2022), but not inhibition of uncoupling proteins (UCPs) such as UCP1 (Fig. S2S). **We have now added additional data** demonstrating that proton leak can also be rescued by adding CATR *before* the addition of ADP (Fig. 2J-K), which lends additional credence to our report of ANT-dependent proton leak caused by MTFP1 ablation. We show this regulation cannot be explained by altered levels of ANT1 itself, which is the major cardiac isoform of this channel, and we have now corroborated our existing (Fig. 2G) and new western blot findings by **new MS proteomic experiments**, in which we do not observe ANT1 (encoded by *Slc25a4*) or even ANT2 (encoded by *Slc25a5*) to be statistically significantly altered (now Fig. S2A, EV Dataset 2).

Second, our studies have revealed MTFP1 ablation to increase mPTP-dependent cell death. We report markers of cell death (increased serum levels of troponin, Fig. 1N), and activation of this process *ex vivo* (in primary adult cardiomyocytes, Fig. 4A-F) and in cellulo (MEFs, Fig. 4I-N, Fig. S4D-I). This effect is mediated by an increased susceptibility to mPTP

pore to opening in mitochondria isolated from cMKO hearts (Fig. 4G, S4A) and cultured cells (Fig. 4H, S4B), and we can prove this by genetic (*Ppif* deletion) or pharmacological (Cyclosporine A) rescue experiments that inhibit mPTP via Cyclophilin D (Fig. 4G-H, Fig. 5A-J). We have also **included new data in HL-1 cells**, an established mouse cardiomyocyte cell, demonstrating that here too, MTFP1 ablation increases programmed cell death sensitivity (Fig. S5A-D). HL-1 cells devoid of MTFP1 show increased cell death sensitivity without altered bioenergetic profiles (Fig. S5E-J), like MEFs (Fig. S2I-N) and U2OS cells (Fig. S2J-R), suggesting that these phenotypes can in fact be functionally separated.

Finally, regarding the insights into the regulation of cell death we demonstrate that mPTP pore opening in cardiac mitochondria is enhanced in the absence of MTFP1 in pre-symptomatic mice (Fig. 4G, S4A). We have **performed new 2D-BN-PAGE studies** to identify native macromolecular complexes and observe that MTFP1 forms a complex in the inner membrane that co-migrates with other factors that have been demonstrated to regulate the mPTP, such as ANT1 and Cyclophilin D (encoded by *Ppif*), but not with ATP5A (Fig. 6D). In the context of our previous MTFP1 cardiac interactome studies reported in now Fig. 6B,C our data suggest that MTFP1 physically interacts with mPTP regulators to influence the propensity of permeability transition and therefore programmed cell death. We also performed 2D-BN-PAGE studies using cMKO mitochondria isolated from pre-symptomatic mice, which revealed no alteration in complexes containing ANT1, Cyclophilin D, nor ATP5A (Fig. 6D), indicating MTFP1-dependent regulation of mPTP opening does not occur via the assembly or maintenance of the mPTP, but more likely via modulations in the conformation and/or activation.

2- It is not clear to me how MTFP1 influences bioenergetic efficiency, although the authors suggest that bioenergetic efficiency may change.

We take this comment under advisement and have now clarified this in the text. Our bioenergetic studies performed by high-resolution fluo-respirometry using cardiac mitochondria isolated from pre-symptomatic cMKO mice allowed us to simultaneously measure the oxygen consumption rates and the mitochondrial membrane potential under State 2, State 3, and State 4 conditions (Fig. 2). We consistently observed reduced respiratory control ratios (e.g. RCR State3/State4) and lower membrane potential in cardiac mitochondria from cMKO mice (Fig. 2D, S2F), which are a consequence of increased proton leak across the IMM and not because of altered phosphorylation-dependent oxygen consumption (State 3), which is clearly unaffected under all substrate conditions (Fig. 2D,2H, S2F, S2H). We then show that ANT, but not UCPs (e.g. UCP1), is responsible of this futile leak as inhibition with a highly specific and well-defined inhibitor carboxyatractyloside (CATR) can rescue proton leak, thus normalizing RCRs and membrane potential (Fig. 2D-E, 2H-I). These data are also consistent with recent electrophysiological studies in cardiac mitoplasts that show that ANT but not UCP1 is capable of proton leak across the IMM (Bertholet et al., 2019). We have **now added new data** (Fig. 2J-K) that demonstrate that CATR can rescue proton leak before as well as after the addition of ADP (state 3) and have amended the manuscript on line 287 to read *“Independently of Omy, CATR treatment rescued proton leak when added after fueling state 3 respiration with ADP (Figure 2H), normalizing the respiratory control ratio for state 3:4 (Figure 2I). Moreover, the addition of CATR before energizing mitochondria with PGM (Figure 2J) or malate/PC (Figure 2K) was also able to normalize the elevated JO₂ and the decreased membrane potential of cMKO mitochondria to WT levels. Taken together, our data strongly indicate that Mtfp1 deletion increases ANT-dependent proton leak”*. In addition, **we have also performed new proteomic analyses of cardiac tissue** from pre-symptomatic mice (Fig. S2A, Dataset EV2) at a stage when uncoupling was observed, which revealed no general reductions in mitochondrial proteins of the OXPHOS system. Together, our existing and new data show that MTFP1 loss does not impair mitochondrial mass, mitochondrial morphology, cristae morphology, mtDNA content (Fig. 3A-E), the levels of the OXPHOS proteins (Fig. S2B) or OXPHOS complex assembly (Fig. 2F, S2A, C-D) by cardiac proteomics (Fig. S2A, Dataset EV2), which is important since

defects in any one of these could have led to impaired oxygen consumption rates in pre-symptomatic mice. We have **performed new 2D BN-PAGE analyses** from cardiac mitochondria, which show for the first time that ANT1, the major isoform of the adenine nucleotide translocase, and MTFP1 co-migrate in a macromolecular complex (Fig. 6D). Together with the cardiac interactome data indicating a physical interaction between ANT and MTFP1 (Fig. 6A-C), these data enable us to propose that physical interactions between MTFP1 and ANT may affect coupling efficiency of cardiac mitochondria.

3- The authors do not clearly delineate how MTFP1 influences the activity of the ATP5 complex.

As mentioned above, it is the coupling efficiency but not the activity of the ATP synthase complex (also referred to as Complex V) that is affected by the loss of MTFP1. The bioenergetic studies performed on cardiac mitochondria of pre-symptomatic cMKO mice indicate that MTFP1 loss does not influence complex V activity. In these experiments, we assessed oxygen consumption rates and mitochondrial membrane potential under complex I or complex II driven respiration (State 2), ADP- phosphorylating (State 3), and non-phosphorylating conditions (State 4) by high-resolution fluo-respirometry. We consistently observed with either pyruvate, glutamate, and malate (Fig. 2D-E) or succinate and rotenone (Fig. S2E,F) that oxygen consumption rates and membrane potential were similar under State 3 conditions, suggesting that ADP is maximally and equally phosphorylated to ATP by complex V in WT and cMKO mitochondria. If there were increased or decreased activity of complex V, first principles would predict an alteration of state 3 respiration and membrane potential, neither of which are observed. In addition, these data are supported by **new experiments we have performed in which 1D BN-PAGE analyses** of Complex V (Fig. S2C), not previously included in the initial version of the manuscript, that clearly show no significant changes in the steady levels of Complex V monomer and dimers upon MTFP1 loss in cardiac mitochondria in pre-symptomatic mice. We also do not observe Complex V assembly intermediates, which have previously been shown to modulate Complex V activity in the murine heart (Mourier et al., 2014). Consistent with BN-PAGE analysis, we do not observe reductions in the steady state levels of ATP5A protein levels by SDS-PAGE of cMKO cardiac tissue (Fig. S2B) nor by MS-based proteomic analyses of cardiac tissue (Fig. S2A). These data allow us to conclude that the coupling efficiency but not the ATPase activity of Complex V is responsible for reduced bioenergetic efficiency in pre-symptomatic mice.

However, our **new 1D-BN-PAGE** of cardiac mitochondria isolated from mice suffering from DCM do show a mild yet statistically significant reduction in Complex V (Fig. S2C), which is consistent with both the existing bioenergetic profiles originally reported in the manuscript (Fig. 2B-C) as well as the **new data we have generated by transmission electron microscopy** on hearts from symptomatic cMKO mice that show mitochondrial swelling and cristae disorganization (Fig. S3D-E).

4- Figure S4 should report the baseline measurements of LV function and structure pre-doxorubicin treatment to ensure no significant difference in these parameters occurred prior to the treatment protocol.

We thank the reviewer for pointing this out. We have measured cardiac structure and function by echocardiography at 10 weeks of age in cMKO mice (n=18) vs WT littermates (n=13) and observed no significant differences in %LVEF, LVSD, LVDD, which is the reason we selected 8 weeks of age for the Doxorubicin injection in WT and cMKO mice and because the 2 weeks treatment time would take us to 10 weeks and allowed us to compare endpoints (Fig. S4J-N). As reported in Fig. S4G-J, doxorubicin injection negatively impacts LV function and structure of cMKO mice at 10 week of age (after 2 weeks post-doxo) as evidenced by reduced % LVEF and increased LVSD and LVDD. Despite a much smaller, less powerful sample size, this effect

was still observed. This degree of dysfunction was never spontaneously observed in the absence of doxorubicin in WT mice nor in cMKO mice before the age of 18 weeks. The increased sensitivity to doxorubicin treatment observed in vivo is consistent with the increased cell death triggered by doxorubicin treatment in primary cardiomyocytes and MEFs lacking MTFP1. Therefore, we believe this allows us to conclude that the LV function and structure defects are the result of the doxorubicin treatment.

5- The authors do not provide any insights into how MTFP1 modified PTP activity.

We thank the reviewer for this comment and we fully agree that understanding how MTFP1 modifies mPTP activity is important. As the reviewer may appreciate, this is complicated by the fact that there is no clear consensus on the molecular composition of the mPTP (Bonora et al., 2021). Nevertheless, we provide several insights not the least of which is the physical interaction of MTFP1 (by co-IP in cardiac mitochondria) with proteins that have been previously shown to regulate the mPTP. These data are shown in Fig. 6B-C. Indeed, some of these proteins have been described to co-migrate with MTFP1 in complexesome profiling experiments (BN-PAGE coupled with MS proteomics of excised slices) in rat mitochondria (Heide et al., 2012), suggesting that these proteins may form a bona fide complex.

We have now **added new 2D-BN-PAGE data** obtained from cardiac mitochondria isolated from pre-symptomatic WT and cMKO mice not previously included in the manuscript showing that MTFP1 co-migrates in a macromolecular complex containing PPIF and ANT1 (Fig. 6D). Our results indicate that migration of this PPIF/ANT1/MTFP1 complex is not altered upon MTFP1 loss in cMKO mice. This leaves us with the most parsimonious explanation that MTFP1 modulates mPTP activity not by regulating its gross assembly or maintenance but rather via altering substrate accessibility and/or its activation.

Reviewer #2

In this manuscript, Authors characterize the cardiac-specific knock-out of mitochondrial fission process 1 (MTFP1), a protein considered to be a player in the mitochondrial fission process. They investigate heart structure and function at both the early and late stages of dilated cardiomyopathy (DCM) and show that transgenic mice lacking MTFP1 develop cardiac damage prematurely. Authors report that the observed higher propensity for cell death depends on increased mitochondrial uncoupling mediated by ANT, and by sensitization of the permeability transition pore (PTP) to opening. Accordingly, cell death occurrence upon bona fide PTP stimulation (Doxorubicin and H₂O₂) is more pronounced in MTFP1 KO cardiomyocytes while it can be prevented by CsA or CyPD ablation in KO MEFs. This is a well-developed and carefully written study containing an impressive amount of work and I would like to compliment the Authors for their excellent manuscript. The results challenge current views about the role of MTFP1 in mitochondrial fission and provide evidence for a new role of the protein.

Specific points

1. Authors tested CATR after ADP (alone or after oligomycin addition) and found that it readily normalizes KO to WT JO₂ values, suggesting that the higher JO₂ in KO mitochondria is caused by ANT-mediated uncoupling (Figure 2H-I). It is important to assess also the effect of CATR BEFORE the addition of ADP, given that ADP may contribute to decrease H⁺ conductance on its own (we predict that the effect of CATR may be larger).

We thank Paolo Bernardi and Michela Carraro for this important control experiment. We have **performed new experiments** in isolated mitochondria from pre-symptomatic mice (Fig. 2J) and observed that inhibition of ANT with CATR before energizing mitochondria with PGM and ADP is able to rescue the reduced mitochondrial membrane potential observed in cMKO

mitochondria. Moreover, CATR addition before fueling mitochondria with substrates (PGM) significantly reduced state 2 respiration in both WT and cMKO mitochondria normalizing the elevated respiration observed in cMKO to WT level. These data further strengthen the involvement of ANT in proton leak and mitochondrial uncoupling in cMKO mitochondria and we are grateful for this important suggestion. We have amended the text to read on line 287: *“Independently of Omy, CATR treatment rescued proton leak when added after fueling state 3 respiration with ADP (Figure 2H), normalizing the respiratory control ratio for state 3:4 (Figure 2I). Moreover, the addition of CATR before energizing mitochondria with PGM (Figure 2J) or malate/PC (Figure 2K) was also able to normalize the elevated JO_2 and the decreased membrane potential of cMKO mitochondria to WT levels. Taken together, our data strongly indicate that Mtfp1 deletion increases ANT-dependent proton leak”.*

Given that fatty acids are essential for the uncoupling activity of ANT, it seems also important to test the effect of e.g. palmitate and then CATR on basal respiration (state 2). Fatty acids are particularly important in cardiomyocyte metabolism and these experiments could significantly strengthen involvement of ANT.

Indeed, recent (Bertholet et al., 2019, 2022) and less recent studies (Andreyev AYu et al., 1989; Brustovetsky et al., 1990; Shabalina et al., 2006) have demonstrated that exogenous addition of fatty acids can enhance ANT-dependent proton leak. However, we observe CATR-sensitive ANT-dependent proton leak in intact isolated cardiac mitochondria without the addition of exogenous fatty acids such as palmitate (Fig. 2D-E, 2H-I, S2F-G), indicating that endogenous fatty acids are sufficient to promote ANT-dependent proton leak in the heart. This is still experimentally consistent with the discovery made by Bertholet and colleagues that all tissues that lack UCP1, which includes the mouse heart, are capable of ANT-dependent proton leak in a fatty-acid dependent manner (Bertholet et al., 2019), since these elegant electrophysiological studies were performed on mitoplasts and not intact mitochondria. It is also worth noting that the addition of fatty-acid free BSA in our mitochondrial isolation protocol has been critically established to ensure mitochondria are not uncoupled during the isolation process.

Nevertheless, we have attempted to conjugate BSA to palmitate with the aim of supplying this to isolated cardiac mitochondria. Despite multiple attempts, our BSA-palmitate substrate failed to increase respiration or lower membrane potential, which could be explained either by a defect in the conjugation process or by the lack of an effect of palmitate. Since production shortages/supply chain issues precluded us from purchasing BSA-palmitate from usual commercial sources, we elected to use palmitoylcarnitine instead. We observed that CATR addition before energizing both WT and cMKO mitochondria with malate followed by palmitoyl-carnitine (PC) rescued the reduced mitochondrial membrane potential and the elevated state 2 respiration sustained by PC observed in cMKO mitochondria (Fig. 2K). These data clearly suggest that fueling mitochondria with malate/PC substrates promotes proton leak and mitochondrial uncoupling which can be rescued by specifically inhibiting ANT with CATR.

2. Figure 4G-I, mitochondrial swelling is a useful method to evaluate PTP opening but several precautions are needed to interpret the results. To compare the propensity of PTP opening in mitochondria of different genotypes a careful calibration is required, and alamethicin should be added at the end of each experiment (to test for the maximal possible absorbance change, which in turn allows to determine the fraction of mitochondria that have undergone the swelling process). Authors are normalizing the absorbance to the initial value, and they cannot tell if mitochondria are already (partially) swollen. For detailed protocols you may want to read Carraro and Bernardi (2020) Measurement of membrane permeability and the mitochondrial permeability transition, Methods Cell Biol. 155, 369-379. This calibration appears to be particularly critical for the experiment with mitochondria from MEFS (Figure 4I) where swelling is apparently marginal.

We appreciate the keen attention to detail and fully agree that the addition of alamethicin is certainly an optimal control to assess the total fraction of mitochondria that undergo to swelling, mainly when mitochondria of different genotype have different degree of swelling “per sé” or due to the technical preparation. However, in our experiments we do not observe any difference in the absorbance values of WT and MTFP1 deficient mitochondria *prior* to the addition of Ca^{2+} , suggesting that the mitochondrial state or mitochondrial swelling at baseline condition is similar between WT and MTFP1 deficient mitochondria (please see graphs below). Therefore, the absorbance changes observed in WT and cMKO mitochondria upon Ca^{2+} stimulation are specific to this swelling trigger and reflects the degree of swelling induced by Ca^{2+} in the different genotypes. In other words, the % of swelling reported in the graphs reflects indeed the total fraction of mitochondria undergoing Ca^{2+} -induced swelling, which is higher in MTFP1 deficient mitochondria isolated from both cardiac tissue (pre-symptomatic mice) as well as from MEFs. Moreover, the modest swelling of mitochondria isolated from MEFs has been previously reported and it is consistent with the low mitochondrial metabolism of these cells (Karch et al., 2013).

Nevertheless, to experimentally address the concern that was raised, we have performed a new mPTP swelling assay on mitochondria isolated from WT and *Mtfp1*^{-/-} MEFs where alamethicin (Ala, 5 μM) was used as control to induce the maximal mitochondrial swelling (please see figure below). First, we observed no difference in the absorbance values of WT and *Mtfp1*^{-/-} mitochondria prior to the addition of Ca^{2+} suggesting that mitochondrial state was similar between the two genotypes. Second, as expected and previously demonstrated, Ca^{2+} stimulation induced higher absorbance value changes in *Mtfp1*^{-/-} mitochondria compared to WT (shown in Fig 4G, S4B). Finally, the addition of Ala on the maximal absorbance changes induced by the Ca^{2+} pulse further increased mitochondrial swelling in both WT and cMKO. The degree of Ala-induced swelling was similar between WT and *Mtfp1*^{-/-} mitochondria, as shown by the $\Delta\text{Abs}_{\text{Abs}} - \text{Ca}^{2+} - \text{Abs}_{\text{Abs}}$. These data allows us to conclude that the *Mtfp1*^{-/-} mitochondria are more sensitive to Ca^{2+} -induced mitochondrial swelling.

The Authors tested a single Ca²⁺ load, while a titration with Ca²⁺ could be extremely useful to define differences in Ca²⁺ sensitivity.

We thank Paolo Bernardi and Michela Carraro for this comment and we have performed an additional experiment in which we have tested multiple addition of Ca²⁺ to reach final Ca²⁺ load of 120 μM CaCl₂ as for Fig. S4G, observing a biphasic and higher swelling in cMKO mitochondria, which we have now reported in Fig. S4A.

4. Page 3, line 58 and page 11, line 80, please specify that ANT catalyzes ADP uptake and ATP release in energized mitochondria.

We have amended the text to read on page 3, line 59 “the adenine nucleotide translocase (ANT), which catalyzes ADP uptake and ATP release in energized mitochondria” and “ANT is an integral IMM transporter that catalyzes ADP uptake and ATP release in energized mitochondria” on page 11, line 280

5. Page 9, line 37 please remove “Active”; lines 39-40, as written rotenone is presented as a substrate, please say “succinate in the presence of rotenone”.

We have amended the text accordingly.

6. Page 10, line 57, in spite of its name (oligomycin sensitivity conferral protein), oligomycin does not bind OSCP. The binding site is actually located between two adjacent c-subunits in contact with the proton half-channel formed by subunit a [Symersky et al. (2012) Oligomycin Frames a Common Drug-Binding Site in the ATP Synthase. Proc. Natl. Acad. Sci. USA 109, 13961–13965].

Thank you for this clarification. Given the efforts we have made in our manuscript to distinguish namesake from actual function of MTFP1, we should have applied the same care to other confusing cases such as OSCP. We have amended the text accordingly to read: “In the mouse heart, reduced Omy sensitivity can result from defects in the assembly of the ATP synthase (Mourier et al., 2014) that alter the affinity of Omy binding to Complex V between two adjacent c-subunits in contact with the proton half-channel formed by subunit a(Symersky et al., 2012).

7. Page 13, line 64, the study of Halestrap and Davidson (1990) reported the presence of PPlase activity in mitochondria but did not identify the species involved. Mitochondrial cyclophilin was isolated only in 1996 [Nicolli et al. (1996) Interactions of Cyclophilin with the Mitochondrial Inner Membrane and Regulation of the Permeability Transition Pore, a Cyclosporin A-sensitive Channel. J. Biol. Chem. 271, 2185-2192].

We have amended the reference accordingly.

8. Page 14, line 81, after “DOXO” please add due reference “which is known to promote onset of the permeability transition [Solem et al. (1994) Disruption of mitochondrial calcium homeostasis following chronic doxorubicin administration, Toxicol. Appl. Pharmacol. 129, 241-222]”.

We have amended the text accordingly on line 378 to read: “To test whether MTFP1 protected against PCD induction in vivo, we injected pre-symptomatic (aged 8 weeks) cMKO and WT mice with DOXO, which is known to promote onset of the permeability transition(Solem et al., 1994)”

9. Page 18, line 96 I am not sure that the sentence is correct, shouldn't it read "mPTP opening and decreased cell death" rather than accelerated.

Oops, we missed that one. We have amended the text accordingly.

10. Page 19, line 31, a reference is missing to the Rabinovitch lab.

We have amended the reference accordingly.

Paolo Bernardi and Michela Carraro

Reviewer #3

The manuscript entitled "Loss of Mitochondrial Fission Process 1 (MTP1) promotes mitochondrial uncoupling, cell death and heart failure in mice" by Donnarumma et al. investigate the role of MTFP1 in mice. MTFP1 is previously described as a protein involved in mitochondrial inner membrane fission. In this manuscript, a novel role for MTFP1 in mitochondrial uncoupling and increased sensitivity to programmed cell death is described. Cardiomyocyte specific MTFP1 knockout (cMKO) resulted in adult-onset dilated cardiomyopathy characterized by inflammation and cardiac fibrosis that progressed to heart failure and caused middle-age death in mice. In their attempt to find a mechanism, the authors demonstrated that MTFP1 promotes the ANT-dependent uncoupling of mitochondrial respiration and opening of the mPTP. They also suggest that the mPTP opening sensitizes the cells to programmed cell death. Overall, it is a well-thought-out paper; however, the following points need to be evaluated further for a novel role for MTFP1:

1) *The role of MTFP1 in mitochondrial biogenesis is assessed using multiple tools from respiratory to transcriptomics and proteomics studies. In cMKO mice approximately 30% difference observed in O₂ consumption at the onset and heart failure (Figs. 2B and C) as well as changes in expression levels of mitochondrial genes and proteins (dataset EV2 and EV1). However, the authors completely ignored their experimental findings, specifically at the proteome level, and suggested that the reduced mitochondrial respiration was not due to reduced levels of mitochondrial proteins (Page 8 line 18 (the number was cut)).*

We believe there has some confusion between the observations made in symptomatic mice manifesting cardiomyopathy and pre-symptomatic mice with no observable defects in cardiac function nor structure, and for this we apologize and have clarified the text accordingly.

Indeed, the cardiac mitochondria isolated from cMKO mice during the symptomatic phase (i.e. cardiomyopathy) manifest dramatic reductions in various mitochondrial parameters including O₂ consumption rates (Fig. 2B,C). We fully agree that this decrease could be attributed to changes in the mitochondrial proteome, as well as changes in mitochondrial ultrastructure or cellular content of the heart, which ultimately may arise as a consequence of cardiac dysfunction rather than be the underlying cause. To be clear, MS-based label-free quantification of cardiac tissue identified that only 37 mitochondrial proteins (there are an estimated ~1500 mitochondrial proteins) to be statistically significantly dysregulated in symptomatic cMKO mice (Fig. S2A), of which only 2 proteins beside MTFP1 were found to be reduced: the acyl-coenzyme A (CoA) synthetase ACSM5 and the tRNA-guanine transglycosylase QTRT1. These observations are inconsistent with the notion that whole-sale mitophagy is engaged in these mice. Of course, we cannot formally exclude that either of these proteins, when impaired or reduced, may contribute to reduced cardiac function and heart failure, although there is no evidence in the literature tying either ACSM5 or QTRT1 to cardiac dysfunction to the best of our knowledge.

We also believe that some of the confusion is attributed to the nomenclature used in Dataset EV1: Column M, which was previously entitled “Differential?” has now been renamed “Differentially Expressed?” and the values “1” and “0” have been replaced with “Yes” and “Not significant”. This updated table now more clearly displays the mitochondrial proteins (Column N: MouseMitoCarta) that are statistically significantly dysregulated.

Our focus has been to identify the mitochondrial changes that occur before the onset of cardiomyopathy (i.e. pre-symptomatic mice), which are therefore the likely triggers of disease development. The data obtained from high-resolution fluoro-respirometry studies performed on cardiac mitochondria from pre-symptomatic cMKO mice show no changes in the state 3 O₂ consumption rates (equivalent to what was shown in Fig. 2B and C), excluding this parameter as a trigger for cardiomyopathy. The only clear mitochondrial defects we are able to detect in pre-symptomatic mitochondria are ANT-dependent uncoupling (reduced bioenergetic efficiency) and increase sensitivity to mPTP-opening and cell death.

Since we had defined cardiac proteome from WT and cMKO in the symptomatic phase, we therefore **performed new LFQ MS proteomics on cardiac tissue** from healthy mice in the pre-symptomatic phase corresponding to the age at which we performed the bioenergetic measurements (Fig. S2A, right) in order to obtain a clearly picture of possible proteomic remodeling. These new data reveal no significant decrease in any known mitochondrial protein required for OXPHOS complexes (Fig. S2A, Dataset EV2, see figure below), consistent our initial reports that state 3 respiration rates are not impaired under any substrates provided (Fig. 2D-E, S2F-I). These new data also allow us to address the points raised by the reviewer below.

a. In addition to the changes observed in label-free quantitation of many proteins related to mitochondrial biogenesis, there is at least 30% reduction in ATP synthase and its doublet (marked as V and V2) in BN-PAGE results detected with ATP5A antibody (Fig. 2F) in 8–10-week-old cMKO mice. Although the proteomics studies performed using 18-week-old mice, log2 value for KO/WT is -0.73679353 for ATP5E which corresponds to 40% decrease in one of the essential components of the ATP synthase (dataset EV1).

We thank the reviewer for pointing this subtle change. We have performed **additional 1D BN-PAGE** analyses on pre-symptomatic mice not previously included in the manuscript and observed no significant decrease in the steady state levels of Complex V monomer or dimer species by immunoblot analyses (Fig. S2C see figure below). These results are consistent with our unbiased proteomic analyses of cardiac tissue of pre-symptomatic mice (Fig S2A). We did not observe significant decrease in the steady state levels of any structural subunit or known assembly factor of Complex V arguing in favor of a normal expression of Complex V subunits in cMKO mice at pre-symptomatic phase.

While ATP5E does show a log2 value of 0.73679353, the adjusted pvalue (padj) of 0.021159308 is above the statistical threshold of 0.01 and was therefore classified as not differentially expressed (Column M, previously “Differential?”, now “Differentially Expressed?”). These data are correct and as we have corrected the nomenclature in a revised Dataset EV1 to avoid further confusion. The mathematical justifications to use p-values is obviously the variances of measured intensities along the replicates that is not taken into account if you use only the average fold-change (as the reviewer has done). The mathematical justifications for using adjusted p-values is the well-known multiple testing issue that arise with multiple p-values: here a reference for this issue for instance (Jafari and Ansari-Pour, 2019).

As mentioned in the Materials and Methods section on line 1563, The FDR is classically fixed to 1% in the literature because it means we have below 1% of false discoveries in the listed proteins.

b. IMMT, also called mitofilin, is MICOS complex subunit Mic60 and it is one of the major components of inner mitochondrial membrane organizing cristae junction formation and distribution. It is detected by 76 peptides and its $\log_2(\text{Mean.in.KO}/\text{Mean.in.WT})$ value is -1.753952004 and the KO/WT ratio is 0.30. Three-fold reduction in this protein would affect mitochondrial respiration and apoptosis described in this paper.

The list can continue like this, as mentioned above authors need to evaluate their data more carefully before proposing a new role for MTFP1 in ANT-dependent uncoupling of inner membrane and proton leakage.

We did not observe significant decrease in the steady state levels of any MICOS subunit, including MIC60 (IMMT) and we hope that clarifications we have made to Dataset EV1 (see previous response in regards to ATP5E) are sufficient. As indicated above, we believe that confusion arose due to the nomenclature used in Dataset EV1: Column M, which was previously entitled "Differential?" has now been renamed "Differentially Expressed?" and the values "1" and "0" have been replaced with "Yes" and "Not significant". This updated table now more more clearly displays the mitochondrial proteins (Column N: MouseMitoCarta) that are statistically significantly dysregulated. We have also included a volcano plot of these data clearly indicating that IMMT does not meet the statistical criteria for differential protein expression.

We agree with the reviewer that loss of mitofilin function could contribute to the impairment of the mitochondrial respiration and promote cell death given its role in cristae maintenance. While we could exclude a role of MIC60 depletion in our model, we still decided to assess mitochondrial structure by transmission electron microscopy in mice at the symptomatic phase. **These new data (not previously included in the manuscript)** of cMKO hearts and littermate controls show disruption of cristae and alterations in mitochondrial surface area (Fig. S3D-E), defects that are absent in pre-symptomatic hearts (Fig. 3C-D). These ultrastructural changes have been observed in non-mitochondrial genetic models of cardiomyopathy (Gupta et al., 2010), we therefore hypothesize this to be the result of ongoing cardiac remodeling and dysfunction.

Our **new cardiac proteomics data** from pre-symptomatic mice show no loss in the levels of any of the MICOS complex proteins (Fig. S2A, right). Since we observe no defects in cristae or maximal respiration rates (state 3) in mitochondria from pre-symptomatic cMKO mice, we are unable to conclude that MICOS defects underscore the initiation of cardiomyopathy, although they may contribute to disease progression.

2) It would be nice to compare complex I, II, and IV oxygen consumption rates in pre-symptomatic as shown for 18- and 34-week-old KO and WT mice for direct comparison. Steady-state expression of oxidative phosphorylation complexes should also be compared for these mice prior to make such a significant claim in addition to the respiratory studies.

We thank the reviewer and we understand this point. Fig. 2B-C show the State 3 respiration of cardiac mitochondria isolated from 18- or 34-week old WT and cMKO which it has been assessed in presence of Complex I, II or IV substrate under ADP-stimulated phosphorylating conditions. The same bioenergetic measurement, has been done on cardiac mitochondria from pre-symptomatic mice (Fig. 2D, S2F), but further characterized by assessing also State 2 and State 4 respirations together with State 3.

Following up on this very reasonable request, we performed new 1D-BN-PAGE studies of the OXPHOS complexes in both pre-symptomatic and DCM-stage mice (Fig. S2C). We observed no alterations in OXPHOS complexes at pre-symptomatic phase (8-10 weeks), whereas we did observe a modest reduction of Complex III and Complex V complexes at symptomatic stage (30 week), which is consistent with the reduced mitochondrial respiration observed in cMKO mice at 18 and 34 weeks of age (Fig. 2B-C).

3) *It would be interesting to see steady-state expression of ANT1 against other mitochondrial relevant loading markers rather than vinculin (Fig 2G). Also, ANT1 is one the most abundant IM proteins in mitochondria; interestingly, only ANT3 was detected in the proteomics analysis (dataset EV1).*

Our new MS proteomics data of cardiac tissue isolated from pre-symptomatic cMKO mice also confirm that ANT1 and ANT2 (encoded by *Slc25A4* and *Slc25A5*, respectively) are not upregulated above the 1.5 fold threshold that is set for significantly dysregulated proteins (Fig. S2A, right). These new data are included in Dataset EV1, following the same nomenclature and layout as for the cardiac proteome LFQ data generated for symptomatic (DCM-stage) mice. We have also performed new western blots with additional mitochondrial loading controls such as OPA1 and FIS1, which clearly demonstrate no significant differences in ANT1 expression.

4) *The difference, if there is any, between Figs 2D, 2H, and 2I needs to be clarified.*

We apologize for the confusion with these figures. We have revisited the graphical representation of these data to illustrate and clarify the essential difference caused by MTFP1 ablation in cardiomyocytes. Fig. 2D shows Complex I driven oxygen consumption rates (JO_2) of WT and cMKO mitochondria isolated from pre-symptomatic mice. JO_2 was measured in presence of pyruvate, malate, and glutamate (PGM) (state 2) followed by the addition of ADP (state 3) and Oligomycin (Omy- state 4). We observed normal state 3 and elevated JO_2 levels in State 2, State 4 and decreased respiratory ratio controls (State3:2, 3:4 RCRs) suggesting mitochondrial uncoupling. Subsequent addition of the ANT inhibitor Carboxyatractyloside (CATR) was able to rescue the elevated state 4 respiration (Fig. 2D left), the decreased RCRs (Fig. 2D, middle, right) and lowered membrane potential (Fig. 2E). Fig. 2H and 2I show that CATR can rescue proton leak if it is used in place of Omy, rather than after Omy, normalizing the State 3:4 ratio. Finally, we have added new data demonstrating that the addition of CATR *before* the addition of PGM and ADP is capable of rescuing proton leak (Fig. 2J-K). These experiments indicate that ANT-dependent proton leak is the cause of mitochondrial uncoupling in *Mtfp1* deficient cardiac mitochondria.

5) *Studies performed for programmed cell death and opening of mPTP using primary cardiomyocytes and cell lines are complementary to what proposed in the manuscript.*

We agree that there is concordance between the studies we have performed in primary cardiomyocytes and cell lines in so far as the mPTP and cell death are concerned. This has allowed us to genetically demonstrate that the mPTP regulator Cyclophilin D (CYPD) (encoded by *Ppif*) is cable of rescuing enhanced mPTP opening and cell death sensitivity caused by the ablation of MTFP1. We have now added new data in HL-1 cells, an established mouse cardiomyocyte cell line, demonstrating that here too, MTFP1 ablation increases programmed cell death sensitivity which is dependent on CYPD (Fig. S5A-D).

6) The difference between the median lifespan of the male and female mice is also interesting. The authors claimed that the MTFP1 protects against proton leak across the inner membrane in ANT-dependent manner and its loss causes the age dependent decline in cardiac energy metabolism in ANT-dependent manner in mice. If the mechanism proposed in this article universal and proved in multiple model systems, why the lifespan between the male and female mice is so different? Figure 6 and the title also refers to a general mechanism in mice. This phenomenon needs to be addressed more carefully before generalizing their findings.

We thank the reviewer for raising this observation and this interesting point. Our study demonstrates that MTFP1 loss in cardiac mitochondria drives to DCM in both genders, with male mice living shorter than females. To date we do not know yet why female mice survive longer than male mice but sexual dimorphisms in cardiovascular diseases (CVDs) have been reported in both humans and experimental and transgenic models of heart diseases, with male more susceptible to CVDs than females (Kabir et al., 2021; Kessler et al., 2019; Olsson et al., 2001; Ozierański et al., 2021). It has been shown that estrogens can impact mitochondrial function (Chen and Yager, 2004; Chen et al., 2004) and that sex influences the expression of mitochondrial oxidative phosphorylation, fatty acid oxidation, TCA cycle genes (Vijay et al., 2015). However, whether these sexual dimorphisms have functional effects and contribute differently to the cardiac decline and survival during heart diseases remains largely unknown and a field to be certainly explored.

To be clear, we have not claimed that the ANT-dependent mechanism is universal nor proved in multiple model systems. So far, we only observe ANT-dependent uncoupling upon loss of MTFP1 in cardiomyocyte-specific knockout mice (cMKO) while ablation of MTFP1 in cultured cells (HL-1, Fig. S5E-J, MEFs; Fig. S2L-O, and U2OS; Fig. S2O-R) does not cause uncoupling. What is clearly consistent across our animal model and cultured cell line models is that MTFP1 increases programmed cell death sensitivity.

Finally, we agree with the reviewer that the title of the paper may benefit from increased precision, so we have modified to read "Loss of Mitochondrial Fission Process 1 (MTFP1) *in cardiomyocytes* promotes mitochondrial uncoupling, cell death and heart failure in mice"

Reviewer #4

In this manuscript, Erminia DONNARUMMA et al described the new roles of MTFP1, as a critical regulator of mitochondrial coupling in cardiac homeostasis, which are distinct from its previous role in mitochondrial fission. MTFP1 deletion in post-natal cardiomyocytes leads to membrane potential dissipation associated to mPTP opening, cell death, and a progressive DCM developing into heart failure and middle-aged death. The amount and quality of data is enough to insist their claim describing in this manuscript. The topic of MTFP1 as a valuable tool for the molecular dissection of mitochondrial uncoupling and mPTP function has emerged as a novel topic in heart diseases, and I believe that this paper will contribute to this new research direction. Here, I have a few concerns need to be addressed prior to next consideration.

1. Authors showed no changes in basal or maximal respiration rates nor any evidence of mitochondrial uncoupling in cultured MTFP1-deficient MEFs (Mtfp1^{-/-}) and U2OS cells

(*Mtfp1*^{-/-}). Whether is MTFP1-dependent proton leak also found in cultured cardiomyocyte cells or not, such as rat cardiomyocyte cells (H9c2) and human cardiomyocyte-like cells from fibroblasts (HCFs)? As authors suggested that MTFP1-dependent proton leak may be cell type specific, the data showing in other myocardial cells of different species is required to strengthen their claims.

We thank the reviewer for this suggestion, which we have experimentally pursued. We have now **included new data in HL-1 cells**, an established mouse atrial cardiomyocyte cell line, demonstrating that here too, MTFP1 ablation increases programmed cell death sensitivity which is dependent on CYPD, encoded by *Ppif* (Fig. S5A-D). However, HL-1 cells devoid of MTFP1, like MEFs and U2OS cells, show no altered bioenergetic profiles (Fig. S5E-J), suggesting that these phenotypes can in fact be functionally separated. These observations lead us to conclude that proton leak and bioenergetic inefficiency observed in our cMKO mouse model is indeed specific to cardiac mitochondria and cardiomyocytes which is consistent with the different mitochondria cristae density and organization between primary cardiomyocytes and immortalized HL-1 cells, as shown in the figure below.

Electron tomography of vitreous sections from cultured mammalian cells
Gruska et al. 2008

2. In Figure 3, the authors report that no indications of mitochondrial elongation (median mitochondrial surface area: WT 3198 μm^2 versus cMKO 2954 μm^2) nor altered cristae organization (Figure 3C-D).

But there is a statistically significant difference in the violin plot of mitochondrial surface area within cardiac posterior wall measured in WT and cMKO mitochondria, as $**p < 0.01$ Mann-Whitney test. Please clarify it.

The reviewer is correct. Mitochondrial surface area was calculated by manually segmenting each mitochondrion using iMOD software in TEM images. These data show that median mitochondrial area is reduced by 8% in cMKO hearts relative to WT hearts. While our data suggest a tendency towards mitochondrial fragmentation in KO mitochondria, we are reticent to insist on the physiological relevance of this modest difference because it corresponds to ~8% reduction, which is far more modest than what we have described for bona fide mitochondrial fragmentation previously (Wai et al., 2015). Therefore, we believe that the statement that mitochondrial elongation was not observed is therefore factually correct.

3. The confocal images of *Mtfp1*^{-/-} MEFs treated with *Mtfp1* siRNA was missing in Figure 3I.

There is no missing confocal image as we did not attempt to deplete *Mtfp1* by siRNA in cells that we already proved were genetically devoid of *Mtfp1*.

4. In Figure 4, the authors showed that MTFP1 was essential for cell survival using the adult cardiomyocytes from 8-10-week-old WT and cMKO mice. How about the cell sensitivity under the treatment of CCCP or H₂O₂ in cardiomyocytes isolated from juvenile mice?

We thank the reviewer for this comment. We have performed cell death assays under CCCP and H₂O₂ treatment on primary adult cardiomyocytes isolated from pre-symptomatic young mice (Fig. 4A-B and 4C-D). The age range of juvenile mice is from 3 to 8 weeks, so our existing experiments address this question. Nevertheless, we may speculate that since MTFP1 is ablated from the perinatal stage via the Cre-mediated deletion of the conditional *Mtfp1* allele in cMKO mice, cardiomyocytes lacking MTFP1 at any post-natal stage would have an increased sensitivity to programmed cell death.

5. I didn't find the description of Figure 6 in the manuscript, where is it?

Figure 6 is now Figure 7 and is called out in the manuscript on lines 432 and 557. The figure legend is on page 29 of the manuscript, line 814.

6. There are several typos or formatting issues throughout the manuscript. Authors should go through their manuscript carefully.

We have corrected typos and formatting issues: amendments have been highlighted in yellow.

References

Andreyev AYu, Bondareva, T.O., Dedukhova, V.I., Mokhova, E.N., Skulachev, V.P., Tsofina, L.M., Volkov, N.I., and Vygodina, T.V. (1989). The ATP/ADP-antiporter is involved in the uncoupling effect of fatty acids on mitochondria. *Eur. J. Biochem.* 182, 585–592. .

Bertholet, A.M., Chouchani, E.T., Kazak, L., Angelin, A., Fedorenko, A., Long, J.Z., Vidoni, S., Garrity, R., Cho, J., Terada, N., et al. (2019). H⁺ transport is an integral function of the mitochondrial ADP/ATP carrier. *Nature* 571, 515–520. .

- Bertholet, A.M., Natale, A.M., Bisignano, P., Suzuki, J., Fedorenko, A., Hamilton, J., Brustovetsky, T., Kazak, L., Garrity, R., Chouchani, E.T., et al. (2022). Mitochondrial uncouplers induce proton leak by activating AAC and UCP1. *Nature* 606, 180–187. .
- Bonora, M., Giorgi, C., and Pinton, P. (2021). Molecular mechanisms and consequences of mitochondrial permeability transition. *Nat. Rev. Mol. Cell Biol.* <https://doi.org/10.1038/s41580-021-00433-y>.
- Brustovetsky, N.N., Amerkanov, Z.G., Yegorova, M.E., Mokhova, E.N., and Skulachev, V.P. (1990). Carboxyatractylate-sensitive uncoupling in liver mitochondria from ground squirrels during hibernation and arousal. *FEBS Lett.* 272, 190–192. .
- Chavez, J.D., Tang, X., Campbell, M.D., Reyes, G., Kramer, P.A., Stuppard, R., Keller, A., Zhang, H., Rabinovitch, P.S., Marcinek, D.J., et al. (2020). Mitochondrial protein interaction landscape of SS-31. *Proc. Natl. Acad. Sci. U. S. A.* 117, 15363–15373. .
- Chen, J.-Q., and Yager, J.D. (2004). Estrogen's effects on mitochondrial gene expression: mechanisms and potential contributions to estrogen carcinogenesis. *Ann. N. Y. Acad. Sci.* 1028, 258–272. .
- Chen, J.Q., Delannoy, M., Cooke, C., and Yager, J.D. (2004). Mitochondrial localization of ERalpha and ERbeta in human MCF7 cells. *Am. J. Physiol. Endocrinol. Metab.* 286, E1011-22. .
- Chiao, Y.A., Zhang, H., Sweetwyne, M., Whitson, J., Ting, Y.S., Basisty, N., Pino, L.K., Quarles, E., Nguyen, N.-H., Campbell, M.D., et al. (2020). Late-life restoration of mitochondrial function reverses cardiac dysfunction in old mice. *Elife* 9. <https://doi.org/10.7554/eLife.55513>.
- Gupta, A., Gupta, S., Young, D., Das, B., McMahon, J., and Sen, S. (2010). Impairment of ultrastructure and cytoskeleton during progression of cardiac hypertrophy to heart failure. *Lab. Invest.* 90, 520–530. .
- Heide, H., Bleier, L., Steger, M., Ackermann, J., Dröse, S., Schwamb, B., Zörnig, M., Reichert, A.S., Koch, I., Wittig, I., et al. (2012). Complexome profiling identifies TMEM126B as a component of the mitochondrial complex I assembly complex. *Cell Metab.* 16, 538–549.
- Jafari, M., and Ansari-Pour, N. (2019). Why, When and How to Adjust Your P Values? *Cell J.* 20, 604–607. .
- Kabir, R., Sinha, P., Mishra, S., Ebenebe, O.V., Taube, N., Oeing, C.U., Keceli, G., Chen, R., Paolucci, N., Rule, A., et al. (2021). Inorganic arsenic induces sex-dependent pathological hypertrophy in the heart. *Am. J. Physiol. Heart Circ. Physiol.* 320, H1321–H1336. .
- Karch, J., Kwong, J.Q., Burr, A.R., Sargent, M.A., Elrod, J.W., Peixoto, P.M., Martinez-Caballero, S., Osinska, H., Cheng, E.H.-Y., Robbins, J., et al. (2013). Bax and Bak function as the outer membrane component of the mitochondrial permeability pore in regulating necrotic cell death in mice. *Elife* 2, e00772. .
- Kessler, E.L., Rivaud, M.R., Vos, M.A., and van Veen, T.A.B. (2019). Sex-specific influence on cardiac structural remodeling and therapy in cardiovascular disease. *Biol. Sex Differ.* 10, 7. .
- Mourier, A., Ruzzenente, B., Brandt, T., Kühlbrandt, W., and Larsson, N.-G. (2014). Loss of LRPPRC causes ATP synthase deficiency. *Hum. Mol. Genet.* 23, 2580–2592. .

Olsson, M.C., Palmer, B.M., Leinwand, L.A., and Moore, R.L. (2001). Gender and aging in a transgenic mouse model of hypertrophic cardiomyopathy. *American Journal of Physiology-Heart and Circulatory Physiology* 280, H1136–H1144. .

Ozierański, K., Tymińska, A., Skwarek, A., Kruk, M., Koń, B., Biliński, J., Opolski, G., and Grabowski, M. (2021). Sex Differences in Incidence, Clinical Characteristics and Outcomes in Children and Young Adults Hospitalized for Clinically Suspected Myocarditis in the Last Ten Years-Data from the MYO-PL Nationwide Database. *J. Clin. Med. Res.* 10. <https://doi.org/10.3390/jcm10235502>.

Shabalina, I.G., Kramarova, T.V., and Nedergaard, J. (2006). Carboxyatractyloside effects on brown-fat mitochondria imply that the adenine nucleotide translocator isoforms ANT1 and ANT2 may be responsible for basal and fatty ... *Biochemical*.

Solem, L.E., Henry, T.R., and Wallace, K.B. (1994). Disruption of mitochondrial calcium homeostasis following chronic doxorubicin administration. *Toxicol. Appl. Pharmacol.* 129, 214–222. .

Symersky, J., Osowski, D., Walters, D.E., and Mueller, D.M. (2012). Oligomycin frames a common drug-binding site in the ATP synthase. *Proc. Natl. Acad. Sci. U. S. A.* 109, 13961–13965. .

Vijay, V., Han, T., Moland, C.L., Kwekel, J.C., Fuscoe, J.C., and Desai, V.G. (2015). Sexual dimorphism in the expression of mitochondria-related genes in rat heart at different ages. *PLoS One* 10, e0117047. .

Wai, T., García-Prieto, J., Baker, M.J., Merkwirth, C., Benit, P., Rustin, P., Rupérez, F.J., Barbas, C., Ibañez, B., and Langer, T. (2015). Imbalanced OPA1 processing and mitochondrial fragmentation cause heart failure in mice. *Science* 350, aad0116. .

Zhang, H., Alder, N.N., Wang, W., Szeto, H., Marcinek, D.J., and Rabinovitch, P.S. (2020). Reduction of elevated proton leak rejuvenates mitochondria in the aged cardiomyocyte. *Elife* 9. <https://doi.org/10.7554/eLife.60827>.

REVIEWERS' COMMENTS

Reviewer #1 (Remarks to the Author):

None

Reviewer #2 (Remarks to the Author):

We appreciated the careful revision, which addressed all the points we made. We commend the Authors for the excellent job and the important contribution.

Paolo Bernardi and Michela Carraro

Reviewer #4 (Remarks to the Author):

The authors have addressed all my concerns.